# PAF15–PCNA exhaustion governs the strand-specific control of DNA replication

Gita Chhetri[1], Sugith Babu Badugu[1], Narcis-Adrian Petriman[1], Mikkel Bo Petersen[1], Aylin Seren Güller[1], Nora Fajri[2], Manon Coulée[2], Ganesha Pandian Pitchai[1], Jan Novotný[1], Frederik Tibert Larsen[1,3], Andreas Fønss Møller[1,4], Morten Frendø Ebbesen[1], Tina Ravnsborg[1], Anoop Kumar Yadav[5,6], Barath Balarasa[1], Anita Lunding[1], Hana Polasek-Sedlackova[5], Ole N. Jensen[1], Kim Ravnskjaer[1,3], Jonathan R. Brewer[1,3], Jesper Grud Skat Madsen[1,3,4], Nataliya Petryk[2], Jens S. Andersen[1] & Kumar Somyajit[1✉]

Eukaryotic genome replication is surveyed by the S-phase checkpoint, which coordinates sequential origin activation to prevent the exhaustion of poorly defined, rate-limiting replisome components[1–3]. Here we show that excessive origin firing saturates chromatin-bound proliferating cell nuclear antigen (PCNA)—a sliding clamp for DNA polymerase processivity and Okazaki fragment processing[4]—thereby restricting further PCNA loading and lagging-strand synthesis when checkpoint control is lost. PCNA-associated factor 15 (PAF15) emerges as a dosage-sensitive regulator of this process[5–9]. During unperturbed S phase, the entire soluble PAF15 pool binds to chromatin, leaving no reserve to stabilize PCNA under conditions of excessive origin activation. PAF15 binds to PCNA specifically on the lagging strand through a high-affinity PIP motif and occupies the DNA-encircling channel, protecting the clamp and associated enzymes from premature unloading by the ATAD5–RFC complex. Conversely, overexpression of PAF15 or forced redistribution to the leading strand disrupts replisome progression and induces cell death. These detrimental effects are mitigated by Timeless–Claspin, which blocks PAF15–PCNA binding on the leading strand. E2F4-mediated repression fine-tunes PAF15 expression to ensure optimal dosage and strand specificity. These findings reveal a previously unrecognized replisome constraint: when PAF15–PCNA assemblies are exhausted, the S-phase checkpoint globally restricts origin activation, linking a strand-specific rate-limiting mechanism to global replication dynamics.

Error-free genome duplication is essential for precise genetic inheritance and for protection against genomic instability—a driver of oncogenic transformation and cancer progression[10,11]. In eukaryotes, such precision is maintained by two fundamental controls: the number of active replication origins and the velocity of active replisomes[12,13]. The DNA replication program is established in G1 by the licensing of an excess pool of origins, marked by the loading of inactive minichromosome maintenance (MCM) helicases 2–7 onto chromatin. During S phase, only around 10% of these origins are activated to form active replisomes, in which MCM2–MCM7 assemble with CDC45 and GINS1–GINS4 (Go-Ichi-Ni-San 1–4 subunits) to form the CMG helicase, together with DNA polymerases and additional regulatory proteins[14]. The S-phase checkpoint, governed by the ataxia telangiectasia and RAD3-related (ATR) pathway, coordinates origin firing with ongoing synthesis in cells both in unperturbed S phase and under replication stress, thereby preventing premature or excessive origin activation and untimely mitotic entry[1,15]. Whereas insufficient origin activation causes DNA under-replication[16], excessive origin firing exhausts replication resources and destabilizes the genome[2,3]. Thus, we hypothesize that the S-phase checkpoint operates within boundaries set by as-yet-unidentified replisome dynamics that define a global replication capacity and prevent replication from exceeding this threshold.

## Unscheduled origin activation affects DNA replication

To test this hypothesis, we examined whether rapid origin firing would deplete core replisome components and thereby reveal the key rate-limiting steps of global genome replication. Therefore, we applied brief ATR inhibition to induce rapid and aberrant cyclin-dependent kinase (CDK) activation in a cell line expressing a TagRFP chromobody targeting endogenous PCNA, a proxy for active replisomes and local replication dynamics[17] (Fig. 1a). Using quantitative image-based cytometry (QIBC)[2,18], we first confirmed that short-term inhibition of ATR (40 min) robustly activates CDKs during S phase, as indicated by phosphorylation of FOXM1—an established CDK1 and CDK2 (hereafter, CDK1/2) target[1]—without triggering a DNA damage response, as evidenced by the

[1]Department of Biochemistry and Molecular Biology, University of Southern Denmark, Odense, Denmark. [2]Genome Integrity and Cancers, Gustave Roussy, Université Paris-Saclay, CNRS, Villejuif, France. [3]Center for Functional Genomics and Tissue Plasticity (ATLAS), University of Southern Denmark, Odense, Denmark. [4]Novo Nordisk Foundation Center for Genomic Mechanisms of Disease, Broad Institute of MIT and Harvard, Cambridge, USA. [5]Institute of Biophysics of the Czech Academy of Sciences, Brno, Czech Republic. [6]Department of Experimental Biology, Faculty of Science, Masaryk University, Brno, Czech Republic. ✉e-mail: ksom@sdu.dk

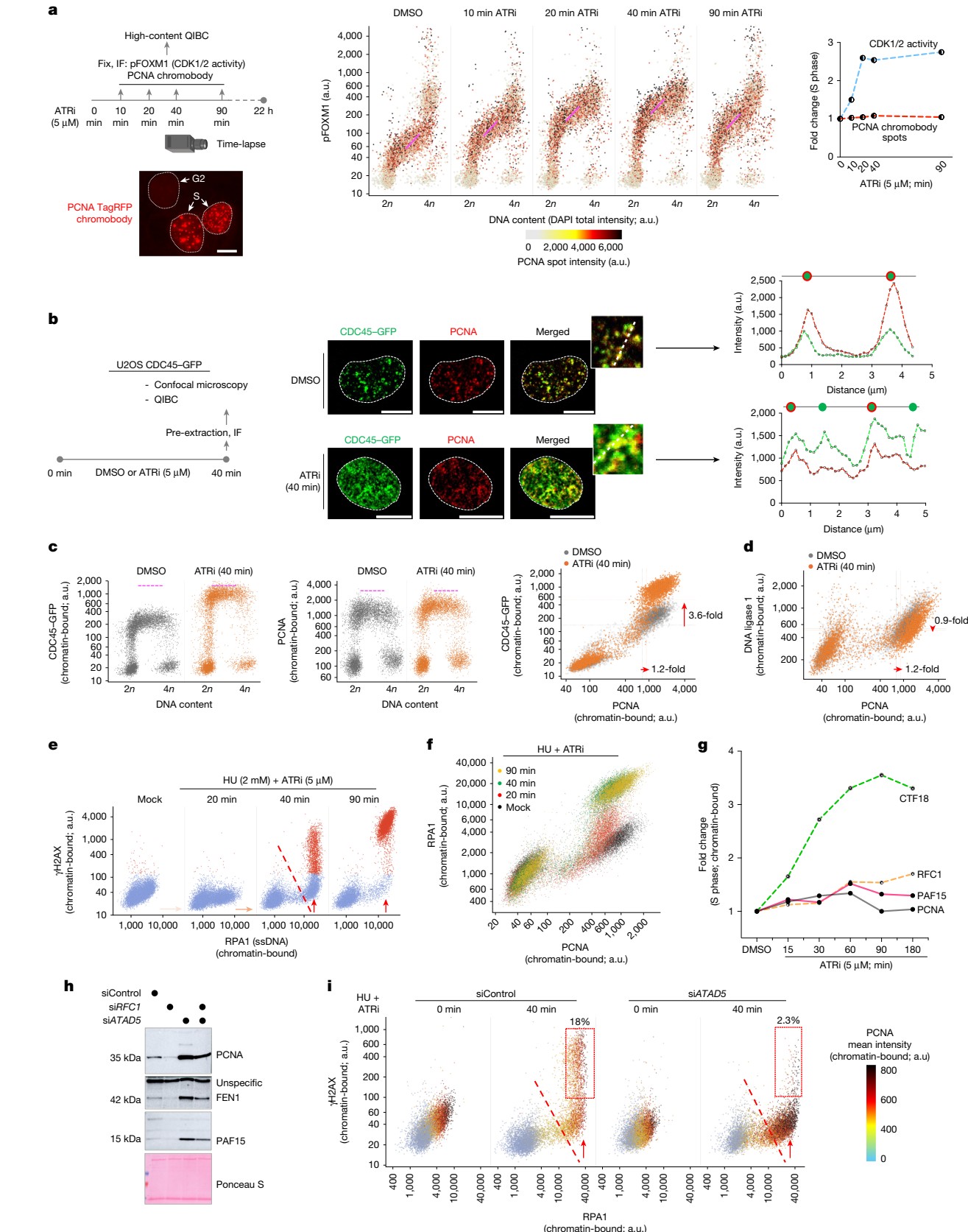

**Fig. 1** | See next page for caption.

absence of pan-nuclear γH2AX (phosphorylated histone H2AX) (Fig. 1a, Extended Data Fig. 1a and Supplementary Fig. 1a,b). As anticipated[19], such ATR inhibition led to a pronounced increase in active fork density

(Extended Data Fig. 1b) and a corresponding reduction in replication fork speed (Extended Data Fig. 1c). However, despite increased CDK activity and abrupt activation of new origins, we did not observe an

increase in PCNA chromobody foci after ATR inhibition, as assessed through QIBC of a fixed cell population and live-cell tracking (Fig. 1a and Supplementary Fig. 1c).

**Fig. 1 | Excess replication origins drive the depletion of chromatin-bound PCNA and lagging-strand activities. a**, Left, workflow using U2OS cells stably expressing the PCNA TagRFP chromobody. Scale bar, 10 μm. ATRi, ATR inhibitor; IF, immunofluorescence; pFOXM1, FOXM1 phosphorylated at Thr600. Middle, QIBC of pFOXM1 in cells treated with dimethyl sulfoxide (DMSO) or exposed to ATRi for the indicated durations. Nuclear DNA was counterstained with DAPI (4′,6-diamidino-2-phenylindole), and total DAPI intensity was used to infer DNA content (2*n*, G1; 4*n*, G2). *n* > 8,000 cells per condition. The colour gradient denotes mean PCNA TagRFP chromobody intensity in individual nuclei. a.u., arbitrary units. Right, fold change in CDK1/2 activity (pFOXM1 intensity) and number of PCNA chromobody spots, normalized to the DMSO-treated condition. **b**, Left, workflow. Middle, three-dimensional confocal images of endogenously tagged CDC45–GFP U2OS cells immunostained for PCNA. Scale bars, 10 μm. Right, intensity profiles from the insets for CDC45–GFP (green) and PCNA (red) within replication foci. **c**,**d**, QIBC of CDC45, PCNA (**c**) and DNA ligase 1 (**d**) chromatin binding (*n* > 5,000 cells). Pink dotted lines in the scatter plots denote

approximate upper limits of chromatin-bound protein levels. Fold changes in S-phase-specific chromatin association are shown. **e**,**f**, QIBC of RPA1 and γH2AX (**e**), and of RPA1 and PCNA (**f**), in U2OS cells under the indicated conditions (*n* > 8,000 cells). The dotted line marks the onset of replication catastrophe. ssDNA, single-stranded DNA. **g**, Fold change on chromatin for the indicated proteins during ATRi treatment relative to DMSO (from data in Supplementary Fig. 2d,e). **h**, Western blot of chromatin from U2OS cells treated with the indicated short interfering RNAs (siRNAs) for 48 h. si*ATAD5*, *ATAD5* siRNA1; si*RFC1*, *RFC1* siRNA. **i**, QIBC of chromatin-bound RPA1 and γH2AX in control or ATAD5-depleted U2OS cells treated with hydroxyurea (HU) or ATRi. The colour gradient reflects the PCNA chromatin-bound intensity. The dotted line marks the onset of replication catastrophe; the red boxes indicate cells undergoing catastrophe (*n* > 10,000 cells). The QIBC plots shown are representative of at least two independent biological replicates. The camera icon in **a** was created with BioRender.com. Somyajit, K. (2025) https://BioRender.com/ergwc0j.

Independent to the PCNA chromobody-specific readout, anti-PCNA immunostaining yielded identical results (Fig. 1b,c and Supplementary Fig. 1d). ATR inhibition led to a robust focal chromatin accumulation of early replisome components—including CDC45 of the CMG complex, the replicative DNA polymerases and the replication progression complex (RPC) factors Timeless and Claspin (Extended Data Fig. 1d). However, chromatin-bound PCNA showed only a modest (1.2-fold) increase, and did not scale with CDC45, which rose by 3.6-fold (Fig. 1b,c).

PCNA, a highly abundant homotrimeric sliding clamp, acts as a structural scaffold for continuous leading-strand synthesis by POLε[20], and coordinates the dynamic interplay among POLδ, FEN1 and DNA ligase 1 for the synthesis and maturation of short Okazaki fragments (OkFs)[21,22]. Notably, although core and leading-strand-specific factors were enriched on S-phase chromatin (Extended Data Fig. 1d), further analysis revealed that lagging-strand processes that depend on PCNA—such as those involving FEN1 and DNA ligase 1—did not increase and were instead mildly diminished on chromatin during excess origin activation (Fig. 1d and Extended Data Fig. 1e). PCNA also orchestrates epigenome maintenance by recruiting DNA methyltransferase 1 (DNMT1) to post-replicative nascent DNA[23]. Of note, the depletion of PCNA and lagging-strand factors from the chromatin pool was accompanied by a failure to load additional DNMT1 onto S-phase chromatin after ATR inhibition (Extended Data Fig. 1f), further highlighting a unique chromatin-level paucity of PCNA induced by unscheduled origin firing.

The functional exhaustion of OkF processing after ATR inhibition was confirmed using proximity ligation assay (PLA)–QIBC, which revealed reduced interactions between PCNA and DNA ligase 1 (Extended Data Fig. 1g), despite increased Timeless–RPA2 proximity in new replisomes (Extended Data Fig. 1g, right). Moreover, these results were corroborated by chromatin fractionation immunoblotting (Extended Data Fig. 1h,i) and QIBC across two additional cell lines (Extended Data Fig. 1j). Finally, the findings obtained using an ATR inhibitor were recapitulated by both WEE1 inhibition and Claspin depletion, both of which enhance CDK activity and are known to promote aberrant origin firing (Supplementary Fig. 2a–c).

Unligated OkFs are processed through a non-canonical pathway, which is mediated by PARP1[24]. This is evidenced by the accumulation of nascent, S-phase-specific polyADP-ribosylation (PAR) chains after short-term inhibition of poly(ADP-ribose) glycohydrolase (PARG)[24] (Extended Data Fig. 2a). Consistent with our observation that PCNA lagging-strand activities are depleted by excessive origin firing (Fig. 1c,d), even under normal replication conditions, inhibition of PARG triggered a rapid accumulation of PAR chains specifically in S phase (Extended Data Fig. 2b). This effect intensified with acute inhibition of ATR, but was completely suppressed by blocking dormant origin firing

(Extended Data Fig. 2b), indicating that unresolved lagging-strand intermediates after excessive origin firing drive the early activation of PARP1, before fork collapse (40 min of ATR inhibition). The essentiality of this non-canonical pathway is underscored by the fact that PARP1 inhibition and ATR or WEE1 inhibition are synthetic lethal (Extended Data Fig. 2c,d).

Premature origin firing and DNA replication stress have a catastrophic effect on the replicating genome, collectively known as 'replication catastrophe'[25]. Exhaustion of the available pool of genome-protective RPA protein marks the onset of such terminal replication collapse, especially when checkpoint failure coincides with DNA replication stress induced by hydroxyurea-triggered nucleotide depletion[2] (Fig. 1e). In light of our new findings, we compared the rate of RPA exhaustion with the chromatin paucity of PCNA. Notably, chromatin-bound PCNA and DNA ligase 1 were already saturated—or declined—before RPA exhaustion under hydroxyurea plus ATR inhibition (Fig. 1f and Extended Data Fig. 2e), suggesting that a loss of core replisome activity precedes RPA depletion in driving replication catastrophe.

Together, these results reinforce the notion that PCNA and lagging-strand processes undergo functional exhaustion during unperturbed origin activation and become further depleted upon unscheduled origin firing.

## Mechanism of chromatin-level PCNA depletion

To delve deeper, we combined mass spectrometry (MS) with TurboID-based biotinylation of PCNA-proximal proteins (Extended Data Fig. 2f–h) to identify factors that might influence its rate-limiting mechanisms. We focused on PCNA loaders, including canonical replication factor C 1–5 (RFC1–RFC5) and the CTF18-RFC variant—which encircle PCNA homotrimers on the lagging and on the leading strands, respectively[4,26]—and PCNA-associated factor 15 (PAF15, also known as PCLAF). PAF15 contains a high-affinity PCNA-interacting peptide (PIP) motif and has been implicated in restraining error-prone DNA polymerases during DNA damage[5–8,27], as well as regulating DNMT1 chromatin association through dual mono-ubiquitination of Lys15 and Lys24 by the E3 ligase UHRF1[28]. However, the direct role of PAF15 in unperturbed DNA replication remains unknown.

Mapping the dynamic range of origin activation after ATR inhibition, QIBC analysis revealed that PCNA—and its associated factor DNA ligase 1—accumulate on chromatin by no more than 1.3-fold, even as new origins continue to fire over a 60–90-min window, as evidenced by increasing levels of chromatin-bound Timeless and RPA (Fig. 1g and Supplementary Fig. 2d). Whereas the leading-strand PCNA loader CTF18 accumulated on replicating chromatin, RFC1—of the canonical RFC complex—showed only limited recruitment, revealing a bottleneck in PCNA loading during excessive origin activation, consistent with recent reports of RFC1 depletion under replication stress and

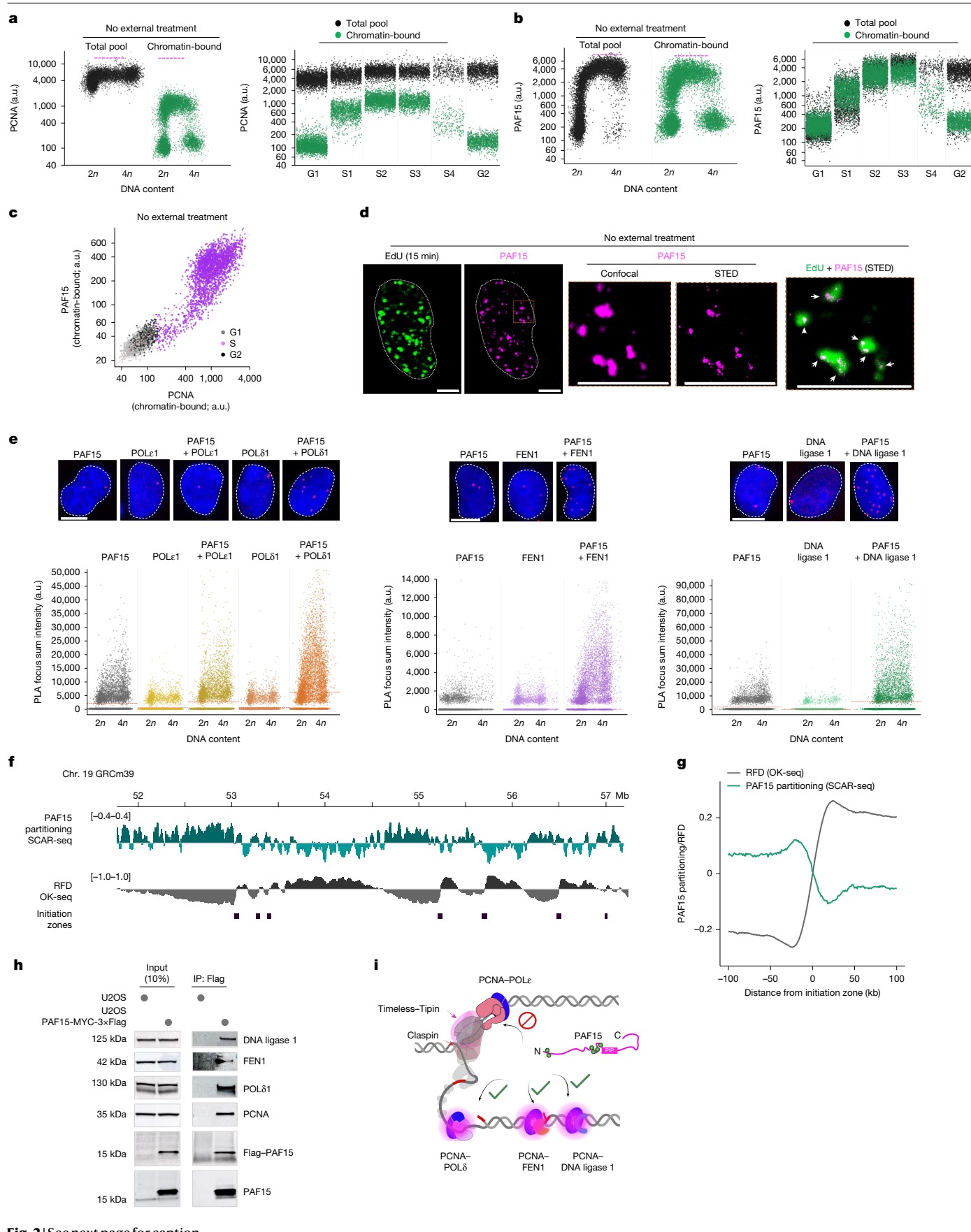

**Fig. 2** | See next page for caption.

**Fig. 2 | PAF15 is a dosage-limited, lagging-strand-specific PCNA factor.**
**a**,**b**, QIBC-based side-by-side imaging and quantification of total and chromatin-bound pools of PCNA (**a**) and PAF15 (**b**) in U2OS cells, together with their distribution across cell-cycle phases. Cell-cycle staging was determined using EdU incorporation and DAPI intensity. $n > 9{,}000$ cells analysed per condition. Pink dotted lines in the scatter plots denote the approximate maximum abundance of PCNA or PAF15. **c**, QIBC analysis showing the linear correlation between chromatin-bound PCNA and PAF15. More than 4,000 cells were quantified across two independent experiments. Cell-cycle phases are colour-coded: G1 (grey), S (purple) and G2 (black). **d**, Confocal and stimulated emission depletion (STED) microscopy of U2OS cells stained for PAF15 and labelled for active DNA synthesis using EdU. Scale bars, 5 µm. **e**, Representative images (top) and QIBC-based quantification (bottom) of PLA focus sum intensity for the indicated antibody pairs: PAF15–POLε1, PAF15–POLδ1, PAF15–FEN1 and PAF15–DNA ligase 1. PLA signals are plotted across total DNA content to visualize cell-cycle-stage-dependent interactions. $n > 8{,}000$ cells were analysed per condition. Scale bars, 10 µm. **f**, Example genomic locus showing replication fork directionality (RFD; grey) local PAF15 strand partitioning (green) and initiation zones (black) in E14 mouse embryonic stem cells. **g**, Average of PAF15 strand partitioning (green) and RFD (grey) around replication initiation zones. **h**, Immunoprecipitation of Flag-tagged PAF15 in naive U2OS cells and U2OS cells constitutively expressing PAF15-MYC-3×Flag. The indicated proteins from input and Flag immunoprecipitation lysates were analysed by western blot. **i**, Model depicting the presence of PAF15 at the lagging strand only, and its exclusion from the leading strand. Each QIBC plot shown is representative of at least two independent biological replicates. The model in **i** was created with BioRender. com. Somyajit, K. (2025) https://BioRender.com/yh2ia9e.

checkpoint loss[29,30]. Notably, PAF15 showed chromatin accumulation closely mirroring that of PCNA after ATR inhibition (Fig. 1g and Supplementary Fig. 2e), raising the possibility that its natural depletion exposes a rate-limiting control point for PCNA function after chromatin loading, particularly in the context of strand-specific DNA replication dynamics.

In line with this, depletion of the PCNA unloader ATAD5[31], which stabilizes PCNA, OkF processing factors and PAF15 on chromatin (Fig. 1h and Extended Data Fig. 2i–k), rescued replication catastrophe by preventing the loss of PCNA and DNA ligase 1 during excess origin firing (Fig. 1i and Extended Data Fig. 2l). These findings suggest that continuous PCNA unloading under excessive origin activation leads to chromatin depletion of PCNA, whereas exhaustion of key regulators such as PAF15 imposes a rate-limiting control.

## PAF15 controls PCNA during lagging-strand maturation

We next examined whether chromatin-bound PCNA dynamics depend on the natural paucity of soluble PCNA-associated factors such as PAF15. Quantitative, cell-cycle-resolved comparison of total and chromatin-bound pools showed that whereas PCNA and its effectors—including DNA ligase 1, DNA polymerases and CTF18—remained abundant, RFC1 exhibited notable paucity in total levels[29,30] (Fig. 2a, Extended Data Fig. 3a–e and Supplementary Fig. 3a). PAF15 was almost entirely chromatin-bound, with negligible soluble excess during normal origin firing (Fig. 2b), and this was consistent across cell types, antibodies and synchronization assays (Extended Data Fig. 3f–i).

Despite its low abundance, PAF15 constitutes a bona-fide component of the active replisome (Fig. 2c,d and Extended Data Fig. 3j,k). MS-based interactome analysis of chromatin-fractionated PAF15 suggested that PCNA is the main partner responsible for recruiting PAF15 to the replisome (Supplementary Fig. 3b). Consistently, the highly conserved PIP motif in PAF15 is essential for its localization to replicating chromatin (Extended Data Fig. 4a,b). By contrast, blocking APC/C (anaphase-promoting complex, also known as the cyclosome)- and CDH1-mediated degradation of PAF15 via its KEN-box[5] did not increase PAF15 chromatin loading (Extended Data Fig. 4a,b), suggesting that PAF15 assembly occurs downstream of origin firing, probably with additional regulation.

Because PCNA is essential for processive DNA replication on both strands, we next investigated which PCNA pool recruits PAF15 at the replisome. Using PLA to capture spatially proximal protein–protein interactions[32], we found that PAF15 specifically interacts with lagging-strand components (POLδ, FEN1 and DNA ligase 1) during S phase, but is excluded from leading-strand factors such as POLε (Fig. 2e). To independently assess the partitioning of PAF15 between the leading and lagging strands at replication forks, we used sister chromatid after replication sequencing (SCAR-seq)[33] and OkF sequencing (OK-seq)[34,35] in mouse embryonic stem (ES) cells.

Strand-resolved analyses revealed a pronounced enrichment of PAF15 in lagging-strand regions (Fig. 2f,g), inversely correlated with replication fork directionality (RFD) (Fig. 2f,g and Extended Data Fig. 4c–e). Co-immunoprecipitation analyses revealed the same pattern, showing that PAF15 associates with PCNA in a complex with lagging-strand factors[36], but not with the leading-strand polymerase POLε (Fig. 2h). These results show that PAF15 is associated predominantly with the lagging arms of replication forks (Fig. 2i). Although perhaps unexpected, this enrichment aligns with quantitative MS showing that the levels of PCNA exceed those of PAF15 by ninefold in total and by twelvefold on chromatin (Extended Data Fig. 4f), indicating that PAF15 is selectively recruited to a subset of PCNA that is engaged in lagging-strand synthesis.

PCNA, as a homotrimer, coordinates POLδ, FEN1 and DNA ligase 1 by functioning as a dynamic 'toolbelt'[4,8,21,37]. However, how lagging-strand factors avoid steric conflict during hand-off remains unclear. Structural alignment of the PCNA–POLδ–FEN1–DNA cryo-electron microscopy (cryo-EM) complex with the PCNA–PAF15–DNA crystal structure reveals severe clashes between PAF15 and POLδ–FEN1, indicating that PAF15 cannot coexist with these PCNA-bound enzymes (Extended Data Fig. 4g). By contrast, the sequential occupancy of PCNA subunits by POLδ, FEN1 and DNA ligase 1 is compatible with one or two PAF15 molecules (Fig. 2i, Extended Data Fig. 4g,h and Supplementary Fig. 4), suggesting that PAF15 anchors PCNA and coordinates orderly factor exchange to prevent steric interference. This model requires future structural validation.

PAF15 doubly monoubiquitinated by UHRF1 at Lys15 and Lys24 mediates DNMT1 recruitment during replication[28] and triggers the degradation of PAF15 after DNA damage[6]. Given that PAF15 is localized preferentially to the lagging strand, we tested whether this modification reflects strand-specific engagement. Acute depletion of *LIG1* or *RFC1* markedly reduced PAF15 ubiquitination (Extended Data Fig. 5a), and *Lig1* knockout (KO) mouse ES cells showed a complete loss of dual mono-ubiquitination (Extended Data Fig. 5b), confirming that both PAF15 and its ubiquitination are functionally coupled to lagging-strand synthesis. As previously shown, PAF15 ubiquitination is essential for DNMT1 loading[28]; accordingly, loss of this modification in UHRF1-depleted or *PAF15*-KO cells impaired DNMT1 recruitment during S phase, especially under treatment with decitabine (also known as 5-aza-2′-deoxycytidine) (Extended Data Fig. 5c). Conversely, loss of DNMT1 destabilized chromatin-bound PAF15, whereas depletion of UHRF1 did not affect the loading of either PAF15 or PCNA (Extended Data Fig. 5d). These findings suggest that PAF15 has a mono-ubiquitination-independent role in its association with PCNA, prompting us to investigate how PAF15 itself shapes PCNA dynamics on replicating chromatin.

Notably, *PAF15*-KO cells exhibited reduced levels of chromatin-bound PCNA—a phenotype validated across several cell types and methods of genetic ablation (Fig. 3a, Extended Data Fig. 5e,f and Supplementary Fig. 5a). To assess PCNA stability after rapid loss of PAF15, we C-terminally tagged endogenous PAF15 with AID–GFP. The tagged

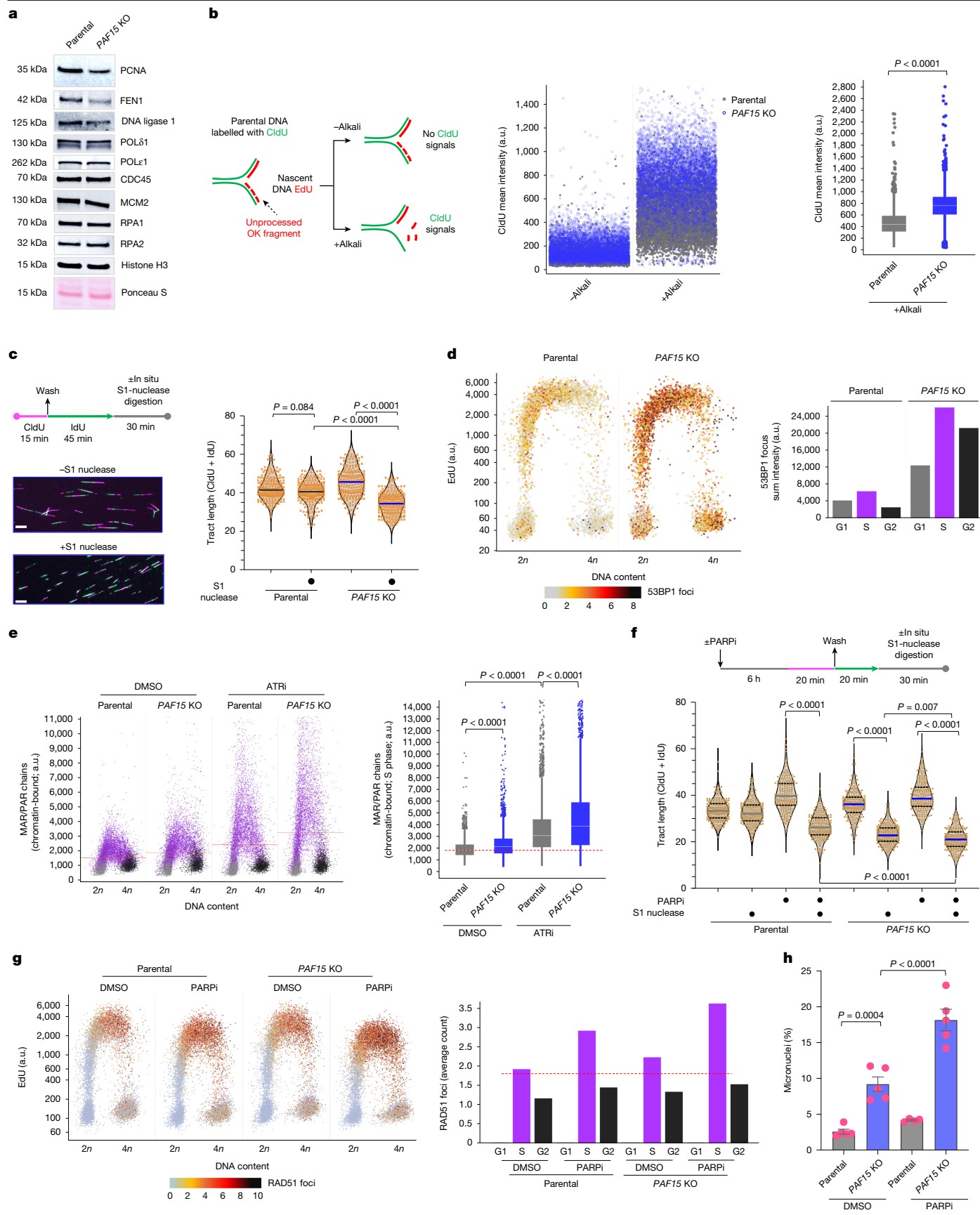

**Fig. 3 |** See next page for caption.

**Fig. 3 | Loss of PAF15 compromises OkF maturation and mounts non-lethal DNA replication stress. a**, Western blot of the indicated proteins from chromatin-purified extracts of parental and *PAF15*-KO U2OS cells. **b**, Left, schematic showing the dissociation of OkFs from nascent DNA (EdU) after alkali treatment. Middle, Rand plot from QIBC of parental (grey) and *PAF15*-KO (blue) cells with or without alkali. Right, box plot of CldU mean intensity (ssDNA) in both cell lines after alkali treatment. *n* > 10,000 cells analysed per condition. **c**, Top, DNA-fibre labelling protocol with in situ S1-nuclease digestion. Bottom, representative DNA fibres with or without S1 nuclease. Scale bars, 20 μm. Right, quantification of total nascent DNA-tract length in parental and *PAF15*-KO cells with or without S1 digestion (*n* = 200 fibres). **d**, Left, QIBC of 53BP1 nuclear bodies in the indicated cells, stratified by cell-cycle phases (*n* > 5,000 cells per condition). Right, distribution of 53BP1 bodies across G1, S and G2 phases in parental and *PAF15*-KO cells. **e**, QIBC analysis of mono- or poly-ADP-ribosylated (MAR/PAR) chain formation in *PAF15*-KO cells with ATR inhibition (*n* > 9,000 cells per condition). S-phase quantification (5,000 cells) is shown. **f**, Quantification of nascent DNA-tract length in parental and *PAF15*-KO cells treated with PARP inhibitor (PARPi; 500 nM) and digested with S1 nuclease (*n* = 200 fibres per condition). **g**, QIBC of RAD51 foci in parental and *PAF15*-KO cells with or without PARP inhibition (*n* > 9,000 cells per condition), shown across G1, S and G2 phases. **h**, Quantification of micronuclei in cells treated with PARPi (500 nM, 24 h), based on 500 nuclei per condition. Data are mean ± s.d.; *n* = 5 biological replicates. *P* values calculated by one-way ANOVA with Tukey's test. Each DNA fibre and QIBC plot shown is representative of at least two independent biological replicates.

protein did not fully recapitulate native PAF15, and showed impaired chromatin loading, probably because the green fluorescent protein (GFP) tag disrupts the structure of PAF15 and its interaction with PCNA (Supplementary Fig. 5b). Nevertheless, this resulted in hypomorphic PAF15 cells with markedly reduced PCNA chromatin stability, which was worsened by auxin-induced PAF15 degradation (Supplementary Fig. 5b).

We next asked whether the marked loss of chromatin-bound PCNA in PAF15-deficient cells alters replisome architecture. MS analysis of PCNA–TurboID showed that PAF15 deletion selectively disrupts PCNA's interactions with lagging-strand factors, whereas other replisome-associated contacts remain intact (Extended Data Fig. 5g). Chromatin fractionation confirmed that although PCNA is destabilized, the core replisome is preserved (Fig. 3a). Instead, OkF proteins such as FEN1 and DNA ligase 1 were selectively reduced (Fig. 3a). This was further validated by PLA–QIBC across thousands of S-phase cells, which revealed diminished PCNA–FEN1 and PCNA–DNA ligase 1 interactions (Extended Data Fig. 5h). Ectopic expression of PAF15 increased the levels of chromatin-bound PAF15 and co-enriched PCNA and DNA ligase 1 (Extended Data Fig. 5i), suggesting that PAF15 has a direct role in fostering lagging-strand factors.

Consistent with the notion that PAF15 shapes OkF maturation, single-molecule fibre assays showed that PAF15 loss increased fork speed but caused fork asymmetry (Extended Data Fig. 6a–c), a hallmark of defective OkF maturation[38]. Such impaired OkF maturation was further evidenced by alkali-sensitive, unligated OkFs (Fig. 3b) and S1-sensitive nascent DNA gaps (Fig. 3c). Moreover, PAF15-deficient cells showed the formation of 53BP1 foci (Fig. 3d and Extended Data Fig. 6d,e) and of micronuclei (Extended Data Fig. 6f), both of which are indicative of replication-associated DNA damage. Under these conditions—both during unperturbed replication and after checkpoint failure—PAF15-deficient cells relied on the non-canonical OkF maturation pathway, as evidenced by a marked increase in S-phase-specific ADP-ribosylation (Fig. 3e). Consequently, under PARP inhibition, loss of PAF15 caused extensive S1-sensitive daughter-strand gaps (Fig. 3f), RAD51-mediated post-replicative repair[39] (Fig. 3g) and ultimately lethal replication stress (Fig. 3h and Extended Data Fig. 6g).

Altogether, these results show that PAF15 stabilizes PCNA on chromatin to enable canonical OkF maturation and suppress replication stress. Its loss—or its natural exhaustion during excessive origin firing—disrupts lagging-strand processing, forcing cells to rely on PARP1-dependent repair pathways.

## Limited levels of PAF15 sustain PCNA stability

Next, we sought to uncover the mechanistic basis by which PAF15 physically and functionally safeguards chromatin-bound PCNA and regulates lagging-strand processing. PAF15 is an intrinsically disordered protein. Crystallographic and nuclear-magnetic-resonance studies of its extended PIP-box suggest that on binding to PCNA, PAF15 also accesses the DNA-encircling channel of PCNA[8,9] (Fig. 4a), although the functional importance of this remains unclear. To investigate, we used AlphaFold modelling[40] to analyse full-length PAF15 in complex with the PCNA homotrimer on DNA; this suggested that PAF15 acts as a molecular wedge that completely traverses the PCNA ring, exhibiting interactions that are distinct from those of other PIP-box proteins (Fig. 4a, Extended Data Fig. 6h and Supplementary Fig. 6).

APBS (Adaptive Poisson-Boltzmann Solver) software analysis of the modelled PCNA–PAF15–DNA complex revealed that each PAF15 molecule contributes with positively charged residues to both the inner PCNA ring and its N terminus, which is likely to stabilize its interaction with DNA (Fig. 4b). Given that purified human PCNA exhibits weak affinity for DNA in vitro and high rates of diffusion[41], we hypothesized that, owing to its unique interaction within the PCNA ring, PAF15 might enhance PCNA–DNA contact, and that the loss of this could explain the reduced levels of PCNA on chromatin.

Indeed, salt sensitivity assays on unperturbed replicating chromatin revealed that the loss of PAF15 rendered chromatin-bound PCNA more susceptible to increasing ionic strength (Extended Data Fig. 6i), indicating a reduced affinity of PCNA for DNA during replication. PCNA is topologically locked onto DNA after loading by the RFC clamp loader and is actively unloaded only by the ATAD5–RFC complex (Elg1–RFC in yeast)[31]. Thus, the reduced chromatin retention of PCNA observed in PAF15-deficient cells suggests that, in the absence of PAF15, PCNA becomes increasingly vulnerable to premature unloading by this pathway. ATAD5 depletion not only stabilized PCNA on chromatin in PAF15-deficient cells, but also reinforced our conclusion that PAF15 participates in lagging-strand processing (Fig. 4c and Extended Data Fig. 7a). Under these conditions, all tested OkF processing factors were selectively enriched on chromatin, whereas the leading-strand polymerase POLε1 remained unchanged (Fig. 4c). This was further corroborated by fluorescence recovery after photobleaching (FRAP), which showed that although depletion of PAF15 markedly increased the exchange rate of PCNA and shortened its chromatin residence time, co-depletion of ATAD5 and PAF15 restored PCNA immobility during S phase, mirroring the effect of ATAD5 depletion alone (Fig. 4d). This restoration of PCNA stability was accompanied by the functional recovery of lagging-strand processing and genome integrity, as evidenced by the re-establishment of PCNA–DNA ligase 1 interactions, detected by PLA–QIBC (Fig. 4e), a significant reduction in S1-sensitive daughter-strand gaps (Fig. 4f) and diminished 53BP1 foci in PAF15-deficient cells (Extended Data Fig. 7b). Of note, cells with co-depletion of PAF15 and ATAD5 maintained active DNA synthesis and proceeded into mitosis (Extended Data Fig. 7c), indicating partial preservation of replication dynamics and cell-cycle continuity.

These results establish PAF15 as a crucial lagging-strand-specific factor that maintains PCNA stability on replicating chromatin and safeguards lagging-strand-associated proteins from premature unloading by the ATAD5 complex (Fig. 4g).

## PAF15 binding to leading-strand PCNA is lethal

Building on the role of PAF15 in maintaining PCNA chromatin stability, we next asked whether its overexpression could overcome the intrinsic

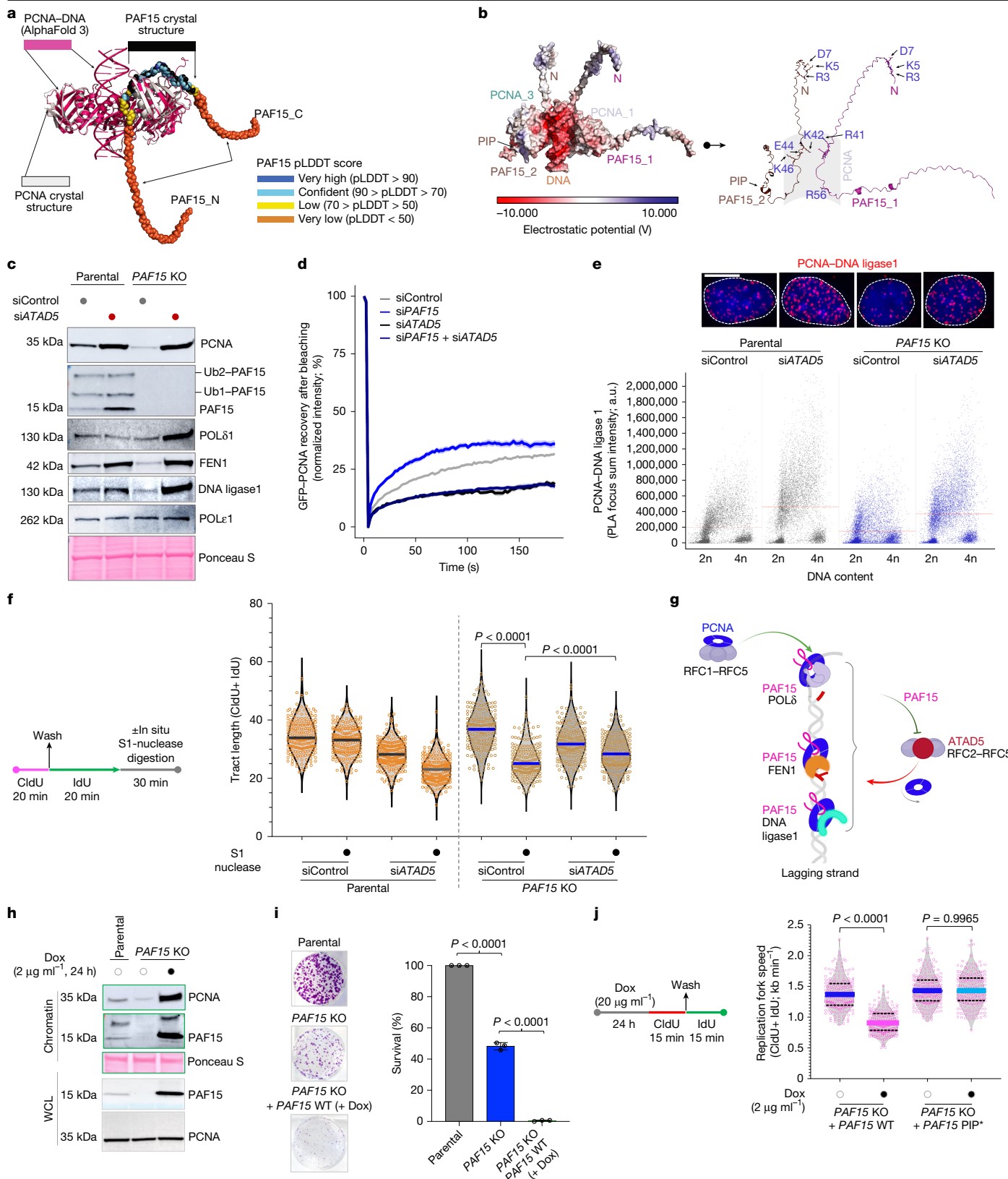

**Fig. 4** | See next page for caption.

rate-limiting nature of the endogenous PAF15 pool and modulate PCNA dynamics or lagging-strand activity. Short-term inducible overexpression of PAF15 in *PAF15*-KO cells rapidly stabilized chromatin-bound PCNA—an effect lost in the PAF15 PIP-box mutant—confirming the

requirement for PAF15–PCNA binding (Fig. 4h and Extended Data Fig. 7d,e). Deleting the first 11 amino acids containing the conserved H3-like basic region markedly reduced PAF15 stability and PCNA retention on chromatin (Extended Data Fig. 7f–h), showing that both the PIP

**Fig. 4 | PAF15 stabilizes PCNA and lagging-strand factors on chromatin, shielding them from unscheduled ATAD5-mediated unloading.**
**a**, Comparison between the PCNA–PAF15 crystal structure (Protein Data Bank (PDB): 6EHT, grey) and the AlphaFold 3-predicted model (pink). One PCNA and one PAF15 molecule are removed for clarity. PAF15 is coloured by per-residue confidence score (predicted local distance difference test; pLDDT). **b**, APBS electrostatic analysis of the modelled PCNA–PAF15–DNA complex containing three PCNA molecules (one removed), two PAF15 molecules and 19-nucleotide DNA. Each PAF15 contributes four positive charges towards the PCNA inner ring. Side-view cartoons and electrostatic surface maps are shown. **c**, Western blot of chromatin fractions from *PAF15*-KO U2OS cells treated with control siRNA (siControl) or *ATAD5* siRNA (si*ATAD5*). **d**, FRAP analysis of PCNA–GFP mobility in U2OS cells depleted of PAF15, ATAD5 or both. Cells received a single bleach pulse followed by real-time recovery imaging. Curves represent mean ± s.d. (n = 10 S-phase cells with similar focal PCNA organization). **e**, Representative PLA images (PCNA–DNA ligase 1) and QIBC quantification of

PLA sum intensity in parental and *PAF15*-KO cells treated with control or *ATAD5* siRNA (n > 8,000 cells per condition). Scale bar, 10 μm. **f**, Schematic of DNA-fibre assay and quantification of nascent DNA-tract lengths in parental and *PAF15*-KO cells treated with control or *ATAD5* siRNA and digested with S1 nuclease (n = 200 fibres). **g**, Schematic of PAF15 function during sequential binding of PCNA to POLδ, FEN1 and DNA ligase 1, protecting lagging-strand factors from ATAD5. **h**, Western blot of PCNA and PAF15 in chromatin and whole-cell lysate (WCL) from parental or *PAF15*-KO U2OS cells with doxycycline (Dox)-inducible *PAF15* expression. **i**, Relative plating efficiency of *PAF15*-overexpressing cells (mean ± s.d.; n = 3). WT, wild type. **j**, Replication fork speed in *PAF15*-KO U2OS cells with induced overexpression of *PAF15* WT or PIP motif mutant (PIP*). n = 200 fibres. P values calculated by one-way ANOVA with Tukey's test. The DNA-fibre and QIBC plots shown are representative of at least two independent biological replicates. The schematic in **g** was created with BioRender.com. Somyajit, K. (2025) https://BioRender.com/yrkrtqc.

---

motif and the N-terminal basic region cooperatively stabilize PCNA on replicating DNA. Although transient increases in chromatin-bound PCNA seemed to alleviate replication catastrophe at early time points (Extended Data Fig. 7i), sustained stabilization of the PAF15–PCNA complex ultimately impaired DNA replication on several levels, culminating in complete cell death (Fig. 4i and Extended Data Figs. 7j and 8a–c). This toxicity was fully rescued by PAF15 variants that lack a functional PIP motif or contain unstable N-terminal truncations (Fig. 4j and Extended Data Fig. 8c). To further assess the consequences of acute PAF15 overexpression, we performed chromatin extraction followed by immunoblotting. Although POLε1 remained stably associated with chromatin, the level of lagging-strand polymerase POLδ1 was markedly diminished, despite enhanced PCNA retention (Extended Data Fig. 8d). In line with this disrupted coordination, PAF15-overexpressing cells showed a pronounced accumulation of PAR chains (Extended Data Fig. 8e), indicative of activated non-canonical OkF maturation and pervasive replication stress arising from defective replisome dynamics. The pronounced replisome slowdown observed under these conditions (Fig. 4j and Extended Data Fig. 8a,b) is likely to reflect impaired leading-strand progression, suggesting that excessive PAF15 perturbs PCNA–POLε interactions without fully displacing POLε itself. Although POLε1 can stably associate with DNA independently of PCNA, previous studies[42,43] have shown that continuous synthesis of the leading strand depends on productive engagement between POLε1 and PCNA.

Thus, maintaining a restricted pool of PAF15 seems to be essential for normal PCNA dynamics at the fork; any acute increase in PAF15 disrupts this balance and triggers a catastrophic replication response.

Structural insights further support this model. The POLε holo complex forms a stable three-point contact with PCNA, occupying all three PIP-box interfaces of POLε and fully saturating the PCNA trimer[20,42,43] (Fig. 5a). This configuration excludes competing PCNA-binding factors such as PAF15. Consistently, structural alignment of PCNA–PAF15–DNA with POLε1–PCNA–DNA reveals pronounced steric clashes (Fig. 5a), providing a potential structural rationale for the replisome slowdown and toxicity induced by rapid PAF15 overexpression. To mechanistically address this further, we investigated whether, under normal conditions, factors at the leading strand protect POLε-engaged PCNA from PAF15 while allowing PAF15 to interact with a more dynamic pool of PCNA on the lagging strand. In our search for these, we found that Timeless and Claspin—RPC components that couple several key replisome interactions, including linking POLε to the CMG helicase[44]—have a crucial role in shielding leading-strand-bound PCNA from PAF15.

First, loss of PAF15 enhanced the chromatin binding of the Timeless–Claspin complex—with Timeless being upstream of Claspin for chromatin loading (Extended Data Fig. 8f). Second, the increased chromatin binding of the Timeless–Claspin complex after PAF15 loss is driven by PCNA, because it was reversed by the PCNA inhibitor T2AA[45], which mirrors the PCNA-binding properties of the PAF15 PIP motif (Extended

Data Fig. 9a,b). Consistent with mouse Claspin, which contains a PIP motif and interacts with PCNA[46], residues 304–341 of human Claspin also contain a PIP motif (Extended Data Fig. 9c). However, the functional importance of this motif in human Claspin remains unknown. We therefore hypothesized that the Timeless–Claspin complex transiently shields PCNA during its loading onto the leading strand.

Supporting this notion, removing Timeless resulted in the mislocalization of PAF15 to POLε1 (Fig. 5b and Extended Data Fig. 9d,e), further exacerbating fork slowdown in PAF15-overexpressing cells (Extended Data Fig. 9f) and leading to pronounced cell-cycle arrest and cell death (Fig. 5c,d and Extended Data Fig. 9g,h). Notably, these lethal phenotypes were reversed when PAF15 was simultaneously removed across various cell types and ablation methods (Fig. 5c,d and Extended Data Fig. 9g,h). Further structure–function studies on strand-specific PCNA loaders and the Timeless–Claspin–PCNA interface will be key to understanding how PCNA is differentially delivered and regulated on the two strands. Moreover, a transient remodelling and displacement of Timeless occurs during redox-sensitive replisome slowdown[13,18,47], and local replication attenuation at double-strand breaks[48] and topological stress[49] are emerging as genome surveillance mechanisms that are likely to reflect a broader adaptive response to replication stress. Such temporary loss of Timeless (and Claspin) might expose PCNA interaction surfaces, enabling PAF15 to access leading-strand PCNA.

Collectively, these findings indicate that a controlled pool of PAF15 is required for normal PCNA chromatin turnover, because excess PAF15 leads to replication collapse. Timeless–Claspin protects the leading strand by restricting PAF15 to its proper sites.

## E2F4 rate-limits PAF15 dosage and genome integrity

PAF15 is an oncoprotein that is overexpressed across nearly all tumour types (Fig. 5e), and high levels of PAF15 correlate with reduced overall survival—more strongly than PCNA overexpression (Fig. 5f, right). Consistently, a comparison of PAF15 levels in normal and cancer-derived cell lines revealed pronounced overexpression in the latter (Fig. 5g and Supplementary Fig. 7a,b). When normalized for genomic content, PAF15 levels were essentially equivalent across the tested cell lines (Fig. 5g, right), underscoring a strict regulatory constraint that prevents the catastrophic consequences of PAF15 surplus. An analysis of single-cell RNA sequencing (RNA-seq) data from breast tissues and bulk RNA-seq profiles from various cell types in the ENCODE database revealed that, although tumour cells overexpress *PAF15*, the ratio of *PCNA* to *PAF15* remains comparably high across different cell types (Extended Data Fig. 10a and Supplementary Fig. 7c,d). This transcript-level relationship mirrors the relative abundance of total and chromatin-bound PCNA and PAF15 (Extended Data Fig. 4f).

Mechanistically, inhibition of global protein degradation using two proteasome-specific inhibitors indicated that, under unperturbed

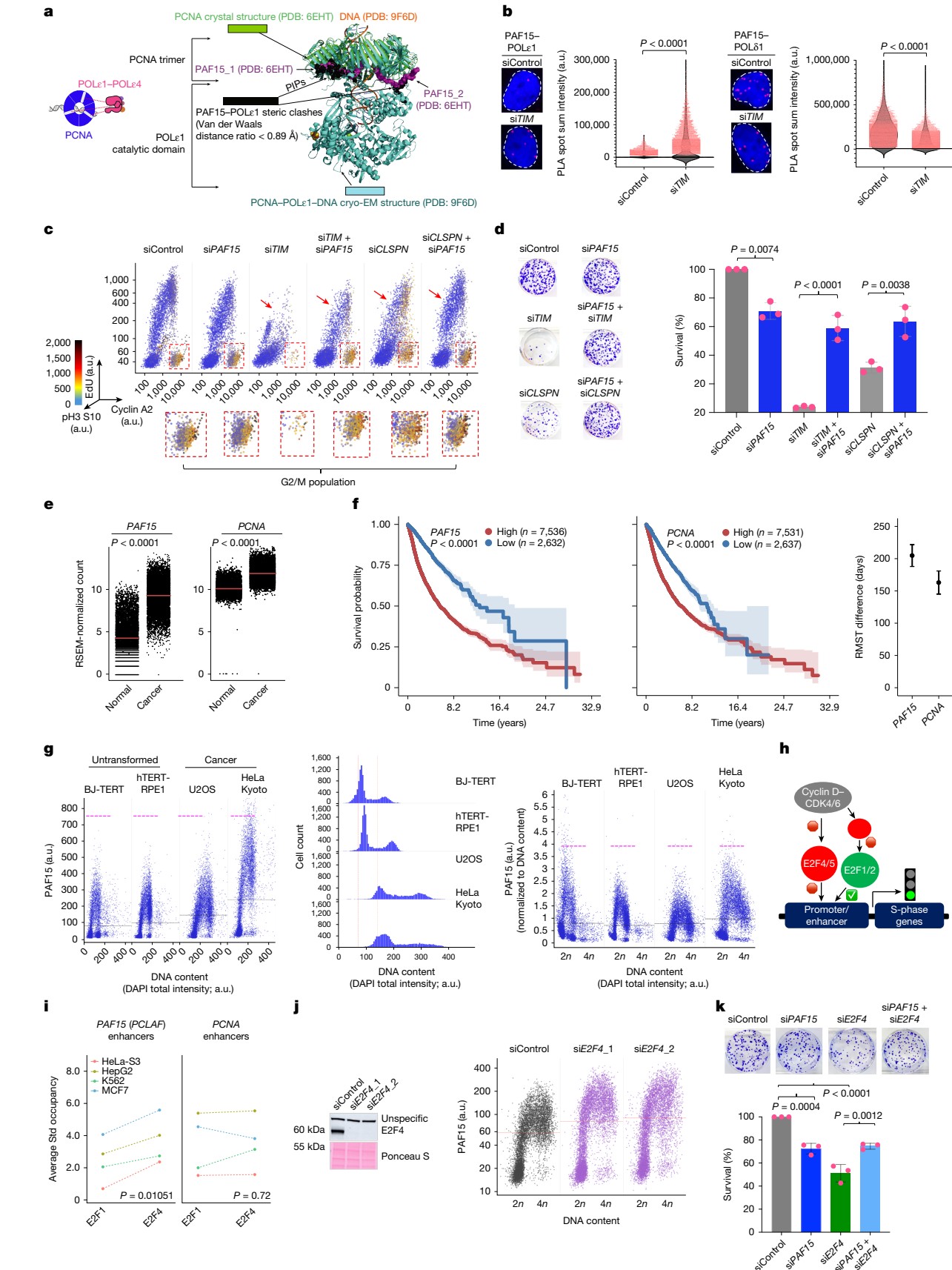

**Fig. 5 | See next page for caption.**

**Fig. 5 | PAF15 access to leading-strand PCNA is lethal and PAF15 excess is restrained by E2F4-mediated dosage control. a**, Left, schematic of the POLε1–POLε4 holo complex engagement with the PCNA ring with three PCNA monomers. Right, alignment of the PCNA–PAF15–DNA structure (PDB: 6EHT; green) with the cryo-EM PCNA–PAF15–DNA–POLε1 structure (PDB: 9F6D; cyan). Steric clashes are shown as black spheres. **b**, Representative images and quantification of S-phase PLA focus intensity from QIBC analysis of PAF15–POLε1 and PAF15–POLδ1 in U2OS cells depleted of *TIMELESS* (also known as *TIM*). si*TIM*, *TIMELESS* siRNA. *n* > 2,500 cells. **c**, QIBC analysis of EdU, cyclin A2 and mitotic phosphorylated Ser10 on histone 3 (pH3 S10) fractions in U2OS cells treated with the indicated siRNAs. *n* > 10,000 cells analysed per condition. Red arrows, S-phase population. **d**, Relative plating efficiency of U2OS cells treated with the indicated siRNAs, shown as mean ± s.d. (right). **e**, Expression of *PAF15* and *PCNA* in normal (*n* = 8,167) and tumour (*n* = 9,471) tissue from 33 cancer types in the Pan-Cancer Atlas. Mann–Whitney *U*-test. **f**, Kaplan–Meier time-to-event analysis with log-rank *P* value for Pan-Cancer Atlas mortality.

Right, difference in restricted mean survival time (RMST) between high and low *PAF15* and *PCNA*. **g**, QIBC of PAF15 levels in the indicated cell lines before (left) and after (right) normalization to DNA content (middle). *n* > 8,000 cells. Black horizontal lines, median values; dotted pink horizontal lines, arbitrarily defined maximum values. **h**, Schematic of transcriptional regulation by E2F activator (E2F1/2) and repressor (E2F4/5). CycD, cyclin D; Rb, retinoblastoma protein. **i**, Average occupancy of E2F4 or E2F1 at *PCNA* and *PAF15* (also known as *PCLAF* and *KIAA0101*) putative enhancers, by ChIP–seq. **j**, Left, immunoblot validation of E2F4 depletion with two *E2F4* siRNAs in U2OS cells. Right, QIBC of PAF15 levels in U2OS with *E2F4* depletion with two siRNAs. Horizontal lines, average values. *n* > 7,000 cells per condition. **k**, Plating efficiency of U2OS cells treated with the indicated siRNAs (mean ± s.d., one-way ANOVA with Tukey's test). Each QIBC plot shown is representative of at least two independent biological replicates. The model in **a** was created with BioRender.com. Somyajit, K. (2026) https://BioRender.com/3gfy32b.

conditions, PAF15 degradation is not responsible for enforcing PAF15 dosage limitation during S phase (Extended Data Fig. 10b). Therefore, we examined whether the transcriptional program that controls PAF15 expression could be responsible. Inhibiting transcription and blocking canonical E2F transcription factors by inhibiting CDK4/6, which prevents Rb phosphorylation and maintains Rb in an active, E2F-repressive state[50], resulted in markedly reduced levels of PAF15 (Supplementary Fig. 7e). Thus, we tested whether context-specific binding of E2F to regulatory elements—through activators (E2F1–E2F3) and repressors or co-repressors (E2F4–E2F6)[51,52]—regulates the dosage of PAF15 to maintain a low PAF15–PCNA ratio, as seen at the transcript number level (Fig. 5h). To dissect this, using the activity-by-contact (ABC) model, which integrates enhancer activity (through chromatin accessibility) with the frequency of enhancer–promoter contact, we identified the causal enhancers for PAF15 and PCNA (Extended Data Fig. 10c). Next, we used ENCODE-based chromatin immunoprecipitation followed by sequencing (ChIP–seq) datasets to assess the occupancy of E2F1 versus E2F4, which revealed a significant enrichment of repressive E2F4 relative to E2F1 on the *PAF15* enhancer, whereas *PCNA* enhancers exhibited balanced binding (Fig. 5i). Depletion of E2F4 led to a marked increase in PAF15 expression without affecting PCNA, as confirmed by RNA-seq and QIBC of PAF15 levels alone and by examining PAF15 and E2F activities in tandem through multiplexed immunostaining of PAF15 in stably integrated E2F reporter cells[50] (Fig. 5j and Extended Data Fig. 10d,e).

Finally, we found that the increase in PAF15 protein levels after E2F4 loss was accompanied by enhanced chromatin loading of PAF15 and PCNA (Extended Data Fig. 10f), and rescued replication catastrophe at the early time point (Extended Data Fig. 10g). Nevertheless, long-term loss of E2F4 ultimately caused cell-cycle arrest, reduced viability and genome instability—effects that were significantly reversed by subsequent PAF15 loss (Fig. 5k and Extended Data Fig. 11a–e).

Thus, by modulating PAF15 dosage, E2F4 induces replication defects that resemble those caused by PAF15 mislocalization after the loss of Timeless or Claspin, although additional factors are likely to be involved, consistent with the broad regulatory role of E2F4 in replication and cell-cycle gene expression.

## Discussion

Collectively, our findings reveal a dosage-limited, strand-specific mechanism that safeguards genome replication during an unperturbed cell cycle. We identify PAF15 as a low-abundance factor that binds to PCNA exclusively on the lagging strand, and is excluded from the leading strand by the Timeless–Claspin complex. Unlike typical PIP-box proteins, PAF15 traverses the PCNA–DNA channel[53] to stabilize PCNA on chromatin and prevent premature unloading by the ATAD5–RFC complex, thereby maintaining lagging-strand processing factors

at active forks. Loss of PAF15 destabilizes chromatin-bound PCNA, impairs OkF maturation and induces replication stress, forcing cells to rely on PARP1-dependent backup pathways (Extended Data Fig. 11f). Because PAF15–PCNA assemblies saturate under normal origin firing, excess initiation after checkpoint loss depletes lagging-strand capacity. The convergence of PAF15–PCNA exhaustion and ATR-enforced origin-firing restriction reveals another fundamental role of the replication checkpoint. Without this safeguard, replication would proceed with uncoordinated leading- and lagging-strand synthesis, ultimately triggering genome instability and catastrophic replication failure.

The late evolutionary appearance of PAF15, absent in yeast, is likely to compensate for the weaker DNA-binding affinity of metazoan PCNA through its unique binding[8]. Notably, the emergence of PAF15 seems to coincide with its integration into the PCNA–DNMT1 axis, which safeguards DNA methylation fidelity and epigenomic integrity[28]. Our data suggest that PAF15 acts first at the lagging strand to coordinate OkF maturation and later promotes the recruitment of DNMT1, thereby linking DNA replication with faithful epigenome inheritance.

Since the foundational 'replicon theory' defined a replicon as requiring both an initiator and a replicator[54], uncovering rate-limiting steps in genome replication has remained central to understanding genome maintenance. Several such mechanisms have since been identified, all converging on the S-phase checkpoint[14,25,55,56]. Our finding that PCNA lagging-strand capacity becomes exhausted concurrently with stochastic origin activation adds an unexpected twist to the well-established fact that PCNA—the driving force behind lagging-strand processing—is one of the most abundant replisome factors.

The rate-limiting regulation of lagging-strand-specific PCNA by PAF15 highlights several key principles of replisome adaptation in physiology and disease. First, limiting the pool of PAF15 enables cells to fine-tune PCNA dynamics specifically on the lagging strand, ensuring fork progression even when PAF15 is fully used—consistent with the ability of cells with lagging-strand defects to proliferate even under heightened replication stress[57]. Second, the looping nature of the lagging strand exposes checkpoint-activating substrates such as primed ssDNA structure[58]; under natural exhaustion or depletion of PAF15 or hyperactivity of ATAD5, these loops might expand into post-replicative gaps, activating S-phase surveillance. Third, transcriptional control of PAF15 by E2F4 and E2F1 enables cells to adjust their replication output according to proliferative demands, a feature that could be exploited by cancer cells to sustain genome expansion. Conversely, excessive or mislocalized PAF15 disrupts replisome coordination and is lethal (Extended Data Fig. 11f). Following this reasoning, we propose that targeted modulation of PAF15–PCNA interactions—for instance, by using PAF15-derived peptides that bind to PCNA–DNA—could selectively disrupt aberrant replication dynamics in cancers with a high origin density and excessive chromatin-bound PCNA.

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

# Methods

## Cell culture

The human U2OS osteosarcoma cell line (ATCC, HTB-96), HeLa Kyoto cervical carcinoma cell line (CVCL_1922), primary immortalized retinal epithelial cell line hTERT-RPE1 (ATCC, CRL-4000), primary immortalized foreskin fibroblast BJ cells (ATCC, CRL-2522) and their derivatives were grown in Dulbecco's modified Eagle's medium (DMEM, high glucose, Glutamax) containing 10% fetal bovine serum (FBS) and penicillin–streptomycin antibiotics (Thermo Fisher Scientific). The E14-derived mouse ES cell lines were obtained from P.-A. Defossez. The J1 mouse ES cell line and PAF15KRKR mutants were obtained from S. Bultmann and H. Leonhardt. Mouse ES cells were cultivated in ESC medium (DMEM (Gibco), 15% KnockOut Serum Replacement (Gibco), 1 mM sodium pyruvate (Gibco), 1× non-essential amino acids (Gibco) and 100 U ml$^{-1}$ penicillin–streptomycin, 2 mM L-glutamine (Life Technologies), supplemented with LIF (mouse leukaemia inhibitory factor, 10 ng ml$^{-1}$; Miltenyi) and 0.1 mM β-mercaptoethanol (Sigma)). All cell lines and their derivatives were cultured under standard cell culture conditions (37 °C with 5% $CO_2$, humidified atmosphere). Cells were routinely tested for mycoplasma contamination (MycoAlert, Lonza) and were always negative.

## Chemical reagents

Reagents were as follows: hydroxyurea (ribonucleotide reductase (RNR) inhibitor; Sigma-Aldrich, H8627; solubilized in $H_2O$), cerala-sertib (AZD6738) (ATR inhibitor; Selleckchem, S7693), adavosertib (MK-1775) (WEE1 inhibitor; Selleckchem, s1525), palbociclib (PD-0332991) (CDK4/6 inhibitor; Selleckchem, S1116), MG132 (proteasome inhibitor; Selleckchem, s2619), bortezomib (proteasome inhibitor; Selleckchem, S1013), PDD 00017273 (PARG inhibitor; Tocris, 5952), doxycycline (Thermo Fisher Scientific, BP2653-5), T2AA (PCNA inhibitor; Tocris, 4723), nocodozole (Tocris, 1228), 5-chloro-2′-deoxyuridine (CldU; Sigma-Aldrich, c6891), 5-iodo-2′-deoxyuridine (IdU; Sigma-Aldrich, I7125), olaparib (PARP inhibitor; Selleckchem S1060), decitabine (DNMT1i; Tocris, 2624), PHA-767491 hydrochloride (CDC7 inhibitor; Tocris, 3140), 5-Ph-IAA (Merck, SML3574) and 5-ethynyl-2′-deoxyuridine (EdU; Thermo Fisher Scientific, A10044). The drugs were reconstituted in DMSO and were used as indicated in the figure legends.

## Generation of knockout and complementation cell lines

Knockout of the *PAF15* gene in U2OS, HeLa Kyoto and hTERT-RPE1 cells was done using a single guide RNA (gRNA) (targeting exon 2, GGCTGCTCGAGCCCCCAGAA) cloned into pSpCas9(BB)-2A-Puro (PX459) V2.0 (Addgene plasmid 62988, a gift from F. Zhang) via the BbsI restriction site, followed by transfection with Lipofectamine LTX Plus. After two days, transfected cells were selected in medium containing 1–8 µg ml$^{-1}$ puromycin (InvivoGen, ant-pr-1). After 30–40 h, transfected cells were recovered in plain medium and serially diluted into single cells per well of a 96-well plate to obtain single colonies, expanded and tested for knockout efficiency by immunofluorescence of PAF15 using high-content imaging (QIBC) screening, western blotting and Sanger sequencing of gRNA targeting sites. Only cell lines that passed all validation steps were used. Clones with successful knockout of *PAF15* were selected for phenotypic validation. A similar approach was used, by transfecting *p53* (also known as *TP53*) single gRNA (PX459-TP53-exon4, Addgene plasmid 217455, a gift from J. Diffley) or in combination with PX459-PAF15-exon2 (this paper) to generate *p53* knockout or *p53* + *PAF15* double knockout in hTERT-RPE1 cells.

## Constitutive and T-REx-inducible cell lines

For complementation assays with constitutive expression, *PAF15*-KO U2OS cell lines were transfected with the following variant plasmids: PAF15wt-1×MYC-1×-Flag-tag (Origene, RC200694), PAF15 PIP-box mutant (F68F69 to AA mutation)-1×Flag or PAF15 KEN-box mutant (K78A mutation)-1×Flag. Appropriate DNA constructs were transfected using Lipofectamine LTX Plus reagent in *PAF15*-KO cells. Transfected cells were serially diluted into single cells per well of a 96-well plate to obtain single colonies under selection with DMEM containing geneticin (Gibco, 10131-027) for 12 days. Individual colonies were expanded and tested by immunofluorescence, using QIBC for C terminus Flag-tag or MYC-tag PAF15 (cellular localization), and expression level was tested by western blotting, using antibodies against PAF15.

All of the inducible cell lines were generated through tetracycline-regulated expression of the gene of interest, using T-REx (Thermo Fisher Scientific, K102002). U2OS *PAF15*-KO cell lines were introduced with doxycycline inducible of each PAF15 wt-1×Flag, PAF15 PIP-box mutant-1×Flag, PAF15 KEN-box mutant-1×Flag, and ΔPAF15 variants (2–11 amino acid deletion) were cloned into the doxycycline-inducible expression vector pcDNA4/TO. PAF15 variant constructs cloned in pcDNA4/TO were co-transfected with the pcDNA6/TR plasmid (expression vector for Tet repressor) by Lipofectamine LTX Plus reagent (Thermo Fisher Scientific, 15338-100). After two days, transfected cells were selected in DMEM supplemented with 10% FBS containing blasticidine and zeocine (Thermo Fisher Scientific; blasticidine, A1113902; zeocine, R25001). After reaching 60–70% confluency, cells were serially diluted into single cells per well of a 96-well plate to obtain single colonies, expanded and tested for PAF15 induction after doxycycline treatment by QIBC, western blotting and high-resolution microscopy.

## *PAF15* degron cell line

A derivative of the U2OS cell line expressing C-terminally endogenously AID–GFP-tagged PAF15 was generated using CRISPR–Cas9 as previously described[59]. In brief, gRNA targeting the C terminus of the *PAF15* locus (guide: TAACGTCTCCTTGTTTACCC) was cloned into pX330-U6-Chimeric_BB-CBh-hSpCas9 (Addgene plasmid 42230, a gift from F. Zhang) via the BbsI restriction site. Cells were co-transfected by Lipofectamine LTX Plus reagent (Thermo Fisher Scientific, 15338-100) with pX330 plasmid containing cloned gRNA, a donor plasmid containing the tag (AID–GFP) with flexible linker flanked by 900-bp homology arms complementary to the C terminus of PAF15 and pCMV6-A-Puro-Tir1-9xMYC plasmid conferring puromycin resistance. After 24 h of transfection, the cells were selected with DMEM containing puromycin (1 µg ml$^{-1}$) for three days and then serially diluted onto 100-mm dishes. The cells were grown in DMEM to obtain single colonies, which were expanded for further characterization by junction PCR spanning the C terminus of *PAF15*. Selected clones were functionally validated by immunofluorescence (subcellular localization) using QIBC. Only cell lines that passed all validation steps were used.

## PCNA chromobody, TurboID–PCNA and E2F reporter cell lines

U2OS naive cells and *PAF15*-KO U2OS cells were transfected with plasmid (RFP-pCellCycleChromobody) containing RFP–PCNA chromobody (ChromoTek, ccr) encoding a single-chain antibody to endogenous PCNA. Single clones were selected by geneticin as described in the previous section. Using the same procedure, pBABE-NLS-HA-TurboID-PCNA (Addgene plasmid 215074, a gift from M. Pagano) and pLV-hCDC6p-Venus-puro (Addgene plasmid 212666, a gift from T. Meyer and Y. Konagaya) were introduced into naive U2OS cells. Single clones were selected using puromycin as described above.

## Gene silencing by siRNA

Transfections of siRNA duplexes were done with Lipofectamine RNAiMAX (Thermo Fisher Scientific) at a final concentration of 1–20 nM for 48 h (see figure legends for more details). For knockdown of *TIMELESS* and *CLSPN*, transfection was performed with 1–5 nM for 24–30 h to prevent adverse cell-cycle effects. The siRNAs were purchased from Thermo Fisher Scientific as Silencer Select reagents targeting the following genes: *PAF15* (A: s18863; B: s18862), *TIMELESS* (s17054), *CLSPN* (s34330), *UHRF1* (A: s26553; B: s26555), *LIG1* (s8173),

*RFC1* (s224528), *DNMT1* (s4215) *ATAD5* (s36632) or *E2F4* (1: 114194; 2: s4415). Non-targeting siRNA from Thermo Fisher Scientific (Ambion negative control 1; 4390844) was used as a control in all experiments.

## DNA-fibre analysis

DNA-fibre spreads were performed as described[18]. In brief, $10^5$ cells were pulse-labelled with 25 μM CldU for the indicated time (see figures for respective labelling protocols), washed three times with DMEM and pulse-labelled with 250 μM IdU with or without the indicated treatment for the indicated time. Labelled cells were collected on ice-cold phosphate-buffered saline (PBS) and mixed with unlabelled cells (1:3). Subsequently, 2 μl of the cell suspension was placed on SuperFrost slides (AB00008032E01MNZ20) and mixed with 8 μl of lysis buffer (0.5% SDS, 200 mM Tris pH 7.5 and 50 mM EDTA) followed by vigorous pipetting for in situ lysis. After 2 min of incubation, slides were tilted to allow the lysate to flow along the slide slowly to the end of the slide. Next, slides were fixed in methanol:acetic acid (3:1) for 12–15 min, washed four times in PBS and transferred to 2.5 M HCl for DNA denaturation for 80 min. Afterwards, slides were neutralized by washing four times in PBS and blocked in blocking buffer (1× PBS, 0.1% Triton X and 1% bovine serum albumin (BSA)) for 5 min. CldU was stained by incubating slides with rat anti-BrdU antibody (Abcam ab6326, 1:100 in blocking buffer) for 90 min. Afterwards, slides were washed once with PBS containing 0.1% Tween, followed by three wash steps with PBS, fixed with 4% formaldehyde for 12 min and incubated with AlexaFluor 594-conjugated goat anti-rat IgG (1:100; Thermo Fisher Scientific, A-11077) for 60 min. Slides were washed four times with PBS, and IdU was stained using mouse anti-BrdU antibody (1:100; Becton Dickinson, 347580) overnight at 4 °C, followed by AlexaFluor 488-conjugated goat anti-mouse IgG (1:100; Thermo Fisher Scientific; A11029) for 90 min. Fibres were acquired using an Olympus BX53 upright fluorescence microscope with a 40× air objective. For quantifying replication structures, at least 200 DNA fibres were counted per experiment. The lengths of red (CldU) or green (IdU) labelled patches were measured using Fiji ImageJ (National Institutes of Health). Fork speed (in kb min$^{-1}$) was calculated by multiplying the measured length in μm with a conversion factor of 2.59 kb μm$^{-1}$ and dividing by the duration of the labelling pulse.

DNA-fibre experiments involving the ssDNA-specific endonuclease S1 were performed as described previously[39]. In brief, after sequential pulse-labelling with CldU and IdU, cells were collected in ice-cold PBS. Labelled cells were then mixed 1:1 with unlabelled cells and resuspended in hypotonic buffer (10 mM HEPES, pH 7.5, 10 mM NaCl, 0.3 M sucrose and 0.5% Triton X-100) for 15 min on ice, followed by centrifugation at 1,500*g* for 5 min. Cell pellets were incubated in 50 μl S1 nuclease buffer (30 mM sodium acetate, pH 4.6, 10 mM zinc acetate, 5% glycerol and 50 mM NaCl) with or without 10 U ml$^{-1}$ S1 nuclease (Invitrogen, 18001-016) for 10 min at 37 °C. After digestion, the cell suspension was resuspended in 50 μl lysis buffer (0.5% SDS, 200 mM Tris, pH 7.5 and 50 mM EDTA) and processed according to the standard DNA-fibre spreading protocol.

To assess the distance between adjacent active replication forks, DNA fibres were prepared as described previously[59,60]. In brief, CldU-labelled cells were mixed at a 1:10 ratio with unlabelled cells before DNA-fibre preparation. ssDNA was visualized using rabbit anti-ssDNA antibodies (Tecan, IBL International, 18731, 1:500). The distance between active forks was determined by measuring the centre-to-centre spacing between two neighbouring CldU-labelled tracks (referred to as the 'distance between adjacent active forks'). Local fork density was calculated by counting the number of CldU-labelled tracks within ssDNA-positive DNA fibres and normalizing this value to the length of DNA analysed (Mb).

## Immunofluorescence

Cells were grown on round, 12-mm diameter, 1.5-mm-thick glass coverslips (cleaned in 96% ethanol, dried and autoclaved). Unless stated chromatin-bound, cells were washed with ice-cold PBS and fixed in 4% buffered formaldehyde for 12 min at room temperature before permeabilization with PBS containing 0.2% Triton X-100 for 5 min. For assessing chromatin-bound proteins, cells were first pre-extracted with ice-cold PBS containing 0.2% Triton X-100 for 2 min on ice before fixation in 4% buffered formaldehyde for 10 min at room temperature. When Click-iT EdU staining was performed, cells were incubated for 30 min in 10 μM EdU before fixation or pre-extraction. EdU staining was performed according to the manufacturer's instructions (Thermo Fisher Scientific) before incubating with primary antibodies. All antibodies were diluted in DMEM (high glucose, Glutamax) containing 10% FBS. Primary antibody incubations were performed at room temperature for one hour. Coverslips were washed three times with PBS containing 0.2% Tween (Sigma-Aldrich). Secondary-antibody incubations were performed at room temperature for 30 min and were supplemented with 0.5 mg ml$^{-1}$ DAPI (Sigma-Aldrich, D8417) to counterstain DNA.

For accessing unligated OkFs, the relevant cells (see also figure legends) were incubated with 10 μM CldU for 48 h to label the parental template DNA. After washing, the cells were exposed to 10 μM EdU for the last 60 min to label nascent DNA—both within the replisome and in post-replication regions—to mark S-phase cells. The cells were then fixed and permeabilized as detailed above and incubated with an alkaline buffer (1×: 50 mM NaOH and 1 mM EDTA in PBS) for 30 min before proceeding with EdU Click-iT staining, immunostaining of CldU (using a rat anti-BrdU antibody, Abcam ab6326, 1:500) and counterstaining of DNA with DAPI as described above.

After three washes in PBS, coverslips were washed twice in distilled water, dried on 3-mm paper and mounted in 4.5 ml Mowiol-based mounting medium (containing Mowiol 488 (Calbiochem), glycerol and Tris-HCl, pH 8.5). For all of the confocal and STED imaging, slides were mounted with Prolong Diamond Antifade Mountant (Thermo Fisher Scientific, P36961).

## QIBC

QIBC was performed as previously described[2,18,39,59,61]. In brief, images were acquired with a ScanR inverted microscope high-content screening station (Olympus) equipped with wide-field optics, a 203, 0.75-NA (UPLSAPO 203) air objective, fast excitation and emission filter-wheel devices for DAPI, FITC, Cy3 and Cy5 wavelengths, an MT20 illumination system and a digital monochrome Hamamatsu ORCA-R2 CCD camera (yielding a spatial resolution of 320 nm per pixel at 203 and binning of 1). Images were acquired in an automated manner with the ScanR acquisition software (Olympus, 3.4). Depending on cell confluency, 100 images were acquired, containing more than 5,000 cells per condition. Acquisition times for the different channels were adjusted for nonsaturated conditions in a 12-bit dynamic range, and identical settings were applied to all the samples in one experiment. Images were processed and analysed with ScanR analysis software. First, a dynamic background correction was applied to all images. The DAPI signal was then used to generate an intensity-threshold-based mask to identify individual nuclei as main objects. This mask was then applied to analyse pixel intensities in different channels for each nucleus. After segmentation of nuclei, foci were segmented as above, and the desired parameters for the different nuclei or foci were quantified, with single parameters (mean and total intensities, foci count and foci intensities) as well as calculated parameters (sum of foci intensity per nucleus). These values were then exported and analysed with TIBCO v.12.4. This software was used to quantify absolute, median and average values in cell populations and to generate all colour-coded scatter plots. Within one experiment, similar cell numbers were used for the different conditions (at least 2,000–10,000 cells), and for visualization, low *x*-axis jittering was applied (random displacement of objects along the *x* axis) to make overlapping markers visible.

## Confocal microscopy

Confocal images were acquired with a Nikon A1 confocal Ti2 microscope integrated with a 100 × 1.45 NA oil, Plan Apochromat λ objective and NIS-Elements AR software (v.5.20.02). A resonant scanner equipped with an A1-DUG hybrid 4-channel detector was used for image acquisition at 512 × 512 pixels. Laser power, detector gain and exposure time were appropriately adjusted with identical settings applied within a series of experiments. Microscope performance and channel alignment were regularly checked by imaging of 200-nm multicolour fluorescent beads.

## STED

For combined STED and confocal microscopy, EdU-labelled pre-extracted samples were imaged using an Abberior Facility Line STED microscope (equipped with a 100× NA 1.4 oil objective (UPLSAPO100X, Olympus). Click-iT EdU coupled to AlexaFluor 488 was imaged for confocal microscopy but PAF15 (using STAR RED goat anti-rabbit (1:250, STRED-1007, Abberior) was imaged for both confocal and STED microscopy. Regions of interest (5× 5 μm$^2$) showing immobilized dye signal were imaged for the indicated number of frames using standard confocal and STED imaging conditions, 10 μs dwell time, 50 nm pixel size and a repetition frequency of 40 MHz. To collect emission spectra, 488-nm and 561-nm laser lines were used with a power of 10 μW as measured at the sample plane. All data were acquired using the iMSPECTOR v.16.3.13787 acquisition software. A mean filter with a radius of one pixel was applied to remove background noise.

## SCAR-seq

*PAF15* SCAR-seq was performed as described[33] with the following modifications. Cells were grown to 80% confluence and EdU was added for 20 min (final concentration 20 μM) to label newly synthesized DNA and cross-linked by adding formaldehyde to a final concentration of 1% for 10 min, then the reaction was quenched with glycine (final concentration 0.1 M) and washed twice with ice-cold PBS. Fixed cells were lysed in ice-cold lysis buffer at room temperature for 20 min. Lysis buffer was prepared by adding one-third dilution buffer (100 mM Tris-HCl pH 8.6, 100 mM NaCl, 5 mM EDTA and 5.0% Triton X-100) and two-thirds SDS buffer (100 mM NaCl, 50 mM Tris-HCl pH 8.1, 5 mM EDTA and 0.5% SDS) supplemented with cOmplete protease inhibitor (Roche, 11873580001). Chromatin extracts were sonicated using Diagenode Bioruptor Pico to generate chromatin fragments averaging 250–300 bp. For each chromatin immunoprecipitation (ChIP), approximately 300 μg of chromatin, based on DNA concentration, and 4 μg of anti-PAF15 antibody (Santa Cruz sc390515; PCLAF) were used. The PAF15–DNA complexes were pulled down using anti-mouse IgG Dynabeads (Thermo Fisher Scientific, 11201D), and the DNA chromatin complexes were de-cross-linked at 55 °C for four hours. One per cent of the lysate was reserved as an input control.

Click biotinylation of newly replicated DNA, new-strand isolation and libraries were prepared with xGEN UDI-UMI adapters (IDT). Libraries were sequenced in paired-end mode on an Illumina NextSeq 2000.

## SCAR-seq data processing

Raw reads were trimmed. Reads with poor quality (lower than 20) were filtered using cutadapt (v.2.6) and aligned to the mouse reference genome (GRCm39) using bowtie2 (v.2.4), and duplicated reads were marked and removed by Picard tools (v.3.4).

The resulting processed bam files were split into forward and reverse strands according to SAM flags (SAMtools, v.1.13). Forward strands were defined with both -f 83 and -f 163 SAM flag, and reverse strands with -f 99 and -f 147. SAM flags were inverted according to the inverted orientation of IDT UDI-UMI adapters. Read coverage was computed using multiBamSummary (deepTools, v.3.5.4) in bins of 1 kb. Bins with normalized read counts per million (CPM) < 0.3 were excluded from the following analyses.

PAF15 strand-specific partitioning was computed from mapped binarized files according to the following: Partitioning = $(F - R)/(R + F)$, where $F$ and $R$ correspond to the number of mapped reads to the forward strand and to the reverse strand, respectively.

## OK-seq

OK-seq was performed as described previously[34], with modified adapter sequences as follows:

Adapter mix A:

Adapter1w-mixA: [SpC3] ACACTCTTTCCCTACACGACGCTCTTC CGATCT

Adapter1c-mixA: [SpC3] NNNNNNAGATCGGAAGAGCGTCGTGTAGG GAAAGAGTGT

Adapter2w-mixA: [Phos]AGATCGGAAGAGCACACGTCTGAACTC CAGTCA [SpC3]

Adapter2c-mixA: [SpC3] TGACTGGAGTTCAGACGTGTGCTCTTC CGATCTNNNNNN [SpC3]

Libraries were sequenced in paired-end mode on an Illumina NextSeq 2000.

## OK-seq data processing

Raw reads were trimmed. Reads with poor quality (lower than 20) were filtered using cutadapt (v.2.6) and aligned to the mouse reference genome (mm39) using bowtie2 (v. 2.4), and duplicated reads were marked and removed by Picard tools (v.3.4). The resulting processed bam files were split into forward and reverse strand according to SAM flags (SAMtools, v.1.13). Forward strands were defined using both -f 99 and -f 147 SAM flags, and reverse strands with -f 83 and -f 163. Read coverage was computed using multiBamSummary (deepTools, v.3.5.4) in bins of 1 kb. Bins with normalized CPM < 0.3 were excluded from the following analyses. The RFD score was computed from a binarized file according to the following: RFD = $(R - F)/(R + F)$, where $R$ and $F$ correspond to the number of forward and reverse mapped reads, respectively. Initiation zone positions were computed with only the regions with RFD$_{max}$ > 0 and RFD$_{min}$ < 0 were considered. Initiation zones with a minimum size of 10 kb, an efficiency higher than 10% ($\Delta$RFD > 0,2; $\Delta$RFD is the difference between RFD$_{max}$ and RFD$_{min}$) and a minimum overlap of 50% between 2 biological replicates were retained, resulting in 4,559 regions.

## SCAR-seq and OK-seq data analysis

Average profiles of *PAF15* partition and RFD values around initiation zones were computed up to 100 kb upstream and downstream of each initiation zone by averaging values within each bin position and smoothed by considering neighbouring windows of 15 kb. The difference in *PAF15* partitioning at the left and right ends of the initiation zone was tested with a paired Wilcoxon signed-rank test in three independent biological replicates.

Genome-wide Spearman's rank correlation was calculated in 1-kb bins and represented as a hexplot. For IGV visualization, *PAF15* strand partitioning values and OK-seq RFD values were smoothed considering the neighbouring 15 bins.

## FRAP

U2OS cells expressing GFP-tagged-PCNA were seeded in imaging dishes (Nunc, Lab-Tek, 155361) and before imaging, transferred to $CO_2$-independent medium. FRAP data were acquired using a Nikon A1 Ti2 microscope with a 60× 1.2 NA Plan Apo Water Immersion objective and NIS-Elements (v.5.30.02) software, under stable temperature conditions of 37 °C. After ten pre-bleaching frames (pre), a single bleach pulse (488-nm argon laser set to 100% power) was delivered in a defined region, followed by time-lapse imaging for 3 min at maximum scanning speed (six frames per second) with the laser transmission attenuated to 2.5%. Image analysis was performed by first extracting the mean GFP-associated fluorescence intensity for each time point in

the following regions: bleaching region (Ifrap(t)); background outside the nucleus (Iback(t)); and signal within the nucleus in which bleaching was performed (Iref(t)). After background correction, double normalization was applied, which corrects for differences in the starting intensity in the Ifrap region and for loss in total nuclear fluorescence in the Iref region owing to the bleaching pulse and to acquisition bleaching.

## Cell synchronization

U2OS cells were synchronized at the G2/M phase by the addition of nocodazole. Exponentially growing U2OS cells were incubated with 200 ng ml⁻¹ nocodazole for 20 h. For enrichment of cells into different cell-cycle phases, cells were washed and cultured in fresh DMEM. The cells were collected at 0, 6, 8, 10, 12, 15 and 24 h and fixed with 4% buffered formaldehyde for 10 min at room temperature and permeabilized with PBS containing 0.2% Triton X-100 for 5 min. QIBC-based *PAF15* total pool and cell-cycle analysis was performed with Click-iT EdU staining, assessing sequential cell-cycle phase transition. Complementary to this, a western blot was performed to assess the PAF15 in soluble and chromatin-bound subcellular fractions.

## RNA isolation for RNA-seq and quantitative PCR

Total RNA was isolated using the RNeasy Mini Kit (Qiagen), following the manufacturer's protocol. RNA concentration was measured using a NanoDrop spectrophotometer. cDNA was prepared, according to the manufacturer's instructions, using the AMPIGENE cDNA Synthesis Kit (Enzo Life Sciences). The quantitative PCR (qPCR) reactions were performed in triplicate using iTaq Universal SYBR Green Supermix (Bio-Rad). Relative expression levels were calculated using the $2^{-\Delta\Delta CT}$ method.

The following primer pairs were used:

*ATAD5* forward (5′-GCCAACCCTTCGAAACATCTG-3′) and reverse (5′-CTTCAAAATAGTGCAGGAATCTTCT-3′), *GAPDH* forward (5′-CACC ATCTTCCAGGAGCGAG-3′) and reverse (5′-TGATGACCCTTTTGG CTCCC-3′), *PCNA* forward (5′-GCAGATGTACCCCTTGTTGT-3′) and reverse (5′-ATCCTCGATCTTGGGAGCCA-3′) and *PAF15* forward (5′-GGCGGGATAGTTTTCGGGTC-3′) and reverse (5′-CGAGCAGCCACC ACTTTTCT-3′).

For DMSO and CDK4/6 inhibitor, three biological repeats were performed in hTERT-RPE1 cells. RNA-seq was performed at BGI Genomics using DNBseq stranded mRNA libraries generated on the DNBseq NGS platform after quality control. The quality of the raw sequencing data was assessed using FastQC (v.0.11.9) and MultiQC (v.1.10.1). The raw bulk RNA-seq data were aligned to the human genome assembly (GCF_000001405.39_GRCh38.p13), and differential gene expression was analysed using the DEF analysis plan and a Poisson distribution model at BGI Genomics.

For control and E2F4 depletion, three biological repeats were performed in hTERT-RPE1 cells. RNA-seq was performed according to the manufacturer's instructions (TruSeq2, Illumina) using 500 ng of RNA for the preparation of cDNA libraries. Sequencing reads were mapped to the human genome (hg38) using STAR, and tag counts were summarized at the gene level using HOMER56, allowing only one read per position per length. TiCoNE25 was used to cluster differentially expressed genes as determined by DESeq2.

**Clonogenic survival assay.** Naive cells, gene-knockout cells and cells stably expressing PAF15 variant constructs were transfected with control and other siRNAs for 48 h; these cells were seeded in six-well plates in triplicate (200, 500 and 1,000 cells per well). After 24 h, genotoxic treatments were performed as indicated in the figure legends. Cells were incubated for ten days, fixed with 4% formaldehyde and stained with crystal violet. Individual colonies were counted manually, and the percentage survival was calculated as the value for an indicated siRNA divided by the value for the control siRNA, after correcting for the respective plating efficiency.

**PLA.** PLA was performed as described previously[18] with modifications. In brief, cells were fixed either with methanol (for PCNA interactions) for 15 min or 4% buffered formaldehyde (PAF15 interactions) for 12 min and permeabilized with 0.2% Triton X-100 in PBS for 5 min. Cells were then blocked for one hour in DMEM containing 10% FBS and incubated with primary antibody in a humidity chamber for one hour and secondary-antibody probes in a humidity chamber at 37 °C for one hour. In situ proximity polymerization followed by ligation was performed using a Duolink Detection Kit (Sigma-Aldrich), and the nucleus was counterstained with DAPI. Nuclear foci were imaged using a ScanR inverted microscope and processed for QIBC. At least 5,000 cells per condition were analysed in each experiment.

## Chromatin fractionation

A total of $3 \times 10^6$ cells were collected from experimental conditions as described in the figure legends. The cells were washed with PBS and collected. The soluble protein fraction was removed by incubation in 0.5% Triton X-100 in PBS supplemented with 1× protease and phosphatase inhibitors (PPi) cocktail (Roche). The fractions were centrifuged for 5 min at 4 °C at 16,000g. The samples were washed with PBS containing 0.5× PPi. Finally, the cell pellets were lysed in RIPA buffer (50 mM Tris-HCl, pH 7.5, 150 mM NaCl, 0.1% SDS, 1% Triton X-100 and 0.5% deoxycholate) (Sigma-Aldrich, R0278-500ML), with benzonase (Merck, E1014) and 100 μg ml⁻¹ RNaseA (Thermo Fisher Scientific, EN0531). The samples were sonicated at low amplitude, on ice for two repeats of 20-s pulses after incubation for one hour on ice. Chromatin-bound protein pools were collected by centrifugation at 4 °C, 16,000g for 30 min. Protein concentration was quantified using the Pierce BCA Protein Assay kit (Thermo Fisher Scientific, 23227), and 20–50 μg protein from chromatin fraction was used for western blots.

## Western blotting

Whole-cell extracts (WCE) were obtained by lysis in RIPA buffer (50 mM Tris-HCl, pH 8.0, 150 mM NaCl, 1.0 % IGEPAL CA-630, 0.1% SDS and 0.1% sodium-deoxycholic acid), supplemented with protease and phosphatase inhibitors (Roche) containing benzonase (Novagen). Protein extracts from WCE or chromatin fractions were separated by SDS–PAGE after boiling samples in reducing buffer (DTT and β-mercaptoethanol) as per standard procedures. Separated proteins were transferred from the gel to a nitrocellulose membrane. The membrane was blocked for one hour in TBS 0.1% Tween containing 5% powdered milk (TBS-T) and subsequently incubated with primary antibodies for two hours at room temperature or overnight at 4 °C. Phospho-specific primary antibodies were diluted in 3% BSA in TBS-T solution. Secondary peroxidase-coupled antibodies (Vector Labs) were incubated at room temperature for 1 h. Enhanced chemiluminescence (ECL)-based chemiluminescence was detected with an Amersham Imager 680 system (software v.2.0).

## Immunoprecipitation

A total of $2 \times 10^7$ cells from naive U2OS, PAF15 constitutive overexpression and doxycycline-induced PAF15 cells with the indicated siRNAs (see figure legends) were collected for WCE, or the chromatin-bound proteins were either processed without cross-linking or cross-linked by incubating cells in 0.1% formaldehyde for 15 min at room temperature. The reaction was quenched by incubating with 0.125 M glycine. Cells were then collected by scraping and washed with PBS and incubated on ice for 15 min with 0.5% Triton X-100 in PBS supplemented with PPi cocktail (Roche). The samples were washed with PBS containing 0.5× PPi, and the nuclear pellets were resuspended in RIPA lysis buffer (50 mM Tris-HCl (pH 7.5), 150 mM NaCl, 0.1% SDS, 1% Triton X-100 and 0.5% deoxycholate), with benzonase (Merck, E1014) and 100 μg ml⁻¹ RNaseA (Thermo Fisher Scientific, EN0531). After incubation for one hour on ice, samples were sonicated at low amplitude on ice for two repeats of 20-s pulses. After centrifugation at 16,000g for 30 min, the supernatant was collected, and proteins were quantified using the

Pierce BCA Protein Assay kit. Then, 1,000 µg of the chromatin or WCE was incubated overnight at 4 °C with anti-MYC magnetic beads (Sigma) or anti-Flag magnetic agarose (Thermo Fisher Scientific). Similar reactions were also performed for naive U2OS cells with an equivalent amount of beads, and 5–10% of the chromatin was used as an input control. Bound proteins were eluted from beads by boiling in NuPAGE LDS Sample Buffer (Thermo Fisher Scientific, NP0007) with NuPAGE Sample Reducing Agent, or eluted using Flag peptide (Thermo Fisher Scientific). Total immunoprecipitants were then analysed by immunoblotting or processed for MS analysis as specified in the figure legends.

### Chromatin-bound PAF15 immunoprecipitation and MS
Chromatin-bound PAF15 immunoprecipitation protein products were resolved on a NuPAGE Novex Bis-Tris 4–12% gel (Invitrogen). Lanes for each sample were sliced (around 1 mm³) and gel slices were destained further with a buffer containing 50 mM ammonium bicarbonate and 50% acetonitrile. Gel pieces were dehydrated by the addition of 100% acetonitrile. The gel pieces were incubated with 10 mM DTT for 30–45 min and further incubated for 30 min with 55 mM IAA and subsequently dehydrated with 100% acetonitrile. The proteins were digested with trypsin (Sigma) at 37 °C for 16 h, after which the samples were acidified with 0.5% trifluoroacetic acid (TFA). The resulting peptides were desalted on reversed-phase C18 StageTips columns, eluted with 40 ml of 50% acetonitrile, 0.1% TFA followed by 10 ml of 70% acetonitrile, 0.1% TFA and subsequently dried by vacuum centrifugation. Samples were redissolved in 0.1% formic acid of which 5 of 12 µl were loaded onto an Easy nLC (Thermo Fisher Scientific) equipped with a custom-made two-column set-up (pre-column: 100 µm ID, 3.5 cm, Reposil-Pur 120 C18-AQ, 5 µm (Dr. Maisch); analytical column: 75 µm, 18 cm, Reposil-Pure 120 C18-AQ, 3 µm (Dr. Maisch)). Peptides were eluted with a gradient of solvent B (95% acetonitrile, 0.1% FA) as specified below, with solvent A being 0.1% formic acid, and sprayed directly into an Exploris 480 (Thermo Fisher Scientific) mass spectrometer. The gradient was constructed as follows: 5% to 25% B in 70 min, 25% to 40% B in 19 min and 40% to 95% B in 1 min.

On the Exploris 480, MS1 data were obtained at resolution 120 K, scan range 350–1,600, AGC target 2.5e10 and with the Max IT set to auto. MS2 data were recorded as top 12 at a resolution of 30 K, AGC target 2e5, Max IT 100 ms and 20-s dynamic exclusion. Data were searched against the SwissProt database of human proteins using Proteome Discoverer 2.5 (Thermo Fisher Scientific) with Mascot 2.7 as the search engine. The $m/z$ tolerance was set to 5 ppm for precursors and 0.05 Da for fragment ions. Quantification was done as label-free quantification based on the area under the curve of the (up to) top five most abundant peptides for each protein, also using match between runs.

### Proximity labelling of TurboID–PCNA and MS
Stably expressing PCNA–TurboID U2OS cells treated with control and *PAF15* siRNA were grown to 80% confluency, and proximity labelling was done with 50 µM biotin for 30 min. Collected cells were lysed in RIPA buffer (50 mM Tris, 150 mM NaCl, 0.1% (w/v) SDS, 0.5% (w/v) sodium deoxycholate, 1% (v/v) Triton X-10, pH 7.5), sonicated, cleared by centrifugation and incubated with streptavidin magnetic beads (Pierce) in RIPA buffer for enrichment of biotinylated proteins. The beads were sequentially washed with RIPA buffer, Tris buffer (50 mM Tris, 150 mM NaCl, pH 7.5), 1 M KCl, 0.1 M $Na_2CO_3$ and 2 M urea in 10 mM Tris pH 7.5, using a KingFisher system (Thermo Fisher Scientific).

After enrichment of proximity labelled proteins or chromatin, proteins were precipitated on streptavidin magnetic beads (Pierce) or HILIC microparticles (Resyn BioSciences), respectively, in 70% acetonitrile, reduced, alkylated and washed sequentially in 95% acetonitrile and 70% ethanol to remove detergents[62] and digested on beads with trypsin and Lys-C. The resulting peptides were desalted on StageTips and subjected to LC–MS analysis on an EASY nano-LC 1200 system (80–120-min gradients) coupled to an Orbitrap Exploris 480 mass

spectrometer (Thermo Fisher Scientific). MS data were acquired by data-dependent acquisition and subjected to MaxQuant database searches[63] using a human Uniprot-reviewed sequence database, match between runs and label-free quantification.

### AlphaFold-based modelling of protein structures
AlphaFold 3 with its implementation in the AlphaFold Server was used to predict the DNA–protein structures[40]. PyMOL was used for structure modelling and generating high-resolution images. The protein sequences were extracted from the Uniprot database. The input DNA sequence was extracted from the experimentally determined structure of human PCNA with DNA (PDB: 7QO1)[64]. All five AlphaFold-predicted models were aligned or superimposed using the PyMOL software and only models that showed consistency throughout the models are present in the manuscript. PCNA or its yeast homologue POL30 was used as a reference for PyMOL alignment or superimposition, and only the models showing a root mean standard deviation (RMSD) lower than 1.0 are used throughout the manuscript. The electrostatic surface visualization of PCNA–DNA–PAF15 was performed with a plug-in installation of APBS in PyMOL[65]. The interpretation of predicted alignment error plots generated by AlphaFold 3 was performed with the PAE viewer[66]. The PAE plots were further exported as .svg and .png files and only the latter were annotated in Adobe Illustrator to obtain the final figures.

### ABC modelling and analysis of E2F occupancy on *PCNA* and *PAF15* causal enhancers
We downloaded IDR-thresholded peak lists from ENCODE[67] and generated a consensus peak list using iterative overlap peak merging[68]. We counted reads in peaks using UCSC tools and $z$-transformed the counts. To predict causal enhancers in cell type at the *PCNA* and *PAF15* loci, we used the ABC model[69], using the $z$-transformed chromatin accessibility as a proxy for enhancer activity and a power transformation of the distance between the enhancer and the transcription start site as a proxy for contact frequency, as suggested in the ABC paper. We selected enhancers with an ABC score greater than 0.2 as putative causal enhancers. At these enhancers, we calculated the average occupancy of E2F1 and E2F4 and tested for significant differences between the average occupancy across cell types using a paired $t$-test. List of ENCODE DCC experiment accession numbers: ChIP–seq: E2F1 in K562 (ENCSR153DWR), ChIP–seq: E2F4 in K562 (ENCSR368GJN), ATAC-seq: K562 (ENCSR868FGK), ChIP–seq: E2F1 in HepG2 (ENCSR717ZZW), ChIP–seq: E2F4 in HepG2 (ENCSR924LSO), ATAC-seq: HepG2 (ENCSR291GJU), ChIP–seq: E2F1 in MCF7 (ENCSR000EWX), ChIP–seq: E2F4 in MCF7 (ENCSR505NMN), ATAC-seq: MCF7 (ENCSR422SUG), ChIP–seq: E2F1 in HeLa-S3 (ENCSR000EVJ), ChIP–seq: E2F4 in HeLa-S3 (ENCSR000EVL) and DNase-seq: HeLa-S3 (ENCSR959ZXU).

### Public single-cell RNA-seq data analysis
For investigation of single-cell *PAF15* expression in human breast and kidney cancer, features, barcodes and raw read counts from two publicly available datasets were retrieved from the Gene Expression Omnibus repository GSE176078[70] and the Human Cell Atlas project Haniffa-Human-10x3pv2[71]. Both datasets were pre-processed using the pipeline suggested by Seurat (v.4.0.3)[72]. Low-quality cells with fewer than 200 detected features and mitochondrial gene contributions lower than 20% were removed from the kidney single-cell data. Subsequently, normalization, scaling and identification of variable features were done on both datasets using SCTransform[73] regressing out mitochondrial, ribosomal and haemoglobin gene percentage. Doublets were estimated and removed using scDblFinder (v.1.2.0)[74]. Batch integration was performed with Harmony (v.1.2.0)[75]. Dimensional reduction by uniform manifold approximation and projection (UMAP) was recalculated using the integrated lower-dimensional space, using the first 50 principal components and with nearest neighbours set to 15. The original Louvain algorithm was used for community detection. Automated

cell annotation of clusters was done with CellTypist (v.1.6.3)[76] using the Adult_Human_Kidney and Cells_Adult_Breast models for kidney and breast tissue cells, respectively.

## The Cancer Genome Atlas data analysis

Data were extracted from the The Cancer Genome Atlas (TCGA) TARGET GTEx database. For survival analysis, raw read counts were retrieved from all open-access TCGA projects under the data category transcriptome profiling using TCGAbiolinks (v.2.32.0)[77]. Raw counts were batch-corrected using ComBatseq from the sva package (v.3.52.0)[78]. Patients were split on the basis of the best-performing cut-off in gene-expression level between the lower and upper quartiles using Cox proportional hazards regression modelling. Patients were stratified by high and low expression of *PAF15* and *PCNA*. Cut-off values were chosen on the basis of the most significant values from Cox regression on all values from lower to upper quartiles. An estimate of a survival curve was computed using Kaplan–Meier with the survival package (v.3.7.0). For gene-expression analysis in normal and cancer tissue, RSEM-normalized expression data from the TCGA TARGET GTEx cohort ($n = 19,109$) were retrieved from the UCSC Xena platform[79]. All statistics on TCGA-derived data were performed in R, and data visualization was done with ggplot2 (v.3.5.1)[80].

## Reproducibility

For all QIBC, DNA-fibre, proteomics and RNA-seq experiments, a minimum of three technical repeats and a minimum of two biological replicates were performed. Experiments were not randomized, and no blinding was used during data analysis. In box plots, the centre lines are medians, the boxes indicate the 25th and 75th centiles and the whiskers indicate 5% and 95% values. Sample sizes, statistical tests and numbers of replicates for all image-based experiments are specified in the figure legends.

## Reporting summary

Further information on research design is available in the Nature Portfolio Reporting Summary linked to this article.

## Data availability

All source data, including numerical and statistical source data, are provided. SCAR-seq of *PAF15* and OK-seq data are available at the European Nucleotide Archive (ENA) under the PRJEB98295 project.

## Code availability

All code for SCAR-seq and OK-seq is available at GitHub (https://github.com/ManonCoulee/PAF15_Chhetri_2025).

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

**Acknowledgements** This work was supported by research funding from the Lundbeck Foundation Fellowship (R345-2020-1770), an EU Horizon Europe ERC starting grant (META-SURVEILLANCE, 101077859), the Danish Cancer Society (R325-A18913) and the Novo Nordisk Foundation (NNF)(NNF24OC0095792) to K.S. N.-A.P. and J.S.A. were supported by the Independent Research Foundation of Denmark (4283-00362B) and the NNF (ExoAdapt 31199) to J.S.A. N.P. was supported by the CNRS–Inserm programme Atip-Avenir 2021, the Gustave Roussy Foundation and the Agence Nationale de Recherche Scientifique (ANR-24-CE12-3079-Metafork and ANR-25-ERCC-0002-RepliMe); J.G.S.M. was supported by the NNF (NNF21OC0068929 and NNF21SA0072102); F.T.L. and K.R. were supported by the Danish National Research Foundation (grant DNRF141) to the Center for Functional Genomics and Tissue Plasticity (ATLAS); and H.P.-S. was supported by the Czech Science Foundation Junior Star (grant 22-20303M), the EU Horizon 2022 Widera Talent programme (ERA grant agreement 101090292) and an EMBO installation grant (IG-5689-2024). The Danish Molecular Biomedical Imaging Center (DaMBIC) at the University of Southern Denmark is supported by the NNF (NNF18SA0032928). MS and proteomics research at the University of Southern Denmark in Odense is supported by a grant to O.N.J from the NNF to establish the INTEGRA research infrastructure (NNF20OC0061575) and a grant from the Danish Agency of Higher Education and Science to establish PLATO: the Danish National Mass Spectrometry Platform for Proteomics and Biomolecular Imaging (grant 5229-00012B, www.sdu.dk/PLATO). RNA-seq was done at the Functional Genomics and Metabolism research unit at the University of Southern Denmark. We thank R. Nielsen for sequencing assistance, and J. Lukas, S. Mandrup, D. P. Pandey, R. Siersbæk and members of the K.S. laboratory for discussions and critical feedback. For SCAR-seq and OK-seq, we acknowledge the sequencing expertise of the I2BC high-throughput sequencing facility, supported by France Génomique and funded by the French national programme 'Investissement d'Avenir' ANR-10-INBS-09. Model figures were created using BioRender.com (SDU FGM Research Unit licence).

**Author contributions** K.S., G.C. and S.B.B. conceived the project and prepared the figures. G.C. and S.B.B. performed most of the experiments, with support from M.B.P., A.S.G., J.N., G.P.P., B.B. and A.L., under the supervision of K.S. G.C. generated the *PAF15* knockout and inducible complementation cell lines, performed most of the QIBC experiments, co-immunoprecipitation experiments, DNA-fibre assays and survival assays and analysed PLA–QIBC data. S.B.B. generated the PCNA–TurboID cell line and performed PCNA proximity-labelling proteomics, survival assays, most chromatin extractions and western blotting and all PLA experiments. M.B.P. generated the hTERT-RPE1 *p53* and *p53* plus *PAF15* double-knockout lines and contributed to DNA-fibre data analysis. A.S.G. did the MS sample preparation for the PAF15 interactome, isolated RNA for RNA-seq and performed qPCR. G.P.P. and B.B. performed initial experiments supporting project development. J.N. assisted with FRAP and time-lapse data analysis. A.L. contributed to molecular cloning. F.T.L. performed TCGA and single-cell RNA-seq analyses under the supervision of K.R. N.-A.P. and J.S.A. contributed to structural modelling, proteomic data acquisition and analysis related to PCNA proximity proteomics and PCNA and PAF15 stoichiometry. N.F. performed SCAR-seq and OK-seq and M.C. analysed the data, under the supervision of N.P. N.P. performed PAF15 ubiquitination analysis in mouse ES cells. A.K.Y. generated the PAF15 AID–GFP cells under the supervision of H.P.-S. M.F.E. assisted G.C. with the FRAP and STED imaging set-up under the supervision of J.R.B. T.R. performed proteomic data acquisition and analysis for the PAF15 interactome under the supervision of O.N.J. A.F.M. analysed *PCNA/PAF15* ratios from ENCODE RNA-seq data under the supervision of J.G.S.M. J.G.S.M. performed ABC-based enhancer mapping for *PAF15* and RNA-seq analysis after E2F4 depletion. K.S. wrote the manuscript. All authors read and approved the final version.

**Competing interests** The authors declare no competing interests.

**Additional information**
**Correspondence and requests for materials** should be addressed to Kumar Somyajit.

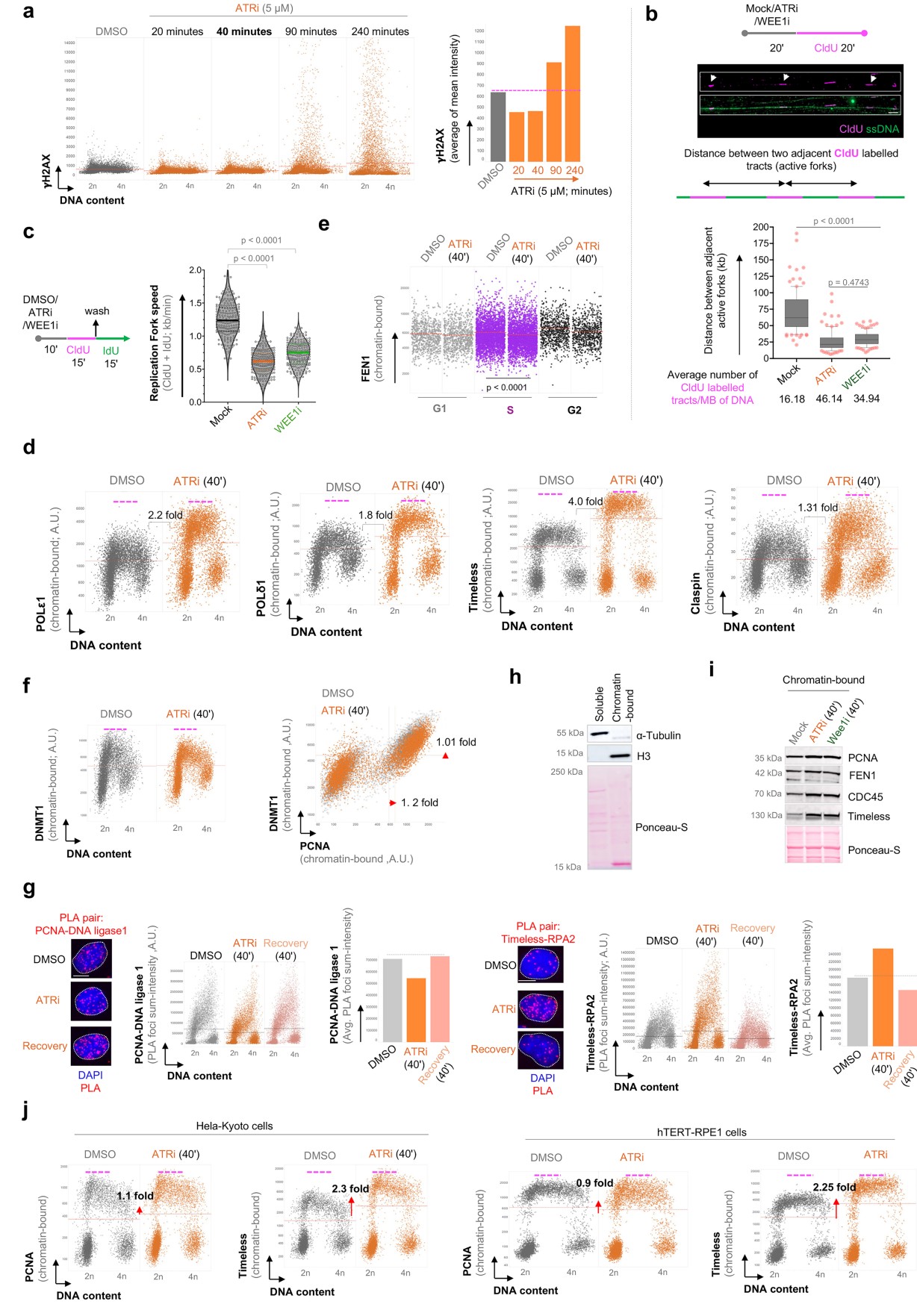

**Extended Data Fig. 1** | See next page for caption.

**Extended Data Fig. 1 | Rapid upregulation of CDK1/2 activities and origin firing after ATR inhibition. a**, Left: QIBC analysis of γH2AX levels in the cells shown in the top panel. Nuclear DNA was counterstained with 4′,6-diamidino-2-phenylindole (DAPI) (2n: G1 phase; 4n: G2 phase, n > 9,000 cells per condition). Right: Quantification of the average mean intensity of γH2AX in cells exposed to the specified time points of ATR inhibition. **b**, Top: DNA-fibre labelling protocol and representative image. Bottom: distance between active forks and average local fork density. ATRi, ATR inhibitor; WEE1i, WEE1 inhibitor (Adavosertib). n = 100 initiation events per condition. In box plots, centre lines denote medians, the boxes indicate the 25th and 75th centiles, the whiskers indicate 10 and 90 percent values. **c**, Left: DNA-fibre labelling protocol. Right: Replication speed of forks after DMSO or treatments with ATRi (5 μM) and WEEi (10 μM). n = 200 fibres for each condition. **d**, QIBC analysis of chromatin-bound POLε1, POLδ1, Timeless, and Claspin in cells treated with DMSO and ATRi (5 μM). 2n: G1 phase, 4n: G2 phase; n > 4,000 cells per condition. Fold change in S-phase specific chromatin-bound levels is indicated. **e**, QIBC analysis of chromatin-bound FEN1 across different cell-cycle phases upon ATR inhibition. Cell-cycle gating was performed using the cell-cycle–specific chromatin-binding profile of PCNA. The horizontal line in QIBC plot depicts the average values. **f**, QIBC of chromatin-bound PCNA and DNMT1 in U2OS cells after ATR inhibition. 2n: G1 phase, 4n: G2 phase; n > 9,000 cells per condition. Fold change in S-phase-specific chromatin-bound levels is indicated. **g**, (Left) Representative images. Cells were exposed to ATRi for 40 min, followed by 40 min of recovery, before being processed for PLA and QIBC. (Right) Quantification of the total PLA focus sum intensity per cell nucleus for the PLA pairs—PCNA and DNA ligase 1, and Timeless and RPA2. The horizontal line depicts the average values for 9,000 cells per condition, Scale bar 10μm. **h**, Subcellular fractionation of U2OS cells followed by immunoblotting for H3 (chromatin-bound fraction) and α-tubulin (soluble fraction). **i**, Immunoblot the indicated proteins from purified chromatin fractions upon 40 min treatment with ATR and WEE1 inhibitors. **j**, QIBC of chromatin-bound PCNA and Timeless in indicated cell lines after ATR inhibition. 2n: G1 phase, 4n: G2 phase; n > 5,000 cells per condition. P values were calculated by one-way ANOVA with Tukey's test. All QIBC measurements were quantified in arbitrary units (A.U.). Each DNA fibre and QIBC plot shown is representative of at least two independent biological replicates.

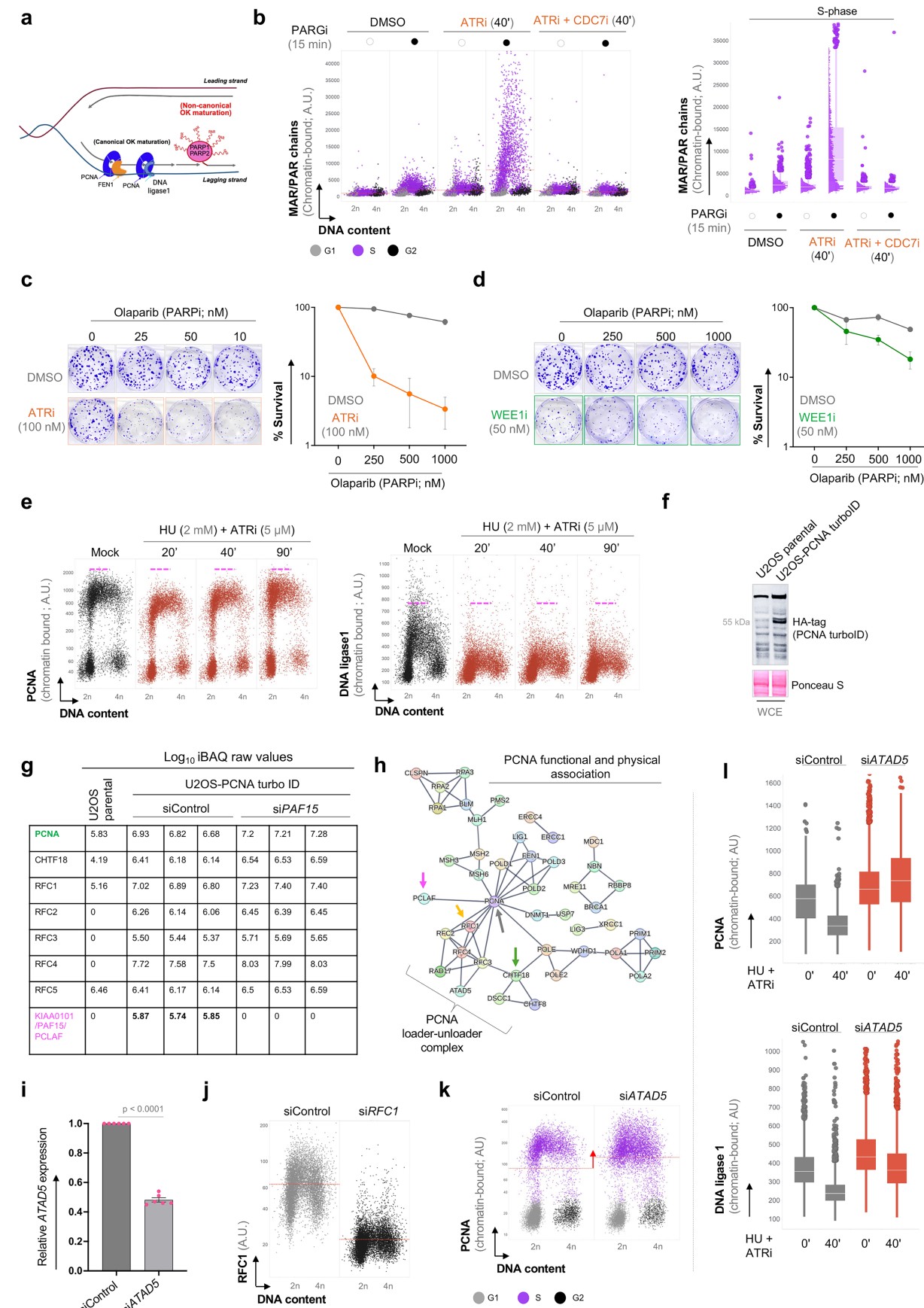

**Extended Data Fig. 2 | See next page for caption.**

**Extended Data Fig. 2 | Natural paucity of chromatin-associated PCNA-lagging strand factors during normal and excessive origin firing.**
**a**, Schematic visualization of non-canonical maturation of OkFs. PARP1/2 processes OkFs unligated by FEN1 and DNA ligase 1 on the lagging strand, indicated by the formation of ADP-ribosylated chains. **b**, (Left) QIBC analysis of the formation of chromatin-bound mono- or poly-ADP-ribosylated (MAR/PAR) chains in U2OS cells treated with PARG, ATR, and CDC7 inhibitor. 2n corresponding to G1 phase and 4n to G2 phase; n > 4,000 cells per condition. (Right) QIBC-based quantification of S-phase chromatin-bound mono- or poly-ADP-ribosylated (MAR/PAR) chains derived from data in the left. In all conditions, cells are treated with PARGi. n > 5,000 cells per condition. All the experimental data are derived from a minimum of 2 independent experiments. **c**, Survival analysis of cells treated with a concentration range of PARP inhibitor Olaparib (250–1,000 nM) in combination with ATRi or **d**, WEE1 inhibition. Values in line plot denote mean ± s.d. **e**, QIBC analysis of chromatin-bound PCNA and DNA ligase 1 in cells treated with hydroxyurea (HU) and ATRi at indicated timepoints. 2n corresponding to G1 phase and 4n to G2 phase; n > 9,000 cells per condition. Note that PCNA (left) and DNA ligase 1 (right) do not accumulate on chromatin beyond the level observed in the mock condition but rather show a decrease in chromatin loading. Pink dotted lines indicate the approximate maximum levels of each protein. **f**, Immunoblot of U2OS cells with TurboID-HA-PCNA tag. **g**, Table showing the results of PCNA proximity-based proteomic screen with TurboID. Raw iBAQ values are shown **h**, Visualization of PCNA-proximal proteins in String network analysis. The lines depict a direct physical and functional association of PCNA with indicated proteins. All the experimental data are derived from minimum of 2 independent experiments. **i**, qPCR analysis of ATAD5 mRNA after siRNA-mediated depletion of *ATAD5* for 48 h. **j**, QIBC analysis of total RFC1 levels after siRNA-mediated depletion of *RFC1* for 48 h. **k**, QIBC analysis of the chromatin-bound PCNA in U2OS cells treated with control or siRNA against *ATAD5* for 48 h. 2n corresponding to G1 phase and 4n to G2 phase; n > 9,000 cells per condition. **l**, QIBC-based quantification of S-phase chromatin-bound PCNA and DNA ligase 1 in U2OS cells with control or siRNA depletion against *ATAD5*, treated with mock DMSO, combined with HU and ATRi. n > 2,500 cells per condition. The dotted horizon line shows average levels. All the experimental data are derived from a minimum of 2 independent experiments. P values were calculated by one-way ANOVA with Tukey's test. All QIBC measurements were quantified in arbitrary units (A.U.). Each DNA fibre and QIBC plot shown is representative of at least two independent biological replicates. The schematic in **a** was created with BioRender.com (Somyajit, K. (2025) https://BioRender.com/7yy82vn).

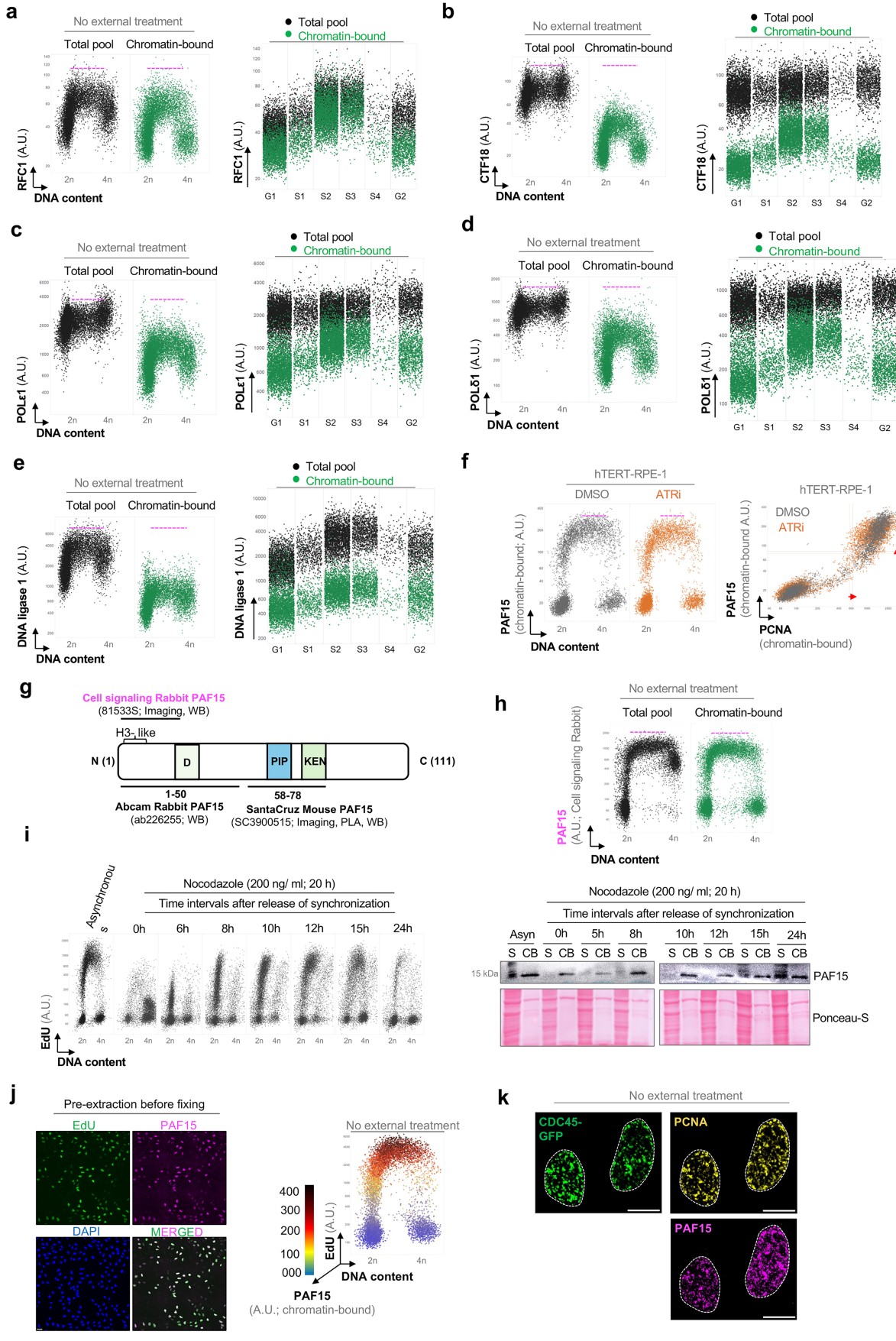

**Extended Data Fig. 3** | See next page for caption.

**Extended Data Fig. 3 | Comparison of the total and chromatin-bound pool of PCNA-associated proteins in S phase and PAF15 as a rate-limiting, bona-fide replisome protein. a**–**e**, QIBC analysis of total and chromatin-bound fractions of RFC1, CTF18, POLε1, POLδ1, and DNA ligase 1 in naïve U2OS cells. EdU and DAPI to stratify cell-cycle progression. n > 8,000 cells. **f**, QIBC analysis of PAF15 and PCNA in hTERT-RPE1 cells treated with ATR inhibitor. 2n corresponding to G1 phase and 4n to G2 phase; n > 5,000 cells. **g**, Schematic depiction of binding sites of different PAF15 antibodies. **h**, QIBC analysis of soluble and chromatin-bound fractions of PAF15 in U2OS cells using Cell Signaling antibody, shown in pink. **i**, (Left) QIBC analysis of cell cycle with EdU following a 20 h G2M block with microtubule dynamics inhibitor Nocodazole (200 ng/ml) at indicated timepoints post-release. (Right) Western blot analysis of the protein levels of PAF15 soluble (S) and chromatin-bound (CB) fractions. In QIBC plots, 2n corresponds to G1 phase and 4n to G2 phase; n > 10,000 cells per condition. **j**, (Left) Representative images of chromatin-bound EdU and PAF15 in U2OS. Cell nuclear DNA counterstained by DAPI. (Right) The corresponding QIBC correlates with EdU with PAF15. The colour gradient indicates PAF15 level. 2n corresponding to G1 phase and 4n to G2 phase; n > 9,000 cells. **k**, Representative 3D confocal microscopy images showing co-localization of PAF15 and PCNA in U2OS cells. Scale bar, 10 μm.

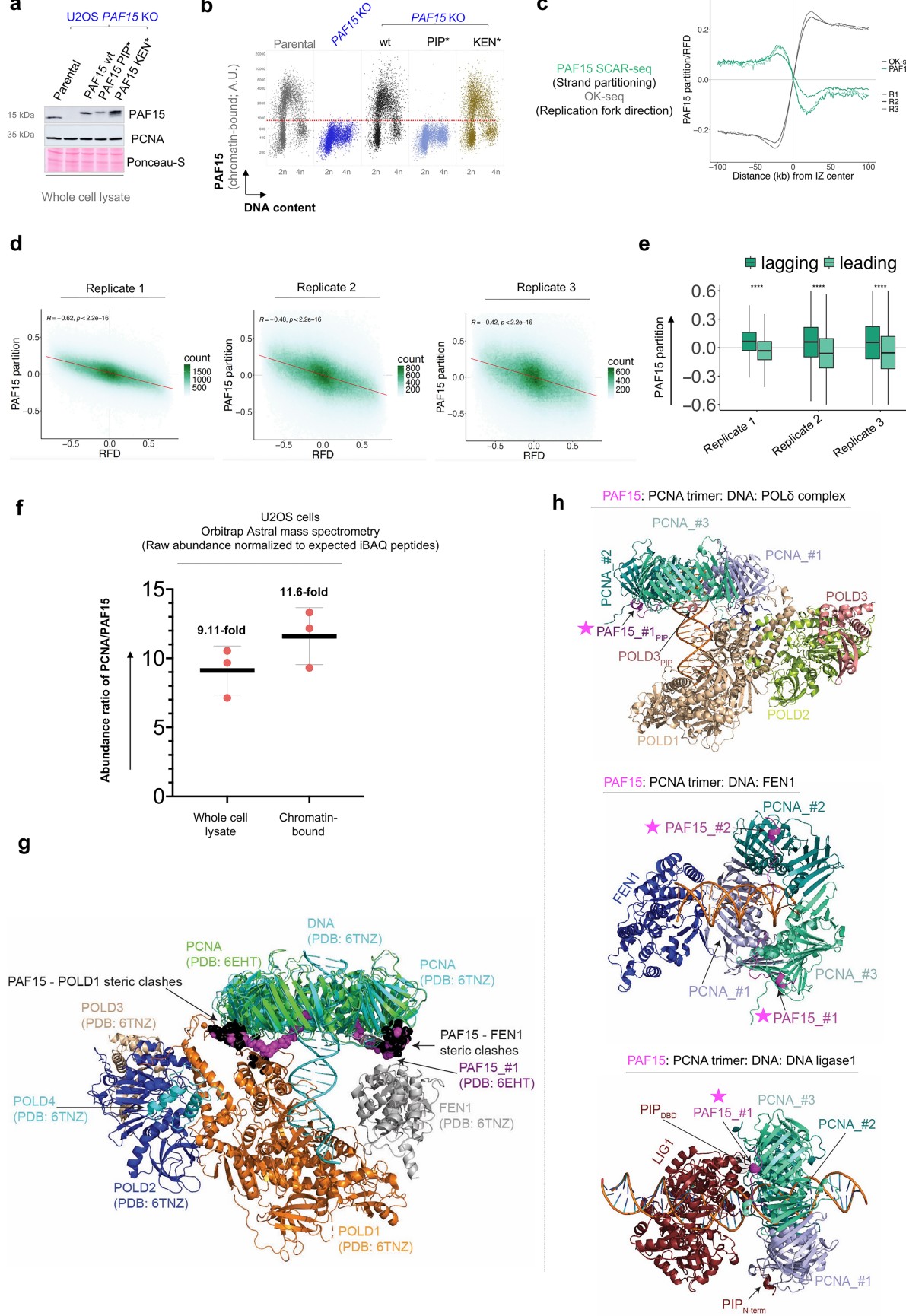

**Extended Data Fig. 4** | See next page for caption.

**Extended Data Fig. 4 | PAF15 is recruited to the active replisome by PCNA and is exclusively present at the lagging strand. a**, Immunoblot of the PAF15 in whole-cell extracts (WCE) of U2OS cells with *PAF15* KO and constitutive complementation of wild type and mutated PIP/KEN-box. **b**, QIBC analysis of the chromatin-bound fraction of PAF15 in naïve U2OS and U2OS *PAF15* KO stably expressing indicated versions of PAF15. 2n corresponding to G1 phase and 4n to G2 phase; n > 5,000 cells per condition. All the experimental data are derived from minimum of 2 independent experiments. **c**, Average profile of two biological replicates of OK-seq and 3 biological replicates of SCAR-seq. **d**, Hexplot of RFD versus PAF15 partitioning. Spearman's rank correlation coefficient is shown in the top right corner. Spearman's rank correlation coefficient is shown in the top right corner. Replicates 1-3 of PAF15 SCAR-seq versus replicate 1 of OK-seq **e**, PAF15 partitioning at downstream (leading, light green shaded) and upstream edges (lagging, dark green shaded) of initiation zones with a significant partition difference in each replicate (paired Wilcoxon signed-rank test), with p < 2.2⁻¹⁶, data derived from three experiments. All the QIBC experimental data are derived from a minimum of 2 independent experiments. **f**, MS analysis of whole-cell lysate and chromatin-bound PCNA/ PAF15 ratio derived from normalizing raw abundance to expected iBAQ peptides (PCNA = 16, PAF15 = 6). **g**, PCNA allows binding for either PAF15 or POLδ–FEN1. Rigid-body alignment of the PCNA–PAF15–DNA crystal structure (PDB: 6EHT) with the cryo-EM structure of PCNA–POLδ–FEN1–DNA (PDB: 6TNZ) reveals severe steric clashes occurring between PAF15 and POLδ1–FEN1 (black spheres). The alignment is performed on corresponding PCNA molecules using the align function of PyMOL (RMSD 0.95 Å). **h**, AlphaFold predicts a nonameric PCNA–PAF15-POLδ-DNA complex. (Top) PAF15 occupies one PCNA monomer whilst POLδ1 and POLδ3 saturates the other two. The model is predicted with high confidence as revealed by predicted alignment errors (PAE < 10 Å, Middle) and high per-residue confidence scores (pLDDT>90, Right). A few intrinsically disordered regions that comprises the residues 1–74 of POLδ1, 457–467 of POLδ2, 144–455 of POLδ3 and 1–40 of POLδ4 are predicted with very low pLDDT scores and therefore omitted from the model for clarity. (Middle)The structural model of a heptameric PCNA–PAF15–FEN1–DNA shows that docking of FEN1 to one PCNA monomer allows PAF15 to occupy the other two (Left). The model is predicted with very high confidence as highlighted by low PAEs (Middle) and high pLDDTs (Right). (Bottom) AlphaFold predicts a hexameric PCNA–PAF15– DNA ligase 1–DNA complex where DNA ligase 1 binds the PCNA trimer by its PIP N-terminus and PIPDBD domains, allowing one PAF15 molecule to access the unoccupied PCNA (Left). The model is predicted with high confidence as revealed by low PAE scores (Middle) and high pLDDT scores (Right). Intrinsically disordered regions present at the N- and C-termini of PAF15 and between the residues 1–260 of DNA ligase 1 are predicted with very low pLDDT scores and thus omitted from the model for clarity.

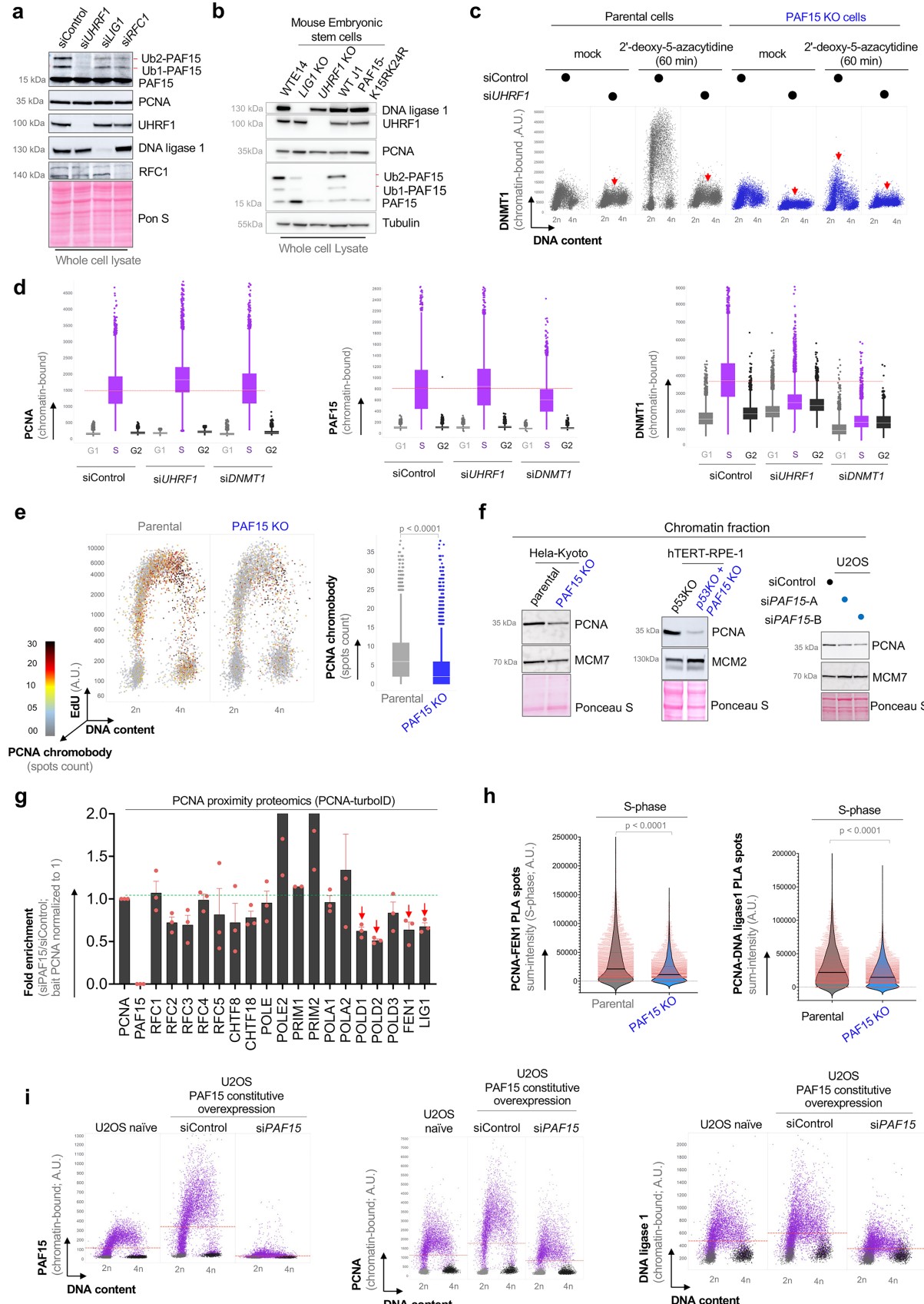

**Extended Data Fig. 5** | See next page for caption.

**Extended Data Fig. 5 | PAF15 contributes to the chromatin stability of PCNA lagging-strand factors. a**, Immunoblot of whole-cell extracts from U2OS cells treated with control siRNA and siRNA against *UHRF1*, *DNA ligase 1* and *RFC1* for 48 h. PAF15 dual mono-ubiquitination (ub1 and ub2) are marked with red arrows. **b**, Immunoblot of indicated proteins in whole-cell extract of *LIG1* KO (DNA ligase 1 knockout), *UHRF1* KO (UHRF1 knockout) and PAF15-K15RK24R (PAF15 K15R/K24R double mutant) in mouse embryonic stem cells. **c**, QIBC analysis of chromatin-bound DNMT1 in parental and *PAF15* KO U2OS cells treated with siRNA against *UHRF1* and exposed to Decitabine (5-aza-2′-deoxycytidine) for 60 min. 2n corresponding to G1 phase and 4n to G2 phase; n > 7,000 cells per condition. **d**, QIBC-based analysis of cell-cycle phase chromatin-bound PCNA (left), PAF15 (middle) and DNMT1 (right) in U2OS cells treated with control siRNA and siRNA against *UHRF1* and *DNMT1*. Red dotted horizontal line in each box plot indicates the average S-phase chromatin-bound PCNA, PAF15, and DNMT1. All the experimental data are derived from minimum of 2 independent experiments. **e**, (Left) QIBC of PCNA chromobody foci in U2OS parental and *PAF15* KO cells. n > 9,000 cells. (Right) Quantification of PCNA chromobody foci in EdU-positive S-phase cells derived from data in the Middle. P values were determined by one-way ANOVA with Tukey's test. **f**, Immunoblot validation of destabilization of chromatin-bound PCNA in (left) HeLa Kyoto *PAF15* KO (middle) hTERT-RPE1 *p53* KO *PAF15* KO (right) with two PAF15 siRNAs in U2OS cells. **g**, Proximity-based proteomic analysis of TurboID–PCNA in U2OS with PAF15-depletion. PCNA was normalized to 1 in both siControl and si*PAF15* cells. **h**, QIBC-based quantification of S-phase PLA foci of PCNA–FEN1 and PCNA–DNA ligase 1 in U2OS parental and *PAF15* KO cells. n > 5,000 S-phase cells. P values were determined by one-way ANOVA with Tukey's test. **i**, QIBC analysis of chromatin-bound PAF15 (left), PCNA (middle), and DNA ligase 1 (right) in naïve and constitutively expressing ectopic PAF15 upon treatment with control and siRNA against *PAF15*.

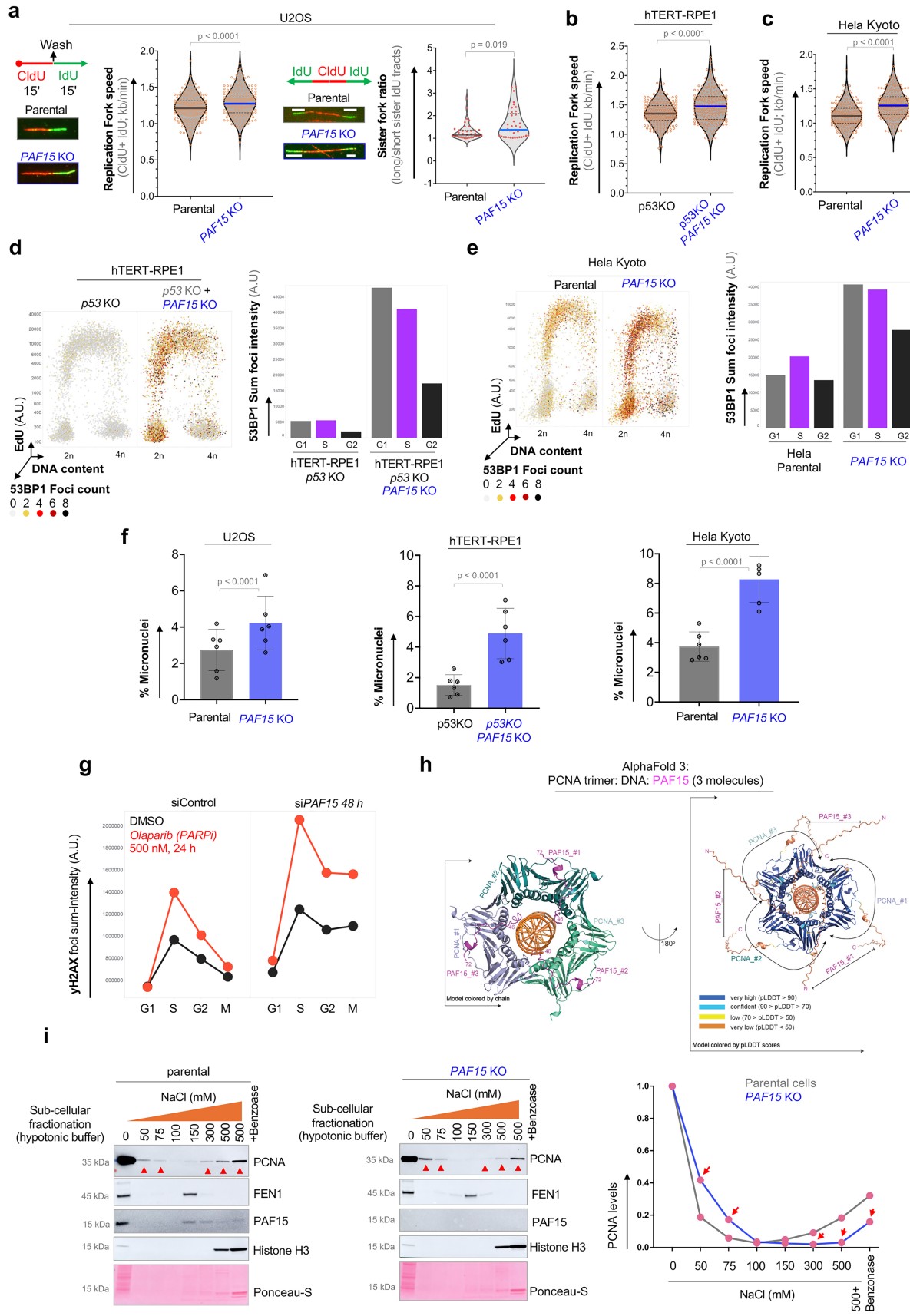

**Extended Data Fig. 6** | See next page for caption.

**Extended Data Fig. 6 | Loss of PAF15 fuels DNA replication stress. a**, (Left) DNA-fibre labelling protocol and representative images. Replication fork speed of parental and *PAF15*-KO U2OS cells. n = 200 fibres per condition. P values were determined by one-way ANOVA with Tukey's test. Right: Analysis of sister fork asymmetry in U2OS cells. n = 50 bidirectional fibres per condition. P values were determined by one-way ANOVA with Tukey's test. **b,c**, Replication fork speed comparison in indicated cell lines between parental and *PAF15* KO. n = 200 fibres per condition. P values were determined by one-way ANOVA with Tukey's test. **d,e**, QIBC of 53BP1 foci/nuclear bodies in *p53* KO and HeLa Kyoto parental and *PAF15* KO cells labelled with EdU and DAPI to stratify cell-cycle progression. n > 5,000 cells for each condition; colours indicate the number of 53BP1 nuclear bodies per nucleus. **f**, Quantification of micronuclei formation in indicated parental and *PAF15* KO cell lines. n = 500 cells per condition. All the experimental data are derived from minimum of 2 independent experiments. **g**, QIBC analysis of yH2AX foci in PAF15-depleted U2OS cells treated with PARP inhibitor Olaparib (500 nM, 24 h). 2n: G1, 4n: G2; n > 10,000 cells per condition. Cells are labelled with EdU to stratify cell-cycle gating and progression. **h**, AlphaFold model of PAF15-PCNA–DNA complex. Up to 3 molecules of PAF15 can interact with PCNA. PAF15 binds as a molecular wedge between the inner PCNA ring and DNA. This structural model is folded with high confidence, indicated by a high pLDDT score (right). **i**, Western blot of the indicated protein. chromatin fractions are exposed to increasing salt concentration after the extraction from the parental (left) and *PAF15* KO (middle) U2OS cells. (Right) The graph shows quantification of PCNA extraction from the corresponding western blots. The red arrow indicates differences in PCNA extraction from parental (grey) and *PAF15* KO (blue) cells.

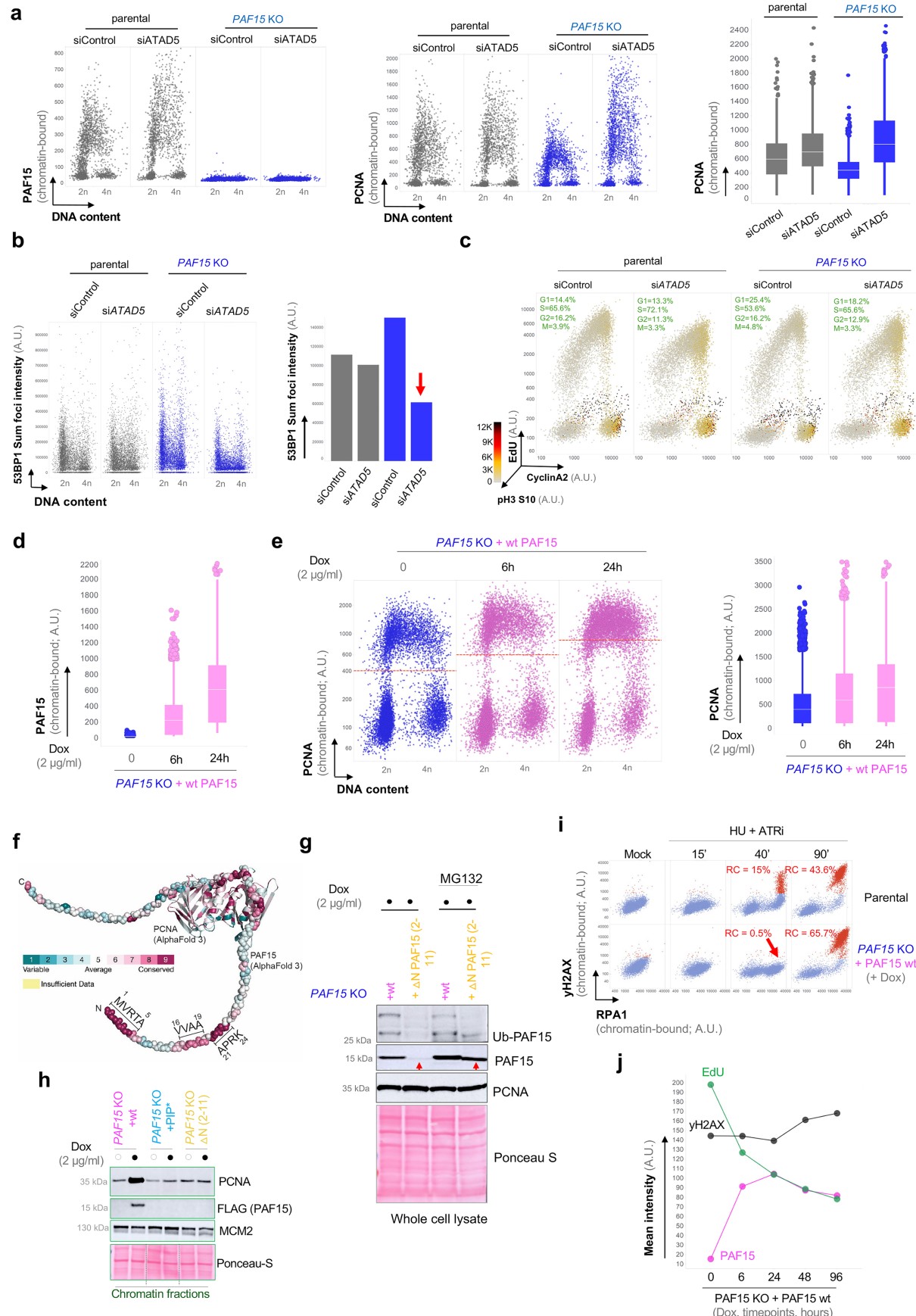

**Extended Data Fig. 7** | See next page for caption.

**Extended Data Fig. 7 | PAF15 prevents unscheduled chromatin unloading of PCNA from the action of ATAD5. a**, QIBC analysis of chromatin fractions of PAF15 (left) and PCNA (middle) and its quantification (right) in U2OS and *PAF15* KO cells with ATAD5 depletion. n > 2,000 cells were analysed per condition **b**, QIBC analysis of 53BP1 sum foci intensity and its quantification (right) in U2OS parental and *PAF15* KO cells treated with control siRNA and siRNA against *ATAD5*. n > 5,000 cells were analysed per condition. The red arrow indicates the ATAD5 loss in *PAF15* KO cells. **c**, QIBC analysis of S-phase (EdU), CyclinA2 (CycA2) and mitotic (phosphorylated serine 10 on histone 3 – pH3 S10) in U2OS and *PAF15* KO treated with control siRNA and siRNA against *ATAD5*. n > 8,000 cells were analysed per condition. The percentage of each cell-cycle phase (G1, S, G2 and M) is mentioned in green colour. **d,e**, QIBC analysis of the chromatin-bound PAF15 (**d**) and PCNA (**e**) in doxycycline-induced overexpression of PAF15 WT in *PAF15* KO (U2OS) cells. > 5,000 cells were analysed per condition. **f**, Evolutionary conservation in amino acid positions for PCNA and PAF15 generated with the ConSurf Server. The conservation plot has been constructed on the AlphaFold 3-predicted structure of PCNA trimer in complex with two PAF15 molecules and DNA. For clarity, only one molecule of PCNA and PAF15 was used as a scaffold for displaying the evolutionary conservation of amino acids. **g**, Immunoblot of PAF15 and PCNA in U2OS *PAF15* KO with doxycycline overexpression of either WT or N-terminus truncated versions of PAF15 cells treated with proteasomal inhibitor MG132. **h**, Immunoblot of chromatin-bound PCNA and FLAG-tagged PAF15 versions: WT, PIP mutant and N-terminus truncated PAF15 in *PAF15* KO (U2OS) with doxycycline-induced overexpression. **i**, QIBC analysis of chromatin fractions of yH2AX and RPA1 in parental and doxycycline overexpression of PAF15 in U2OS *PAF15* KO cells treated with a combination of hydroxyurea (HU) and ATR inhibitor. 2n: G1, 4n: G2; n > 5,000 cells per condition. All the experimental data are derived from minimum of 2 independent experiments. **j**, QIBC derived average mean values of EdU, yH2AX and PAF15 in the indicated time points of doxycycline induction in U2OS PAF15 KO doxycycline overexpression PAF15 cells. n > 8,000 cells per condition.

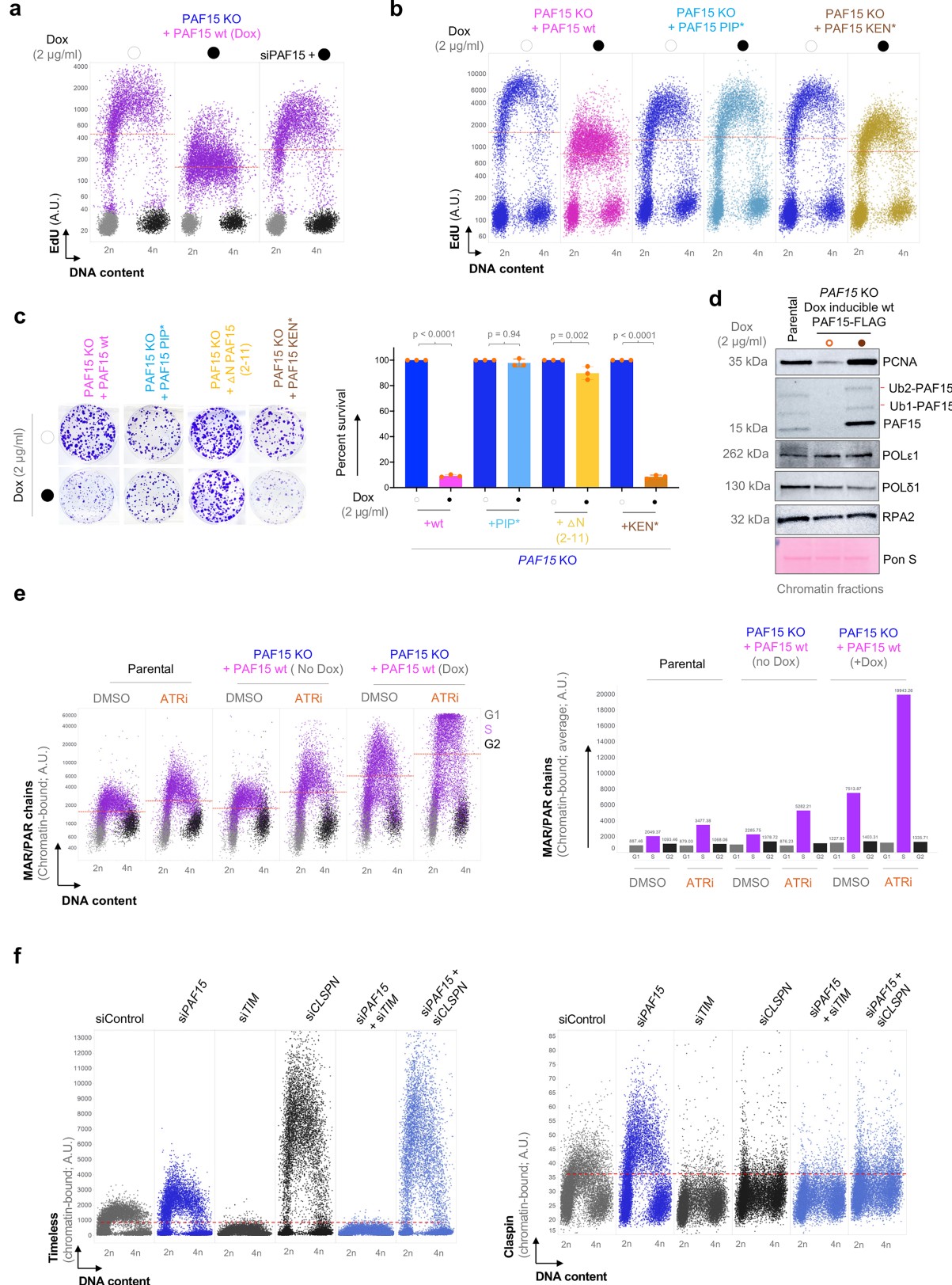

**Extended Data Fig. 8** | See next page for caption.

**Extended Data Fig. 8 | *PAF15* overexpression acutely compromises cell viability. a**, QIBC analysis of EdU in *PAF15* KO and *PAF15*-overexpressing U2OS cells treated with *PAF15* siRNA. 2n: G1, 4n: G2; n > 8,000 cells per condition. **b**, QIBC analysis of EdU in U2OS *PAF15* KO cells doxycycline-induced overexpression of WT, PIP-box mutated and KEN-box mutated PAF15. 2n: G1, 4n: G2. n > 8,000 cells per condition **c**, (Left) Representative images of cell survival assay plates (right) survival analysis in U2OS cells overexpressing WT, PIP-box mutated, N-terminus truncated and KEN-box mutated PAF15. **d**, Immunoblot of indicated chromatin-bound proteins in U2OS parental and *PAF15* KO doxycycline PAF15 WT overexpression cells in the absence or presence of doxycycline. **e**, (Left) QIBC analysis of chromatin-bound mono- and poly-ADP-ribosylated (MAR/PAR) chain in PAF15 overexpression U2OS cells treated with PARG and ATR inhibitor. 2n: G1, 4n: G2; n > 8,000 cells per condition. (Right) Quantification of MAR/PAR chain intensity in different cell-cycle phases gated on PCNA. Average values are indicated in different cell-cycle phases. **f**, QIBC analysis of chromatin-bound Timeless, Claspin in U2OS cells with depletion of the indicated protein's gene expression. 2n: G1, 4n: G2; n > 8,000 cells were analysed per condition. Red dotted horizontal line in QIBC scatter plots indicates the approximate background signal of Timeless or Claspin on chromatin. All the experimental data are derived from minimum of 2 independent experiments.

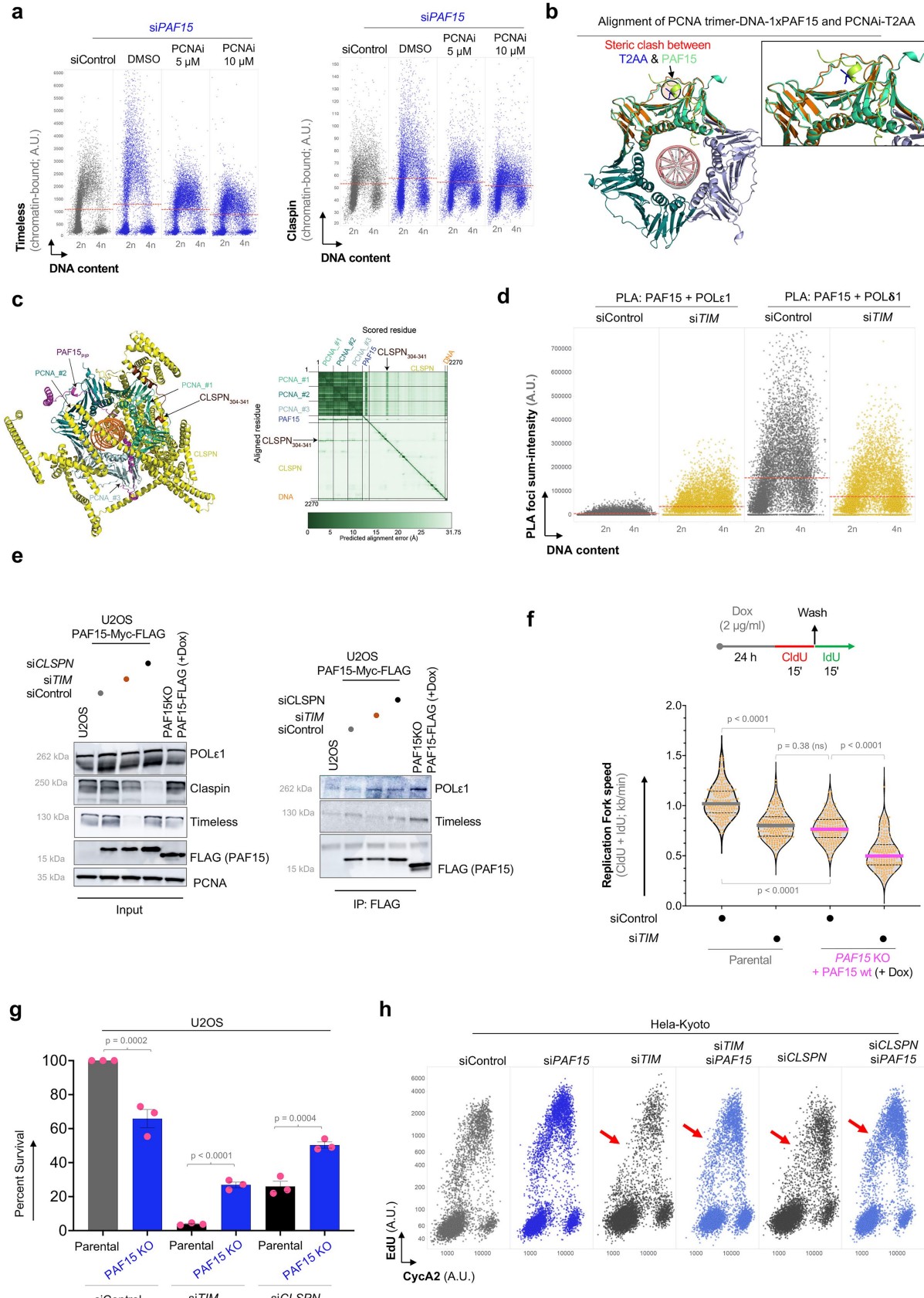

**Extended Data Fig. 9** | See next page for caption.

**Extended Data Fig. 9 | PAF15 access to leading strand PCNA is lethal and is counteracted by Timeless and Claspin. a**, QIBC analysis of chromatin-bound (left) Timeless and (right) Claspin in U2OS cells with simultaneous depletion of PAF15 and addition of PCNA inhibition (T2AA, 5 and 10 μM). **b**, Modelling of PCNA inhibitor T2AA (PDB: 3WGW) onto the AlphaFold 3-predicted structure of PCNA (trimer)-DNA-PAF15 shows a steric clash between PAF15 PIP and T2AA. **c**, AlphaFold-predicted structural model of three PCNAs in complex with PAF15, Claspin (CLSPN), and dsDNA shows high confidence in the folding of PCNA and DNA and very low confidence in the folding of CLSPN. According to the PAE score of the model, the 304–341 residues of Claspin that contain the PIP motif (residues 311–318) are in the closest proximity of PCNA, but they are within a predicted alignment error of 20–25 Å, which can be situated beyond the interaction distance. **d**, QIBC analysis of PLA focus sum intensity in pairs of PAF15-POLε1 and PAF15-POLδ1 in U2OS cells treated with control siRNA and siRNA of *TIMELESS*. S-phase quantification is used in Fig. 5b. **e**, Immunoprecipitation of FLAG-tagged PAF15 in U2OS parental, U2OS cells constitutively expressing PAF15-Myc-FLAG and U2OS *PAF15* KO cells doxycycline inducible PAF15 WT. U2OS cells constitutively expressing PAF15-Myc-FLAG were treated with control, *TIMELESS*, *CLSPN* siRNAs and indicated proteins from input and Flag IP lysates were analysed by western blot. **f**, (Top) Schematic of DNA-fibre protocol, (bottom) Replication fork speed in U2OS parental and U2OS *PAF15* KO cells treated with 20 μg/ml doxycycline to induce WT PAF15 expression upon *TIMELESS* gene depletion. n = 200 fibres per condition. P values were determined by one-way ANOVA with Tukey's test, n = 2 experiments. **g**, Survival analysis of parental and *PAF15*-KO U2OS cells with depletion of Timeless and Claspin. Values in bar graphs denote mean ± s.d. All the P values were determined by one-way ANOVA with Tukey's test. **h**, QIBC analysis of Cyclin A2 and EdU in HeLa Kyoto cells with depletion of indicated proteins. 2n: G1, 4n: G2, n > 5,000 cells were analysed per condition. Red arrows indicate S-phase population revival in Timeless or Claspin depletion together with PAF15. All the experimental data are derived from a minimum of 2 independent experiments.

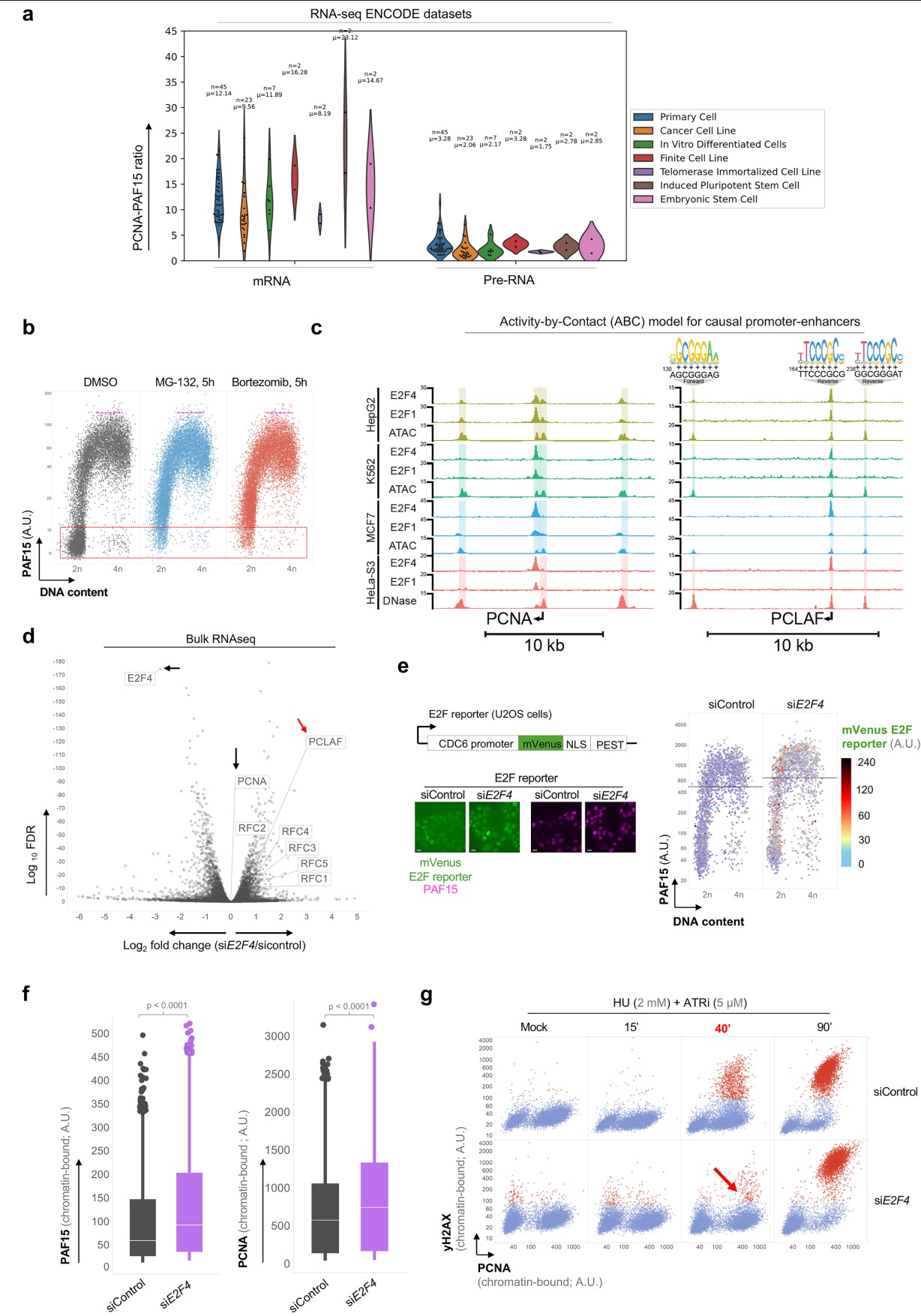

**Extended Data Fig. 10** | See next page for caption.

**Extended Data Fig. 10 | PAF15 expression is strictly regulated by the E2F family of transcription factors. a**, PCNA–PAF15 ratio of pre-RNA and mRNA of the indicated cell lines retrieved from RNA-seq ENCODE datasets; n: number of replicates of each cell line, μ: mean of PCNA–PAF15 ratio. **b**, QIBC analysis of PAF15 levels in U2OS cells treated with proteasome inhibitors MG132 and Bortezomib for 5 h. 2n: G1, 4n: G2: n n > 10,000 cells. The rectangular red box depicts that PAF15 is stabilized in G1 cell population upon proteasome inhibition. Pink dotted lines in QIBC scatter plots indicate the approximate maximum levels of each protein. **c**, Screenshot of Detected peaks of ATAC-seq/DNase-seq and E2F1/4 ChIP–seq from indicated cells at the PCNA and PAF15 (PCLAF) locus (data are representative of two independent experiments, irreproducible discovery rate cut-off = 0.05). Highlighted regions indicate enhancers predicted to be causal using the Activity by Contact (ABC) model. **d**, Volcano plot depicting differentially expressed genes in cells treated with control or E2F4 siRNA. Y-axis denotes-log10 P values, while X-axis shows log2 fold change values showing enrichment of PAF15 with no change of PCNA expression upon E2F4 depletion. **e**, (Left) Scheme of the E2F transcriptional reporter. Representative microscopy image of U2OS cells expressing mVenus (E2F reporter). (Right) QIBC analysis of E2F activity and PAF15 protein levels in U2OS cells stably expressing E2F transcriptional reporter treated with control and E2F4 siRNA for 48 h. The colour gradient indicates the mVenus signals translating to E2F1 activity at the single-cell level. All values are in arbitrary units (A.U.). A total of 2,000 cells were analysed per condition, n = 2 experiments. **f**, QIBC analysis of chromatin-bound levels of PAF15 and PCNA under E2F4 depletion. A total of 10,000 cells were analysed per condition. P values were determined by one-way ANOVA with Tukey's test. **g**, U2OS cells treated with control and E2F4 siRNA were incubated with HU (2 mM) and ATRi (5 μM) for the indicated times and analysed by QIBC to assess the chromatin binding of RPA1 and $\gamma$H2AX and PCNA. The arrow at 40-minute time point indicates the onset of replication catastrophe. 2n: G1, 4n: G2: n > 10,000 cells per condition.

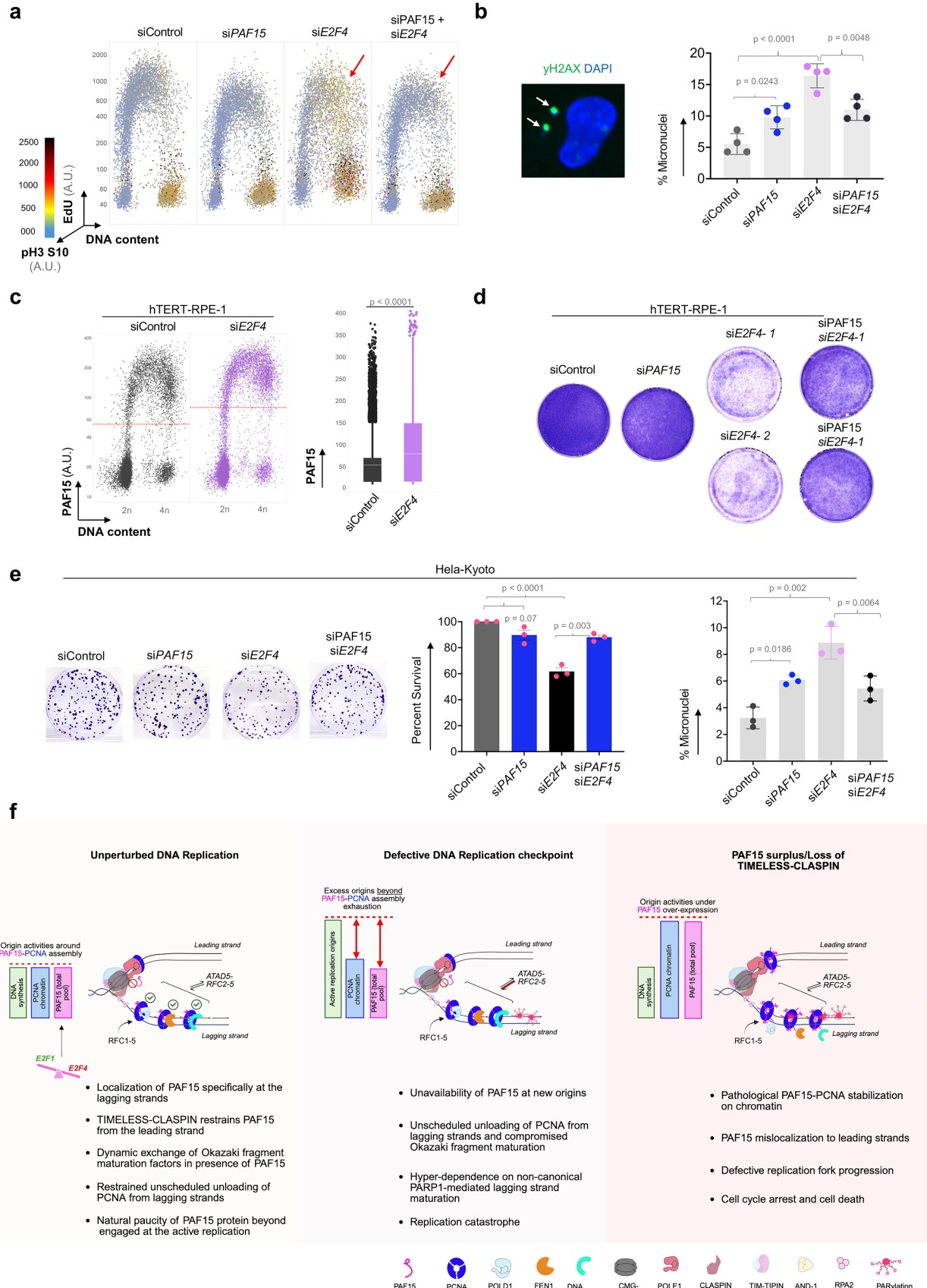

**Extended Data Fig. 11 |** See next page for caption.

**Extended Data Fig. 11 | PAF15-dependent rate-limiting mechanism for PCNA lagging-strand processing during unperturbed DNA synthesis.** **a**, QIBC analysis of S-phase and mitotic fractions of U2OS cells, as indicated by EdU and phosphorylation of Serine 10 on Histone H3 (pH3 S10), respectively, treated by siRNAs targeting PAF15 and E2F4. n > 5,000 cells. n = 2 experiments **b**, Quantification of micronucleation (500 nuclei per condition) in U2OS cells treated with indicated siRNAs. Values denote mean ± s.d. P values were determined by one-way ANOVA with Tukey's test. **c**, QIBC analysis of PAF15 levels in hTERT-RPE1 with depleted E2F4 transcription factor. The horizontal line depicts the average values for 10,000 cells per condition. **d**, Survival analysis of simultaneous depletion of PAF15 and E2F4 in hTERT-RPE1. **e**, (Left) Representative images of survival assay plates, (Middle) Quantification of the number of colonies formed during survival assay on HeLa Kyoto cells depleted for PAF15 and E2F4.

(Right) Quantification of micronuclei formation in HeLa Kyoto cells depleted for PAF15 and E2F4. Values in bar graphs denote mean ± s.d. All the P values were determined by one-way ANOVA with Tukey's test. All the experimental data are derived from a minimum of 2 independent experiments. **f**, (Left) Schematic depicting origin firing and chromatin-bound PCNA–PAF15 in lagging-strand maturation on replicating chromatin. (Middle) Schematic, loss of PAF15 compromises PCNA chromatin stability by unleashing ATAD5 activity and is dependent on non-canonical, PARP1-mediated, lagging-strand processing pathways. (Right) Schematic representation of surplus PAF15 on chromatin stabilizes PCNA–PAF15 assembly and compromises replication fork progression. The model in **f** was created with BioRender.com (Somyajit, K. (2025) https://BioRender.com/f8sb057).

# Reporting Summary

## Statistics

For all statistical analyses, confirm that the following items are present in the figure legend, table legend, main text, or Methods section.

| n/a | Confirmed | |
|---|---|---|
| ☐ | ☒ | The exact sample size (*n*) for each experimental group/condition, given as a discrete number and unit of measurement |
| ☐ | ☒ | A statement on whether measurements were taken from distinct samples or whether the same sample was measured repeatedly |
| ☐ | ☒ | The statistical test(s) used AND whether they are one- or two-sided<br>*Only common tests should be described solely by name; describe more complex techniques in the Methods section.* |
| ☒ | ☐ | A description of all covariates tested |
| ☒ | ☐ | A description of any assumptions or corrections, such as tests of normality and adjustment for multiple comparisons |
| ☐ | ☒ | A full description of the statistical parameters including central tendency (e.g. means) or other basic estimates (e.g. regression coefficient) AND variation (e.g. standard deviation) or associated estimates of uncertainty (e.g. confidence intervals) |
| ☐ | ☒ | For null hypothesis testing, the test statistic (e.g. *F*, *t*, *r*) with confidence intervals, effect sizes, degrees of freedom and *P* value noted<br>*Give P values as exact values whenever suitable.* |
| ☒ | ☐ | For Bayesian analysis, information on the choice of priors and Markov chain Monte Carlo settings |
| ☒ | ☐ | For hierarchical and complex designs, identification of the appropriate level for tests and full reporting of outcomes |
| ☒ | ☐ | Estimates of effect sizes (e.g. Cohen's *d*, Pearson's *r*), indicating how they were calculated |

*Our web collection on statistics for biologists contains articles on many of the points above.*

## Software and code

Policy information about availability of computer code

| Data collection | scanR acquisition software (Olympus v.3.4 for scanR high-content microscope)<br>Nikon NIS-Elements AR software (version 5.20.02 for confocal and FRAP data acquisition)<br>iMSPECTOR (version 16.3.13787 STED data acquisition) |
|---|---|
| Data analysis | ScanR analysis software (Olympus version version 12.4 .3.4 for QIBC)<br>Spotfire TIBCO Software (version 12.4 for  QIBC and RNA-seq data visualization)<br>Graphpad Prism (version 10.0 for data visualization and statistics)<br>PyMOL Molecular Graphics System (Schrodinger Licensing version 3.1)<br>R studio ggplot2 (version 3.5.1 for TCGA data analysis)<br>Fiji software (DNA fiber) |

For manuscripts utilizing custom algorithms or software that are central to the research but not yet described in published literature, software must be made available to editors and reviewers. We strongly encourage code deposition in a community repository (e.g. GitHub). See the Nature Portfolio guidelines for submitting code & software for further information.

## Data

Policy information about availability of data

All manuscripts must include a data availability statement. This statement should provide the following information, where applicable:
- Accession codes, unique identifiers, or web links for publicly available datasets
- A description of any restrictions on data availability
- For clinical datasets or third party data, please ensure that the statement adheres to our policy

> All the source data, including numerical and statistical source data for QIBC, DNA fiber, and mass spectrometry analyses, as well as the uncropped Western blots. provided with the manuscript. All codes for SCAR-seq and OK-seq are available at GitHub (https://github.com/ManonCoulee/PAF15_Chhetri_2025). SCAR-seq of PAF15 and OK-seq data are available at ENA under the PRJEB98295 project/. Any additional data or information in support of this study will be available from corresponding authors upon reasonable request.

## Research involving human participants, their data, or biological material

Policy information about studies with human participants or human data. See also policy information about sex, gender (identity/presentation), and sexual orientation and race, ethnicity and racism.

| | |
|---|---|
| Reporting on sex and gender | Not applicable because this study does not involve human research participants |
| Reporting on race, ethnicity, or other socially relevant groupings | Not applicable because this study does not involve human research participants |
| Population characteristics | Not applicable because this study does not involve human research participants |
| Recruitment | Not applicable because this study does not involve human research participants |
| Ethics oversight | Not applicable because this study does not involve human research participants |

Note that full information on the approval of the study protocol must also be provided in the manuscript.

# Field-specific reporting

Please select the one below that is the best fit for your research. If you are not sure, read the appropriate sections before making your selection.

☒ Life sciences        ☐ Behavioural & social sciences        ☐ Ecological, evolutionary & environmental sciences

For a reference copy of the document with all sections, see nature.com/documents/nr-reporting-summary-flat.pdf

# Life sciences study design

All studies must disclose on these points even when the disclosure is negative.

| | |
|---|---|
| Sample size | No statistical method was used to predetermine the sample size. Sample size was determined to be adequate on the levels and consistency of the differences between the groups. For all QIBC, experiments minimum of three technical repeats and minimum of two biological replicates were performed. Sample size and statistical tests for all image-based experiments are specified in the figure legends and adhere to the current standard in the high-content imaging field (Toledo et al., Cell. 2013 Nov 21;155(5):1088-103.; Somyajit et al. Science. 2017 Nov 10;358(6364):797-802; Sedlackova et al., Nature. 2020 Nov;587(7833):297-302; Somyajit et al., Dev Cell. 2021 Feb 22;56(4):461-477). |
| Data exclusions | Failed experiments with sub-optimal cell confluency and staining or failing to pass quality control measures were excluded for data acquisition. All data acquired were included in the analysis. |
| Replication | All attempts of replication were successful. |
| Randomization | No randomization was done. Samples were organized into groups based on treatments and conditions. Appropriate controls were included in all experiments and duly indicated in the figure legends. |
| Blinding | Blinding was not done. For all QIBC related data acquisition and analysis automated pipeline was employed. All other experiments were perfomed in unbiased fashion and included appropriate controls. |

# Reporting for specific materials, systems and methods

We require information from authors about some types of materials, experimental systems and methods used in many studies. Here, indicate whether each material, system or method listed is relevant to your study. If you are not sure if a list item applies to your research, read the appropriate section before selecting a response.

## Materials & experimental systems

| n/a | Involved in the study |
|---|---|
| ☐ | ☒ Antibodies |
| ☐ | ☒ Eukaryotic cell lines |
| ☐ | ☐ Palaeontology and archaeology |
| ☐ | ☐ Animals and other organisms |
| ☐ | ☐ Clinical data |
| ☐ | ☐ Dual use research of concern |
| ☐ | ☐ Plants |

## Methods

| n/a | Involved in the study |
|---|---|
| ☐ | ☒ ChIP-seq |
| ☐ | ☐ Flow cytometry |
| ☐ | ☐ MRI-based neuroimaging |

# Antibodies

| | |
|---|---|
| Antibodies used | All antibody catalog numbers are provided in antibodies list table. |
| Validation | Antibodies were validated through siRNA knockdown or CRISPR knockout genetic ablations as well as cellular localizations. Validation of all commercially available primary antibodies is provided on the manufacturers' websites. |

# Eukaryotic cell lines

Policy information about cell lines and Sex and Gender in Research

| | |
|---|---|
| Cell line source(s) | The human U2OS osteosarcoma cell line (ATCC, HTB-96), HeLa Kyoto cervical carcinoma cell line (CVCL_1922), primary immortalized retinal epithelial cell line hTERT-RPE1 (ATCC, CRL-4000), primary immortalized foreskin fibroblast BJ cells (ATCC, CRL-2522), were used in the study as parental cells and to generate their derivatives. |
| Authentication | The parental U2OS and hTERT-RPE1 cell lines were authenticated by STR profiling. No further authentication of cell line was performed. |
| Mycoplasma contamination | All cell line were routinely tested for mycoplasma contamination (MycoAlert, Lonza) and always found negative. |
| Commonly misidentified lines (See ICLAC register) | None |

# Palaeontology and Archaeology

| | |
|---|---|
| Specimen provenance | *Provide provenance information for specimens and describe permits that were obtained for the work (including the name of the issuing authority, the date of issue, and any identifying information). Permits should encompass collection and, where applicable, export.* |
| Specimen deposition | *Indicate where the specimens have been deposited to permit free access by other researchers.* |
| Dating methods | *If new dates are provided, describe how they were obtained (e.g. collection, storage, sample pretreatment and measurement), where they were obtained (i.e. lab name), the calibration program and the protocol for quality assurance OR state that no new dates are provided.* |

☐ Tick this box to confirm that the raw and calibrated dates are available in the paper or in Supplementary Information.

| | |
|---|---|
| Ethics oversight | *Identify the organization(s) that approved or provided guidance on the study protocol, OR state that no ethical approval or guidance was required and explain why not.* |

Note that full information on the approval of the study protocol must also be provided in the manuscript.

# Animals and other research organisms

Policy information about studies involving animals; ARRIVE guidelines recommended for reporting animal research, and Sex and Gender in Research

| | |
|---|---|
| Laboratory animals | *For laboratory animals, report species, strain and age OR state that the study did not involve laboratory animals.* |
| Wild animals | *Provide details on animals observed in or captured in the field; report species and age where possible. Describe how animals were caught and transported and what happened to captive animals after the study (if killed, explain why and describe method; if released, say where and when) OR state that the study did not involve wild animals.* |
| Reporting on sex | *Indicate if findings apply to only one sex; describe whether sex was considered in study design, methods used for assigning sex. Provide data disaggregated for sex where this information has been collected in the source data as appropriate; provide overall* |

numbers in this Reporting Summary. Please state if this information has not been collected.  Report sex-based analyses where performed, justify reasons for lack of sex-based analysis.

Field-collected samples | For laboratory work with field-collected samples, describe all relevant parameters such as housing, maintenance, temperature, photoperiod and end-of-experiment protocol OR state that the study did not involve samples collected from the field.

Ethics oversight | Identify the organization(s) that approved or provided guidance on the study protocol, OR state that no ethical approval or guidance was required and explain why not.

Note that full information on the approval of the study protocol must also be provided in the manuscript.

# Clinical data

Policy information about clinical studies

All manuscripts should comply with the ICMJE guidelines for publication of clinical research and a completed CONSORT checklist must be included with all submissions.

Clinical trial registration | Provide the trial registration number from ClinicalTrials.gov or an equivalent agency.

Study protocol | Note where the full trial protocol can be accessed OR if not available, explain why.

Data collection | Describe the settings and locales of data collection, noting the time periods of recruitment and data collection.

Outcomes | Describe how you pre-defined primary and secondary outcome measures and how you assessed these measures.

# Dual use research of concern

Policy information about dual use research of concern

## Hazards

Could the accidental, deliberate or reckless misuse of agents or technologies generated in the work, or the application of information presented in the manuscript, pose a threat to:

No | Yes

☐ ☐ Public health

☐ ☐ National security

☐ ☐ Crops and/or livestock

☐ ☐ Ecosystems

☐ ☐ Any other significant area

## Experiments of concern

Does the work involve any of these experiments of concern:

No | Yes

☐ ☐ Demonstrate how to render a vaccine ineffective

☐ ☐ Confer resistance to therapeutically useful antibiotics or antiviral agents

☐ ☐ Enhance the virulence of a pathogen or render a nonpathogen virulent

☐ ☐ Increase transmissibility of a pathogen

☐ ☐ Alter the host range of a pathogen

☐ ☐ Enable evasion of diagnostic/detection modalities

☐ ☐ Enable the weaponization of a biological agent or toxin

☐ ☐ Any other potentially harmful combination of experiments and agents

# Plants

| | |
|---|---|
| Seed stocks | *Report on the source of all seed stocks or other plant material used. If applicable, state the seed stock centre and catalogue number. If plant specimens were collected from the field, describe the collection location, date and sampling procedures.* |
| Novel plant genotypes | *Describe the methods by which all novel plant genotypes were produced. This includes those generated by transgenic approaches, gene editing, chemical/radiation-based mutagenesis and hybridization. For transgenic lines, describe the transformation method, the number of independent lines analyzed and the generation upon which experiments were performed. For gene-edited lines, describe the editor used, the endogenous sequence targeted for editing, the targeting guide RNA sequence (if applicable) and how the editor was applied.* |
| Authentication | *Describe any authentication procedures for each seed stock used or novel genotype generated. Describe any experiments used to assess the effect of a mutation and, where applicable, how potential secondary effects (e.g. second site T-DNA insertions, mosiacism, off-target gene editing) were examined.* |

# ChIP-seq

## Data deposition

☒ Confirm that both raw and final processed data have been deposited in a public database such as GEO.

☐ Confirm that you have deposited or provided access to graph files (e.g. BED files) for the called peaks.

| | |
|---|---|
| Data access links<br>*May remain private before publication.* | Publicly available datasets downloaded from ENCODE are presented in the study. |
| Files in database submission | ChIP-seq: E2F1 in K562 (ENCSR153DWR), ChIP-seq: E2F4 in K562 (ENCSR368GJN), ATAC-seq: K562 (ENCSR868FGK), ChIP-seq: E2F1 in HepG2 (ENCSR717ZZW), ChIP-seq: E2F4 in HepG2 (ENCSR924LSO), ATAC-seq: HepG2 (ENCSR291GJU), ChIP-seq: E2F1 in MCF7 (ENCSR000EWX), ChIP-seq: E2F4 in MCF7 (ENCSR505NMN), ATAC-seq: MCF7 (ENCSR422SUG), ChIP-seq: E2F1 in HeLa-S3 (ENCSR000EVJ), ChIP-seq: E2F4 in HeLa-S3 (ENCSR000EVL), DNase-seq: HeLa-S3 (ENCSR959ZXU) |
| Genome browser session<br>(e.g. UCSC) | *Provide a link to an anonymized genome browser session for "Initial submission" and "Revised version" documents only, to enable peer review. Write "no longer applicable" for "Final submission" documents.* |

## Methodology

| | |
|---|---|
| Replicates | *Describe the experimental replicates, specifying number, type and replicate agreement.* |
| Sequencing depth | *Describe the sequencing depth for each experiment, providing the total number of reads, uniquely mapped reads, length of reads and whether they were paired- or single-end.* |
| Antibodies | *Describe the antibodies used for the ChIP-seq experiments; as applicable, provide supplier name, catalog number, clone name, and lot number.* |
| Peak calling parameters | *Specify the command line program and parameters used for read mapping and peak calling, including the ChIP, control and index files used.* |
| Data quality | *Describe the methods used to ensure data quality in full detail, including how many peaks are at FDR 5% and above 5-fold enrichment.* |
| Software | *Describe the software used to collect and analyze the ChIP-seq data. For custom code that has been deposited into a community repository, provide accession details.* |

# Flow Cytometry

## Plots

Confirm that:

☐ The axis labels state the marker and fluorochrome used (e.g. CD4-FITC).

☐ The axis scales are clearly visible. Include numbers along axes only for bottom left plot of group (a 'group' is an analysis of identical markers).

☐ All plots are contour plots with outliers or pseudocolor plots.

☐ A numerical value for number of cells or percentage (with statistics) is provided.

## Methodology

| | |
|---|---|
| Sample preparation | *Describe the sample preparation, detailing the biological source of the cells and any tissue processing steps used.* |
| Instrument | *Identify the instrument used for data collection, specifying make and model number.* |

| Software | *Describe the software used to collect and analyze the flow cytometry data. For custom code that has been deposited into a community repository, provide accession details.* |
|---|---|
| Cell population abundance | *Describe the abundance of the relevant cell populations within post-sort fractions, providing details on the purity of the samples and how it was determined.* |
| Gating strategy | *Describe the gating strategy used for all relevant experiments, specifying the preliminary FSC/SSC gates of the starting cell population, indicating where boundaries between "positive" and "negative" staining cell populations are defined.* |

☐ Tick this box to confirm that a figure exemplifying the gating strategy is provided in the Supplementary Information.

# Magnetic resonance imaging

## Experimental design

| Design type | *Indicate task or resting state; event-related or block design.* |
|---|---|
| Design specifications | *Specify the number of blocks, trials or experimental units per session and/or subject, and specify the length of each trial or block (if trials are blocked) and interval between trials.* |
| Behavioral performance measures | *State number and/or type of variables recorded (e.g. correct button press, response time) and what statistics were used to establish that the subjects were performing the task as expected (e.g. mean, range, and/or standard deviation across subjects).* |

## Acquisition

| Imaging type(s) | *Specify: functional, structural, diffusion, perfusion.* |
|---|---|
| Field strength | *Specify in Tesla* |
| Sequence & imaging parameters | *Specify the pulse sequence type (gradient echo, spin echo, etc.), imaging type (EPI, spiral, etc.), field of view, matrix size, slice thickness, orientation and TE/TR/flip angle.* |
| Area of acquisition | *State whether a whole brain scan was used OR define the area of acquisition, describing how the region was determined.* |

Diffusion MRI     ☐ Used     ☐ Not used

## Preprocessing

| Preprocessing software | *Provide detail on software version and revision number and on specific parameters (model/functions, brain extraction, segmentation, smoothing kernel size, etc.).* |
|---|---|
| Normalization | *If data were normalized/standardized, describe the approach(es): specify linear or non-linear and define image types used for transformation OR indicate that data were not normalized and explain rationale for lack of normalization.* |
| Normalization template | *Describe the template used for normalization/transformation, specifying subject space or group standardized space (e.g. original Talairach, MNI305, ICBM152) OR indicate that the data were not normalized.* |
| Noise and artifact removal | *Describe your procedure(s) for artifact and structured noise removal, specifying motion parameters, tissue signals and physiological signals (heart rate, respiration).* |
| Volume censoring | *Define your software and/or method and criteria for volume censoring, and state the extent of such censoring.* |

## Statistical modeling & inference

| Model type and settings | *Specify type (mass univariate, multivariate, RSA, predictive, etc.) and describe essential details of the model at the first and second levels (e.g. fixed, random or mixed effects; drift or auto-correlation).* |
|---|---|
| Effect(s) tested | *Define precise effect in terms of the task or stimulus conditions instead of psychological concepts and indicate whether ANOVA or factorial designs were used.* |

Specify type of analysis:     ☐ Whole brain     ☐ ROI-based     ☐ Both

| Statistic type for inference | *Specify voxel-wise or cluster-wise and report all relevant parameters for cluster-wise methods.* |
|---|---|

(See Eklund et al. 2016)

| Correction | *Describe the type of correction and how it is obtained for multiple comparisons (e.g. FWE, FDR, permutation or Monte Carlo).* |
|---|---|

## Models & analysis

| n/a | Involved in the study |
|-----|----------------------|
| ☐ | ☐ Functional and/or effective connectivity |
| ☐ | ☐ Graph analysis |
| ☐ | ☐ Multivariate modeling or predictive analysis |

**Functional and/or effective connectivity**

*Report the measures of dependence used and the model details (e.g. Pearson correlation, partial correlation, mutual information).*

**Graph analysis**

*Report the dependent variable and connectivity measure, specifying weighted graph or binarized graph, subject- or group-level, and the global and/or node summaries used (e.g. clustering coefficient, efficiency, etc.).*

**Multivariate modeling and predictive analysis**

*Specify independent variables, features extraction and dimension reduction, model, training and evaluation metrics.*

