## [Peer Review File · Nature]

PAF15–PCNA exhaustion governs the strand-specific control of DNA replication

Corresponding Author: Dr Kumar Somyajit

Version 2:

Reviewer comments:

Referee #1

(Remarks to the Author)

A. Summary of the key results

In this manuscript, Chhetri and colleagues report that chromatin-bound PCNA becomes limiting when excessive replication origin activation is induced upon ATR inhibition, impairing lagging strand synthesis. Using a TurboID/MS-based approach, they identified PAF15/PCLAF as a novel dose-dependent regulator of PCNA binding to the lagging strand. This factor binds PCNA via a high-affinity PIP motif and inserts into the PCNA-DNA channel, preventing the premature unloading of PCNA by the ATAD5–RFC complex. This interaction ensures the proper processing of Okazaki fragments and prevents replication stress. The authors also report that the TIMELESS–CLASPIN complex excludes PAF15 from the leading strand, ensuring its lagging-strand specificity. Deletion of PAF15 selectively disrupts PCNA's interaction with lagging-strand factors such as DNA ligase I and FEN1, without affecting other replisome components. In addition, they show that PAF15 expression is tightly regulated by E2F4 and that PAF15 overexpression or its mislocalization to the leading strand impairs fork progression and triggers a replication catastrophe. Altogether, these findings identify PAF15 as a novel DNA replication factor that controls lagging-strand DNA synthesis in a rate-limiting manner. This study also explains why the loss of checkpoint control and excessive origin firing cause uncoordinated replication and genomic instability.

B. Originality and significance

It has been previously reported that ATR prevents replication catastrophe by preventing RPA exhaustion (PMID: 24267891). This study shows that in the absence of a functional replication checkpoint, chromatin-bound PCNA becomes rate-limiting, even before RPA exhaustion. This finding is consistent with two recent publications from the Diffley lab (PMID: 40578346; 40578347). In addition, the authors report the identification of PAF15 as the rate-limiting factor that controls this process and characterize its mode of action. This is undoubtedly a highly original and significant finding that greatly advances our understanding of eukaryotic DNA replication. Other notable findings include PAF15's interaction with the PCNA-DNA channel via an original mechanism that is blocked by the TIMELESS–CLASPIN complex. This explains how PCNA is selectively stabilized on the lagging strand. This reveals a new regulatory mechanism that links local strand-specific events with global replication control. The finding that PAF15 expression is finely tuned by E2F4 is also significant in the context of cancer biology. Finally, the absence of PAF15 in unicellular eukaryotes suggests that this factor evolved to handle the larger genomes and complex epigenetic regulation of multicellular organisms.

C. Data & methodology

The authors used a robust and well-structured methodological approach to dissect the role of PAF15 in DNA replication. Specifically, they combined quantitative imaging (QIBC), biochemical assays, chromatin immunoprecipitation, mutational analysis and structural studies to provide strong mechanistic evidence in support of their main conclusions. The data are generally of high quality and are presented in a clear and logical manner. The findings are consistent across multiple series

of experiments and orthogonal approaches. Overall, the study presents a compelling and methodologically sound case for a novel mechanism regulating genome replication.

D. Appropriate use of statistics and treatment of uncertainties

Quantitative analyses of PCNA dynamics, replication fork progression and origin activation are carefully executed and contextualized within existing literature. However, important issues need to be addressed regarding DNA fiber analyses (see below).

E. Conclusions: robustness, validity, reliability

The conclusions presented in the study appear to be robust, valid, and largely reliable. In general, the conclusions are not based on a single observation but rather converge from biochemical, genetic, and cellular data. The role of PAF15 on the lagging strand is confirmed through molecular interaction assays, mutational analyses, and functional replication studies. The exclusion of PAF15 from the leading strand via the TIMELESS–CLASPIN complex strengthens the specificity of their model. The dose-dependent effects of PAF15, along with its consequences on replication stress and fork stability, are thoroughly examined.

F. Suggested improvements:

1. Line 99: The statement that there is "clear evidence of new origin firing" is too strong. What evidence supports this claim? The fact that forks are slower is, at best, indirect evidence of increased origin firing. Unless direct evidence is provided, this statement should be toned down.

2. Line 242, 'Given the surprising finding that PAF15 is preferentially localized to the lagging strand': It is unclear why this finding is surprising. Considering the large excess of PCNA on the lagging strand compared to the leading strand, the preferential localization of PAF15 to the lagging strand is not surprising.

3. Line 294, 'PAF15-deficient cells accompanied 53BP1 nuclear body formation in G1': Although Fig. 3e and Ext. Data Fig. 12a show that G1 cells contain 53BP1 foci, the QIBC profiles do not show a preferential enrichment of 53BP1 foci in G1 PAF15KO cells than in control cells. Furthermore, the levels of 53BP1 foci appear higher in other cell cycle phases. Additional studies and statistical analyses are needed to support the authors statement that PAF15 deficiency leads to the accumulation of 53BP1 foci in G1.

4. It is interesting that PAF15-deficient cells depend on a PARP1-dependent non-canonical OkF maturation pathway for survival and this should be further explored. Was PAF15 identified as a hit in any genome-wide screens for PARPi sensitivity?

5. Overall, the DNA fiber analyses are convincing and show statistical differences. However, the number of biological replicates is not indicated ($n=1$?). To communicate not only statistical differences between samples but also experimental reproducibility, the authors should follow the recommendations of Lord et al. (PMID: 32346721) and display their data as superplots.

6. The lead author has previously reported that redox-sensitive mechanism of TIMELESS eviction from the replisome protects genome integrity by slowing replication fork speed (PMID: 29123070). However, the current study indicates that displacement of Timeless from the leading strand induces pathological relocalization of PAF15 to the leading strand. How do the authors reconcile these apparently conflicting results?

7. The authors' finding that high PAF15 levels correlate with poor prognosis is interesting. Since high levels of CLASPIN and TIMELESS are also associated with poor prognosis in cancer cells (PMID: 30796221) it would be interesting to investigate how the levels of PAF15, CLASPIN and TIMELESS vary relative to each other in cancer cells and whether these relative changes impact on cancer aggressiveness.

8. Line 444: The sentence 'When normalized for genomic content—reflecting the high origin frequency requirement—PAF15 levels were essentially equivalent across the tested cell lines' is not clear. If these levels are equivalent when normalized for genomic content, why would PAF15 levels reflect the high origin frequency requirement in cancer cells? The authors should also monitor PAF15 levels in these cell lines using Western blotting.

9. Line 505: Other conditions with stochastic origin firing than ATR inhibition have been described, such as RIF1 depletion. Is PAF15 important for cell viability in the absence of RIF1?

Minor issues

1. Nomenclature: The term "GINS1-4" is sometimes used informally to refer to the four subunits of the complex (Sld5, Psf1, Psf2, and Psf3). However, this name does not adhere to the correct formal nomenclature or the original Japanese name coined by its discoverer (GINS). Similarly, LIGASE1 does not adhere to nomenclature. The authors should use LIG1 for the gene and DNA ligase I for the protein.

2. Line 204: Title should read "... PCNA-associated factor ..."

3. Line 514: The term "lower eukaryotes" is not a formal classification. Authors should rather use 'unicellular eukaryotes' as opposed to multicellular, more complex eukaryotes.

4. Line 543: '... the lagging strand naturally forms single-stranded DNA loops...'. This reference to ssDNA loops is rather imprecise. Are the authors referring to the large loop formed to account for the fact that leading- and lagging-strand synthesis operate in opposite directions, or to the fact that DNA synthesis on the lagging strand is discontinuous and interrupted by multiple ssDNA gaps and nicks?

G. References:

Appropriate credit is given to previous work. However, reference 36 needs to be updated (PMID: 40578346; 40578347).

H. Clarity and context:

The abstract, introduction and conclusions are clear and appropriate for a high-impact journal.

Referee #2

(Remarks to the Author)

PAF15 is PIP-box containing PCNA-binding protein that has been previously shown to influence DNA replication, although the mechanism by which it does so is poorly understood. Prior work suggested that RPA is a critical limiting factor for high-fidelity replication (Toledo et al., Cell, 2013), but this work finds that PCNA becomes limiting on chromatin prior to RPA. The authors then go on to identify PAF15 as a critical regulator of PCNA, showing it controls this limiting effect of PCNA and that it becomes depleted when replication origin firing is elevated. The authors demonstrate that PAF15 stabilizes PCNA at the fork, controlling its interactions with ATAD5-RFC, the PCNA unloader, and with DNA polymerases. Strikingly, they also find that PAF15 shows surprising strand specificity in these effects, acting to regulate the lagging-strand pool of PCNA and its interacting factors, while leaving the leading-strand pool unaffected. They elucidate the mechanism of this protection, showing the leading-strand pool of PCNA is protected from PAF15 by CLASPIN and TIMELESS and that the interaction of PAF15 with PCNA is important for its functions. Finally, the work shows that PAF15 levels are tightly regulated and that this control is important, since excess PAF15 leads to loss of replisome integrity and problems on the leading strand, while a decrease in PAF15 disrupts lagging-strand Okazaki fragment maturation. The control of PAF15 expression in cells is also determined.

This manuscript provides an exciting, unexpected and detailed understanding of the functions of PAF15 while also providing surprising and important insights into the regulation of PCNA on the leading and lagging strands. The work also elegantly elucidates the mechanism by which PAF1 controls PCNA dynamics and interactions on the two strands, and how it is regulated itself. The mechanism of PCNA control by PAF1 is itself novel and unlike that of other factors which interact with and regulate PCNA. Most important, perhaps, the work defines a previously unrecognized, rate-limiting factor in eukaryotic DNA replication that acts in a strand-specific manner to control global DNA synthesis. The authors use a broad range of experimental approaches, including single-cell imaging, proteomics, structural modeling, and functional genetics, and their work reveals how local replisome composition is coupled to replication dynamics. Overall, I think manuscript makes an important conceptual advance that is appropriate for publication in Nature. With a few exceptions, the data also largely support the conclusions made.

Major Comments

1. What is the normal stoichiometric relationship between PAF15 and PCNA?
2. PAF15 overexpression phenotype: PAF15 transient overexpression is toxic and slows down nascent DNA synthesis rate. The authors predict that this might be due to PAF15 blocking PCNA interactions with POLE1 on the leading strand. When PAF15 is overexpressed, does it suppress PCNA-POLE1 interaction by PLA?
3. Mechanism of PAF15–POLE1 interaction upon TIMELESS loss: In Fi. 5d, TIMELESS loss increases PAF15–POLE1 PLA signal, presumably because TIMELESS shields leading-strand PCNA from PAF15. However, the authors also suggest that PAF15 and POLE1 bind PCNA mutually exclusively. How then does TIMELESS loss result in increased PAF15–POLE1 interaction?
4. Limitation of universality in PAF15 regulation: While data in Fig. 6 and Ext. Data Figs 19–20 support E2F4-mediated repression of PAF15, these data are somewhat distracting and actually rather peripherally related to the overall manuscript (which is already dense). I would suggest these data actually be removed.

Technical/Minor Comments

1. Ext. Data Fig. 1c: It is unclear how the fork speed is measured. (Is it derived from the measurement of IdU track length of progressing forks?)
2. Are Fig. 2a and Ext. Data Fig. 6f scatter plot identical?

3. Lines 235-236: In Ext. Data Fig. 9e, the higher PCNA signal in the western blot could be due to a more efficient PCNA antibody compared to PAF15 antibody. The reviewer thus does not agree with the authors' statement that "immunoblotting across various cell types, which consistently show higher chromatin levels of PCNA than PAF15" based on this data.

4. Lines 280-283: Lacks figure call out (Ext. Data Fig. 11f?). In addition, why is PCNA chromatin loading increased in PAF15 overexpressing cells following siPAF15 (middle panel)?

5. Fig. 3c: DNA spreading, in which the leading- and lagging-strand nascent DNA collapse during the process, is used following S1 nuclease digestion. S1-nuclease sensitivity in PAF15-KO cells therefore indicates not only lagging-strand processing issue, but also indicates S1-sensitivity on the leading strand as well. How would the authors reconcile this with the hypothesis that PAF15 functions exclusively on lagging strand? The authors could better address this with combing (Meroni et al., PMID: 38315097), or alternatively change the text slightly.

Referee #3

(Remarks to the Author)

This manuscript investigates whether the checkpoint pathways, particularly those mediated by ATR, control origin firing through the regulation of rate-limiting replisome components whose molecular identity remains incompletely defined. To address this, the authors examined how ATR inhibition affects the assembly and dynamics of replication factors, focusing primarily on PCNA loading and lagging-strand processing. A key methodological approach used throughout the study is quantitative image-based cytometry (QIBC), which enabled high-content and automated single-cell analysis to capture the detailed dynamics of replication stress responses. They demonstrated that ATR inhibitor (ATRi) treatment enhanced origin firing and activated CDK1/2, leading to increased chromatin association of replisome components such as Cdc45. However, PCNA levels on chromatin did not exhibit significant changes under these conditions. These findings suggest that when the ATR-mediated checkpoint is turned off, further loading of PCNA onto chromatin does not occur, despite increased origin firing and replisome assembly. Therefore, lagging-strand synthesis is compromised, resulting in defects in Okazaki fragment ligation and accumulation of DNA damage.

Notably, the authors identified PAF15 as a dosage-sensitive regulator of PCNA. Under unperturbed replication conditions, nearly the entire soluble pool of PAF15 was bound to chromatin, where it stabilizes the association of PCNA with DNA and ensures proper lagging-strand synthesis. Part of this stabilization appears to be mediated by PAF15, opposing the action of the PCNA unloader ATAD5. However, when ATRi deregulated origin firing, PAF15 could not accommodate additional PCNA loading onto chromatin. Consequently, lagging-strand synthesis was disrupted, leading to incomplete Okazaki fragment maturation. Consistent with this, PAF15-deficient cells showed reduced levels of chromatin-bound PCNA and accumulation of unligated Okazaki fragments, accompanied by markers of replication stress. Interestingly, both overexpression of PAF15 and its mislocalization to the leading strand were found to impede replication fork progression and induce cell death.

These findings suggest that the role of the DNA replication checkpoint in suppressing excess origin firing is crucial to prevent exhaustion of the PAF15–PCNA axis on the lagging strand. Moreover, they indicate that a strand-specific, rate-limiting mechanism is closely linked to the global dynamics of chromosome replication.

Overall, the findings provide intriguing insights into how PCNA dynamics and lagging-strand processing are regulated under replication stress and reveal that PAF15 acts as a strand-specific, rate-limiting factor whose availability is crucial for maintaining genomic stability. However, several aspects of the study warrant further clarification and validation to strengthen the mechanistic conclusions and fully establish the broader significance of these observations.

Specific points:

1. The authors showed that siRNA-mediated ATAD5 depletion rescued replication catastrophe and PCNA chromatin loading defects induced by HU+ATRi or PAF15 knockout (Fig. 1g,h; Ext. Data Fig. 14a). However, as ATAD5 suppression is known to cause delays in replication progression (Lee et al., *J. Cell Biol.*, 2013), the authors should clarify whether Okazaki fragment maturation was genuinely restored under these conditions or whether the observed rescue merely reflects slowed replication dynamics.

2. It has been reported that the chromatin loading of PAF15 is regulated not only by its interaction with PCNA but also by its ubiquitination status. However, in the current study, most analyses and discussions regarding chromatin-bound PAF15 appear to focus exclusively on the non-ubiquitinated form detected at approximately 15 kDa. To fully interpret their findings, the authors should examine and present data on the ubiquitinated forms of PAF15 in their chromatin fractionation assays and discuss how these species might influence chromatin association and the function of PAF15, particularly in stabilizing PCNA under the tested conditions. Moreover, the absence of deubiquitinase (DUB) inhibitors in the lysis buffers and the use of RIPA buffer containing 150 mM NaCl may compromise detection of chromatin-bound ubiquitinated PAF15. The authors should consider repeating these experiments under milder extraction conditions and with appropriate DUB inhibitors to ensure accurate assessment of both ubiquitinated and non-ubiquitinated forms of PAF15.

3. The authors observed that DNMT1 levels on chromatin remain unchanged under conditions of excess origin firing. However, DNMT1 is not exclusively a lagging-strand-associated factor, and it would be important for the authors to clarify why DNMT1 does not similarly increase. This raises questions about potential strand-specific regulation of DNMT1 loading or broader changes in replication dynamics under these conditions. Furthermore, given that PAF15 has been implicated in

pathways involved in DNA methylation maintenance, its loss might be expected to result in global or locus-specific DNA hypomethylation. It would be informative to examine how depletion or inhibition of key DNA methylation regulators, such as UHRF1 or DNMT1, affects PCNA chromatin binding.

4. The authors showed in Fig. 3g and Ext. Data Fig. 12d that the combination of PAF15 knockout and PARP inhibition led to increased formation of micronuclei and γ H2AX accumulation, indicating elevated genomic instability. The authors should clarify whether unligated Okazaki fragments indeed accumulate in the context of combined PAF15 loss and PARP inhibition.

5. In Fig. 5, the authors showed that suppression of TIMELESS led to increased interaction between PAF15 and DNA polymerase ϵ (Pol ϵ), which they interpreted as evidence that PAF15 is aberrantly recruited to the leading strand. However, it remains unclear whether an alternative possibility has been excluded—namely, that Pol ϵ itself might instead relocate to the lagging strand under these conditions. The authors should clarify how they have ruled out this possibility. Moreover, it would be essential to validate their interpretation that PAF15 is acting on the leading strand by employing complementary experimental approaches.

6. In Fig. 6, the authors showed that E2F4 transcriptionally regulates PAF15 and that double knockdown of PAF15 and E2F4 rescued the phenotype associated with PAF15 depletion. However, given that E2F4 is known to regulate numerous genes involved in DNA replication and cell cycle control, it is difficult to conclude that the observed rescue effect is mediated solely through changes in PAF15 expression. The authors should consider whether the effects of E2F4 knockdown might also involve alterations in other replication-associated pathways or factors. Such broader effects could potentially confound the interpretation that PAF15 is the primary target responsible for the observed phenotypes.

Minor points:

1. In the current study, the authors tagged PAF15 with an N-terminal Myc-FLAG epitope. However, the N-terminus of PAF15 contains motifs critical for recognition by UHRF1, which mediates its ubiquitination. Adding an N-terminal tag may interfere with UHRF1 binding and consequently alter the ubiquitination status and chromatin association of PAF15. The authors should consider performing key experiments with a C-terminally tagged version of PAF15 to ensure that the observed interactions and functional effects are not artifacts resulting from N-terminal tagging.

2. Previous reports have shown that methylation of LIG1 promotes the recruitment of UHRF1, which acts as the E3 ubiquitin ligase for PAF15. However, in the current study, the authors showed that a LIG1- Δ TARK mutant did not affect PAF15 ubiquitination, which seems inconsistent with this established pathway. Although the authors do not focus on this aspect, it would be helpful if they could comment on how this observation might be reconciled with previous findings.

3. In Ext. Data Fig. 9b, the authors showed that mutation of the KEN box in PAF15 led to increased protein stability but did not substantially enhance chromatin binding or recapitulate the phenotypes seen with PAF15 overexpression. It would be helpful for the authors to discuss why stabilization of PAF15 through KEN box mutation did not produce the same functional effects as overexpression, and whether this suggests that regulation of PAF15 activity involves mechanisms beyond simple protein abundance.

4. The authors should cite Ext. Data Fig. 11f in the main text (lines 280–283) where the data on PAF15 overexpression and increased PCNA/LIG1 chromatin binding are discussed.

5. Ext. Data Figs 14c and 15c,d refer to “ Δ N PAF15 (2–61),” but this mutant is not explained in the main text and may be mislabeled. The authors should clarify its identity and functional significance.

Referee #4

(Remarks to the Author)

I co-reviewed this manuscript with one of the reviewers who provided the listed reports.

Version 3:

Reviewer comments:

Referee #1

(Remarks to the Author)

The authors have adequately addressed all the concerns I raised and have substantially strengthened their manuscript. Notably, they provide compelling new insights regarding points #7 (the roles of PAF15 and CLASPIN in cancer) and #9 (the interplay between PAF15 and RIF1). While these findings are intriguing, I agree with the authors that they extend beyond the scope of the current study and are best explored in a separate work. In my view, the manuscript is now ready for publication, but references 15, 55, 62 still need to be completed.

Referee #2

(Remarks to the Author)

I have gone back over the manuscript and the authors' revisions. They have addressed all of my concerns appropriately and in some cases extensively and I am very happy with the revised work.

Referee #3

(Remarks to the Author)

Most of my major concerns have been addressed, and the revised manuscript now provides sufficient support for the main conclusions. However, some points still require clarification. In the revised version, the authors demonstrated that ATAD5 depletion largely rescues the replication defects observed upon PAF15 loss. Nevertheless, Lee et al. (J. Cell Biol., 2013) reported that ATAD5 depletion promotes PCNA loading but also leads to the formation of inactive replication factories and delayed replication progression. It therefore remains unclear whether the replication factories restored by ATAD5 depletion in this study are indeed functionally active. Moreover, if PAF15 indeed prevents premature, ATAD5-dependent PCNA unloading as proposed, the PAF15–PCNA complex should engage with LIG1, FEN1, and polymerases to promote lagging-strand synthesis. Although the authors suggest in Ext. Data Fig. 14 that these interactions are structurally compatible with AlphaFold models, it would further strengthen the study if they provided experimental evidence, such as co-immunoprecipitation, to confirm their existence.

Referee #4

(Remarks to the Author)

I co-reviewed this manuscript with one of the reviewers who provided the listed reports.

Point-by-point response the Reviewers:

We sincerely thank all four reviewers and the editor for their enthusiasm, constructive feedback, and excellent guidance in strengthening our manuscript. We found their suggestions highly insightful and have carefully addressed both technical and conceptual concerns in the revised version.

We are confident that these revisions resolve key points raised, and importantly, the amendments and the opportunity to revise have reinforced our key conclusions, namely:

- PAF15 is strictly confined to lagging-strands under physiological conditions, as also revealed by SCAR-Seq and OK-seq, a strand-specific chromatin-associated nascent DNA sequencing approach that maps protein associations with leading versus lagging strand synthesis (**Fig. 2f-h, Ext. Data Fig. 8b,c**).
- Additional co-immunoprecipitation experiment demonstrated that loss of TIMELESS/CLASPIN or rapid overexpression of PAF15 forces mislocalization of PAF15 to the leading strand, accompanied by a dramatic replication fork slowdown revealed by DNA fiber assays (**Fig. 5f, g**). Moreover, to further assess the impact of rapid PAF15 overexpression, we performed chromatin extraction followed by immunoblotting (**Ext. Data Fig. 13e**). The results showed that, while POLE1 remained stably bound to chromatin, lagging-strand polymerase POLD1 was markedly reduced despite enhanced PCNA retention. This likely reflects excessive PAF15 occupancy displacing PCNA-interacting factors that, unlike POLE, rely on PCNA for stable association with DNA (**Ext. Data Fig. 13e**). These results are now thoroughly discussed in the revised manuscript.
- We show that the recruitment of PCNA and PAF15 *per se* to replicating chromatin in human cells occurs largely independently of UHRF1-mediated double monoubiquitination of PAF15 (**Ext. Data Fig. 8e, h**). However, consistent with previous findings (Nishiyama et al., *Nat. Commun.* 2020; PMID: 32144273), UHRF1 and PAF15 monoubiquitination remain critical for the subsequent loading and stable establishment of DNMT1 on chromatin (**Ext. Data Fig. 8g-h**).
- We now provide further evidence for the dynamic interplay between PAF15-mediated PCNA stabilization and ATAD5-mediated PCNA unloading in functionally regulating lagging-strand maturation and genome integrity (**Fig. 4f, g; Ext. Data Fig. 12b, c**). Along the same lines, we now provide additional evidence showing that the combined loss of PAF15 and non-canonical Okazaki fragment maturation upon PARP inhibition promotes the accumulation of daughter-strand gaps and fuels post-replicative repair by RAD51 (**Fig. 3f, g**).
- We have directly determined the relative stoichiometry of PCNA and PAF15 in whole-cell lysates and chromatin-bound fractions of U2OS cells using Orbitrap Astral mass spectrometry-based deep proteomics (**Ext. Data Fig. 8d**). Furthermore, analysis of transcriptomic data from 85 cell lines revealed a consistently high PCNA-to-PAF15 ratio at the RNA level, underscoring the natural paucity of PAF15 (**Ext. Data Fig. 16f**).
- We provide evidence of excessive origin firing upon ATRi and WEEi treatment (**Ext. Data Fig. 1c**), along with additional controls and amendments that reinforce our conclusions throughout the manuscript. Also in line with the reviewers' suggestions, particularly from reviewers 2 and 3—we have removed redundant datasets and toned down the interpretations related to the E2F4 data.

All revisions are detailed below, addressed point by point in blue text, and the corresponding amendments in the revised manuscript are highlighted in red.

Referee #1:

A. Summary of the key results

In this manuscript, Chhetri and colleagues report that chromatin-bound PCNA becomes limiting when excessive replication origin activation is induced upon ATR inhibition, impairing lagging strand synthesis. Using a TurboID/MS-based approach, they identified PAF15/PCLAF as a novel dose-dependent regulator of PCNA binding to the lagging strand. This factor binds PCNA via a high-affinity PIP motif and inserts into the PCNA-DNA channel, preventing the premature unloading of PCNA by the ATAD5–RFC complex. This interaction ensures the proper processing of Okazaki fragments and prevents replication stress. The authors also report that the TIMELESS–CLASPIN complex excludes PAF15 from the leading strand, ensuring its lagging-strand specificity. Deletion of PAF15 selectively disrupts PCNA's interaction with lagging-strand factors such as DNA ligase I and FEN1, without affecting other replisome components. In addition, they show that PAF15 expression is tightly regulated by E2F4 and that PAF15 overexpression or its mislocalization to the leading strand impairs fork progression and triggers a replication catastrophe. Altogether, these findings identify PAF15 as a novel DNA replication factor that controls lagging-strand DNA synthesis in a rate-limiting manner. This study also explains why the loss of checkpoint control and excessive origin firing cause uncoordinated replication and genomic instability.

B. Originality and significance

It has been previously reported that ATR prevents replication catastrophe by preventing RPA exhaustion (PMID: 24267891). This study shows that in the absence of a functional replication checkpoint, chromatin-bound PCNA becomes rate-limiting, even before RPA exhaustion. This finding is consistent with two recent publications from the Diffley lab (PMID: 40578346; 40578347). In addition, the authors report the identification of PAF15 as the rate-limiting factor that controls this process and characterize its mode of action. This is undoubtedly a highly original and significant finding that greatly advances our understanding of eukaryotic DNA replication. Other notable findings include PAF15's interaction with the PCNA-DNA channel via an original mechanism that is blocked by the TIMELESS–CLASPIN complex. This explains how PCNA is selectively stabilized on the lagging strand. This reveals a new regulatory mechanism that links local strand-specific events with global replication control. The finding that PAF15 expression is finely tuned by E2F4 is also significant in the context of cancer biology. Finally, the absence of PAF15 in unicellular eukaryotes suggests that this factor evolved to handle the larger genomes and complex epigenetic regulation of multicellular organisms.

C. Data & methodology

The authors used a robust and well-structured methodological approach to dissect the role of PAF15 in DNA replication. Specifically, they combined quantitative imaging (QIBC), biochemical assays, chromatin immunoprecipitation, mutational analysis and structural studies to provide strong mechanistic evidence in support of their main conclusions. The data are generally of high quality and are presented in a clear and logical manner. The findings are consistent across multiple series of experiments and orthogonal approaches. Overall, the study presents a compelling and methodologically sound case for a novel mechanism regulating genome replication.

D. Appropriate use of statistics and treatment of uncertainties

Quantitative analyses of PCNA dynamics, replication fork progression and origin activation

are carefully executed and contextualized within existing literature. However, important issues need to be addressed regarding DNA fiber analyses (see below).

E. Conclusions: robustness, validity, reliability

The conclusions presented in the study appear to be robust, valid, and largely reliable. In general, the conclusions are not based on a single observation but rather converge from biochemical, genetic, and cellular data. The role of PAF15 on the lagging strand is confirmed through molecular interaction assays, mutational analyses, and functional replication studies. The exclusion of PAF15 from the leading strand via the TIMELESS–CLASPIN complex strengthens the specificity of their model. The dose-dependent effects of PAF15, along with its consequences on replication stress and fork stability, are thoroughly examined.

We are sincerely grateful and delighted that the reviewer appreciated the originality of our study and acknowledged the high quality and robustness of our data. We greatly value their constructive suggestions, which have enabled us to further refine and strengthen the manuscript.

F. Suggested improvements:

1. Line 99: The statement that there is "clear evidence of new origin firing" is too strong. What evidence supports this claim? The fact that forks are slower is, at best, indirect evidence of increased origin firing. Unless direct evidence is provided, this statement should be toned down.

Thank you for the comment. We fully agree with the reviewer. To address this point, we measured local origin density using the DNA fiber assay protocol established in our previous studies (Ercilla et al., *Cell Reports* 2020, PMID: 32075739; Sedlackova et al., *Nature* 2020, PMID: 33087936). The new data presented below provide additional evidence of increased origin density when replication checkpoint control is compromised (**Rebuttal Fig. 1**). These results are now included in **Ext. Data Fig. 1c** and, together with our QIBC-based evidence of new replisome assembly under these conditions, support the presence of newly formed replisome sites. We have also cited additional relevant studies in the revised manuscript to further strengthen these conclusions.

Following the reviewer's suggestion, we have also toned down the statement from '*...despite the presence of elevated CDK activity and clear evidence of new origin firing*' to '*...despite elevated CDK activity and abrupt activation of new origins*'.

2. Line 242, 'Given the surprising finding that PAF15 is preferentially localized to the lagging strand': It is unclear why this finding is surprising. Considering the large excess of PCNA on the lagging strand compared to the leading strand, the preferential localization of PAF15 to the lagging strand is not surprising.

We have now modified the statement by removing the word 'surprising'.

3. Line 294, 'PAF15-deficient cells accompanied 53BP1 nuclear body formation in G1': Although Fig. 3e and Ext. Data Fig. 12a show that G1 cells contain 53BP1 foci, the QIBC profiles do not show a preferential enrichment of 53BP1 foci in G1 PAF15KO cells than in control cells. Furthermore, the levels of 53BP1 foci appear higher in other cell cycle phases. Additional studies and statistical analyses are needed to support the authors statement that PAF15 deficiency leads to the accumulation of 53BP1 foci in G1.

Thank you once again for this important comment. As noted, the enrichment of 53BP1 foci in PAF15-deficient cells is evident across all cell cycle phases.

First, to examine this in more detail, we performed time-course PAF15 depletion using siRNA. We found that 53BP1 foci appear in the Cyclin A–negative G1 population within 24 hours and remain only partially resolved in the Cyclin A–positive S–G2 population, with the effect becoming progressively more pronounced over time (**Rebuttal Fig. 2**).

Rebuttal Fig. 2. Left, QIBC analysis of the formation of 53BP1 foci in day dependent depletion of PAF15 in HeLa Kyoto cells. $n > 10,000$ cells per condition. Quantification of 53BP1 foci CycA2 negative (G1) and CycA2 positive (S/G2) cells derived from data in left.

Time-lapse imaging further confirmed that PAF15-deficient cells accumulate more 53BP1 foci during G1, which persist into S–G2, resulting in a sustained population of 53BP1-positive cells throughout the cell cycle (**Rebuttal Fig. 3**).

Rebuttal Fig. 3. Representative time-lapse images of U2OS cells stably expressing RFP-53BP1 and GFP-PCNA treated with control or PAF15 siRNAs 24 h before time-lapse imaging. Arrows indicate 53BP1-NBs/foci.

For simplicity and clarity, we now provide 53BP1 foci measurements stratified by cell cycle phase from representative QIBC plots across the cell lines (**Fig. 3d**; **Ext. Data Fig. 10g, h**).

We have also revised the terminology from “53BP1 nuclear bodies” to “53BP1 foci” to more broadly reflect markers of under-replicated DNA and genome instability.

4. It is interesting that PAF15-deficient cells depend on a PARP1-dependent non-canonical OkF maturation pathway for survival and this should be further explored. Was PAF15 identified as a hit in any genome-wide screens for PARPi sensitivity?

Indeed, in the genome-wide CRISPR screens reported by Olivieri et al. *Cell* 2020 (PMID: 32649862, Durocher lab), PAF15 (*KIAA0101*) was identified as a sensitizer to PARP inhibition, with a \log_2 score of -1.58 , corresponding to an approximately 3-fold reduction in cell survival.

Based on the suggestions from this reviewer and Reviewer 3, we have extended our analysis of the impact of PAF15 loss and PARP1 inhibition by assessing S1-nuclease sensitivity of nascent DNA and RAD51 foci formation in S phase, providing further insight into how loss of canonical and non-canonical pathways enforces post-replication daughter-strand gaps.

These new results have been incorporated into the revised manuscript (**Fig. 3 f, g**).

5. Overall, the DNA fiber analyses are convincing and show statistical differences. However, the number of biological replicates is not indicated ($n=1?$). To communicate not only statistical differences between samples but also experimental reproducibility, the authors should follow the recommendations of Lord et al. (PMID: 32346721) and display their data as superplots.

We thank the reviewer for this valuable comment. In the revised manuscript, we now clearly indicate the number of biological replicates in each figure legend and upload source data.

Regarding the DNA fiber experiments, we would like to emphasize that all major findings have been thoroughly reproduced and consistently validated across multiple experimental contexts already presented in the manuscript. Specifically, the spontaneous acceleration of fork speed in PAF15-deficient cells is reproducibly observed in different cell types and through independent genetic ablation strategies (**Ext. Data Fig. 10d–f; Fig. 3c, f; Fig. 4g**). Likewise, the increased S1 nuclease sensitivity of nascent DNA in PAF15 knockout cells is demonstrated in **Fig. 3c, f and Fig. 4g**, while fork deceleration following acute PAF15 overexpression is shown in **Fig. 5c, f**.

Furthermore, to directly address the reviewer's concern, we provide a representative example of side-by-side fork speed measurements in PAF15-deficient cells from two additional independent biological replicates (**Rebuttal Fig. 4**). Rather than reformatting all violin plots presented in the manuscript, we have clearly indicated the biological replicates in the corresponding figure legends. However, if deemed necessary, we are happy to convert these DNA fiber datasets into a superplot format.

Rebuttal Fig. 4. Replication speed of parental and *PAF15* KO U2OS cells. $n = 200$ fibers per condition. P values were determined by one-way ANOVA with Tukey's test.

6. The lead author has previously reported that redox-sensitive mechanism of TIMELESS eviction from the replisome protects genome integrity by slowing replication fork speed (PMID: 29123070). However, the current study indicates that displacement of Timeless from the leading strand induces pathological relocalization of PAF15 to the leading strand. How do the authors reconcile these apparently conflicting results?

This is a very exciting point raised by the reviewer. Indeed, as shown in the data below, under low HU conditions—where TIMELESS is known to dissociate from the fork (Somyajit et al., *Science*, 2017; PMID: 29123070)—PAF15 gains access to the leading-strand DNA polymerase. This observation aligns with our data showing that depletion of TIMELESS similarly forces an increased interaction between POLE and PAF15 (**Fig. 5e**). In contrast, loss of PRDX2, which prevents TIMELESS dissociation from the replisome, does not result in such pronounced PAF15 association with POLE1 (**Rebuttal Fig. 5**). Supporting this further, fork slowdown upon low-dose HU treatment is incomplete in PAF15 knockout cells, yet not as pronounced as in PRDX2-depleted cells (**Rebuttal Fig. 6**).

While these findings are highly intriguing, we consider them to extend beyond the primary focus of the current study, which centers on identifying replisome-based rate-limiting mechanisms during unperturbed DNA replication. In ongoing work from our laboratory (Petersen et al., manuscript in preparation, 2025–2026), we are extending our previous work of a redox-sensitive mechanism of fork speed regulation mediated by the PRDX2–TIMELESS axis. Specifically, we continue to understand how aberrant localization of PAF15 to the leading strand may affect the conditions under which TIMELESS dissociates from the replisome.

As suggested by the reviewer, we have now included a brief statement in the revised manuscript to conceptually link these new observations to our previous work on TIMELESS.

Rebuttal Fig. 5. (Left) Schematic of experimental flow. Representative images (top) and quantification of the PLA foci sum-intensity using QIBC (bottom) of PLA antibody pair PAF15-POLE1. >10,000 cells per condition. Scale bar, 10 μm.

Rebuttal Fig. 6. (Left) schematic DNA-fiber labelling. Replication speed of parental and PAF15 KO cells treated with PRDX2 siRNA. n= 200 fibers per condition. P values are calculated by one-way ANOVA with Tukey's test.

7. The authors' finding that high PAF15 levels correlate with poor prognosis is interesting. Since high levels of CLASPIN and TIMELESS are also associated with poor prognosis in cancer cells (PMID: 30796221) it would be interesting to investigate how the levels of PAF15, CLASPIN and TIMELESS vary relative to each other in cancer cells and whether these relative changes impact on cancer aggressiveness.

In line with the reviewer's suggestion, we examined the relative expression patterns of PAF15, CLASPIN, and TIMELESS across pan-cancer datasets. Consistent with previous report (Bianco et al., *Nat Commun* 2019; PMID: 30796221), both CLSPN and TIMELESS are significantly upregulated in tumors compared to normal tissues ($p < 0.0001$), and their elevated expression correlates with poor overall survival across multiple cancer types ($p < 0.0001$ for both).

Importantly, PAF15 expression is strongly correlated with both CLSPN ($R = 0.86$) and TIMELESS ($R = 0.85$), indicating a coordinated upregulation of these fork-associated factors in cancer (**Rebuttal Fig. 7**).

Rebuttal Fig. 7. CLSPN and TIMELESS mRNA level correlation across normal and cancer cell types. (left) Expression of CLSPN and TIMELESS in normal and tumor tissue ($n = 8,167$) and tumor tissue ($n = 9,471$) from 33 cancer types. Data was extracted from the TCGA TARGET GTEx database employing Mann-Whitney U test. (middle) Kaplan-Meier time-to-event analysis with log-rank p value for PanCancer ATLAS mortality. Cut-off values were chosen based on the most significant values from Cox-regression on all values from lower to upper quartiles. Scatterplot, representing (Right) correlation of PAF15 with CLSPN (TOP) or TIMELESS (bottom) expression in cancer tumors.

Very interestingly and equally complex to interpret—within the subset of tumors exhibiting high PAF15 expression, only CLASPIN retained significant prognostic value ($p < 0.0001$), whereas TIMELESS did not ($p = 0.27$) (**Rebuttal Fig. 8**). These observations suggests that, beyond absolute expression levels, the relative balance among PAF15, CLASPIN, and TIMELESS may critically influence cancer aggressiveness.

Rebuttal Fig. 8. Kaplan-Meier time-to-event analysis with log-rank p value for PanCancer ATLAS mortality.

We believe that this warrants deeper mechanistic investigation, as it may reflect a functional coupling between PCNA–PAF15 and the TIMELESS–CLASPIN complex in regulating replication fork stability and progression under oncogenic or metabolic stress. Therefore, we plan to further develop and integrate these findings into our ongoing unpublished work (discussed in the response above; Petersen et al., manuscript in preparation, 2025–2026), which focuses on the redox-sensitive regulation of the TIMELESS–CLASPIN complex.

8. Line 444: The sentence ‘When normalized for genomic content—reflecting the high origin frequency requirement— PAF15 levels were essentially equivalent across the tested cell lines’ is not clear. If these levels are equivalent when normalized for genomic content, why would PAF15 levels reflect the high origin frequency requirement in cancer cells?

We agree with the reviewer that the statement directly correlating genomic content with origin frequency requirement is inappropriate, and we have therefore removed the phrasing.

For the second part of the question, we have now performed Western blotting on these four cell lines (using whole-cell extracts from equal cell numbers) to simultaneously monitor the total levels of PAF15 and PCNA. The new data are included in **Ext. Figure 16b**.

9. Line 505: Other conditions with stochastic origin firing than ATR inhibition have been described, such as RIF1 depletion. Is PAF15 important for cell viability in the absence of RIF1?

Thank you for the suggestion.

Based on the reviewer’s comment, we first depleted *RIF1* to test whether RIF1 loss alters the global levels of PAF15–PCNA chromatin loading.

Consistent with the rate-limiting nature of PAF15–PCNA dynamics during unperturbed replication, we observe very subtle alteration in the chromatin association of either PCNA or PAF15 upon RIF1 depletion, in both ATR inhibitor–treated and untreated cells (**Rebuttal Fig. 9**).

Rebuttal Fig. 9. a, Top, Representative images of PAF15 and RIF1 in U2OS parental and *PAF15* KO cells depleted for *RIF1* and treated with ATRi. **b**, QIBC analysis of chromatin bound PAF15 in U2OS parental and *PAF15* KO cells in conditions indicated in the figure. Thin dotted lines indicate average values. Thick pink dotted line on QIBC plot indicate maximum approximate values, **c**, correlation QIBC plot for chromatin bound PCNA and PAF15.

[Redaction]

[Redaction]

For correctness of the statement, we have now modified the term '**stochastic**' to '**normal**' origin firing in the revised manuscript.

Minor issues

1. Nomenclature: The term "GINS1-4" is sometimes used informally to refer to the four subunits of the complex (Sid5, Psf1, Psf2, and Psf3). However, this name does not adhere to the correct formal nomenclature or the original Japanese name coined by its discoverer (GINS). Similarly, LIGASE1 does not adhere to nomenclature. The authors should use LIG1 for the gene and DNA ligase I for the protein.

The gene nomenclature has been rectified and is now used consistently throughout the revised manuscript.

2. Line 204: Title should read "... PCNA-associated factor ..."

PAF15 has now been expanded to 'PCNA-associated factor 15' in the results section heading.

3. Line 514: The term "lower eukaryotes" is not a formal classification. Authors should rather use 'unicellular eukaryotes' as opposed to multicellular, more complex eukaryotes.

Thank you for the suggestion — this has been updated to 'unicellular eukaryotes' when referring to yeast.

4. Line 543: '... the lagging strand naturally forms single-stranded DNA loops...'. This reference to ssDNA loops is rather imprecise. Are the authors referring to the large loop formed to account for the fact that leading- and lagging-strand synthesis operate in opposite directions, or to the fact that DNA synthesis on the lagging strand is discontinuous and interrupted by multiple ssDNA gaps and nicks?

Apologies for the earlier lack of clarity in distinguishing between the natural DNA loops formed at the replisome junction during leading- and lagging-strand synthesis and postreplicative single-stranded DNA (ssDNA) gaps. The statement has now been revised to emphasize the transient DNA loops that naturally arise as the two strands are synthesized in opposite directions and the associated ssDNA adjacent to the primer–template junction—a

well-established determinant of replication checkpoint activation (Byun et al., *Genes Dev.*, 2005; PMID: 15833911). We further propose that, under conditions of PAF15 depletion or defective lagging-strand maturation, these physiological DNA loops and ssDNA regions may persist or expand into daughter-strand gaps, thereby amplifying checkpoint-activating signals and engaging S-phase surveillance pathways.

G. References:

Appropriate credit is given to previous work. However, reference 36 needs to be updated (PMID: 40578346; 40578347).

Both of these new papers from Diffley's lab have been updated and referenced in the revised manuscript.

H. Clarity and context:

The abstract, introduction and conclusions are clear and appropriate for a high-impact journal.

Thank you once again for the positive comment.

Referee #2:

PAF15 is PIP-box containing PCNA-binding protein that has been previously shown to influence DNA replication, although the mechanism by which it does so is poorly understood. Prior work suggested that RPA is a critical limiting factor for high-fidelity replication (Toledo et al., *Cell*, 2013), but this work finds that PCNA becomes limiting on chromatin prior to RPA. The authors then go on to identify PAF15 as a critical regulator of PCNA, showing it controls this limiting effect of PCNA and that it becomes depleted when replication origin firing is elevated. The authors demonstrate that PAF15 stabilizes PCNA at the fork, controlling its interactions with ATAD5-RFC, the PCNA unloader, and with DNA polymerases. Strikingly, they also find that PAF15 shows surprising strand specificity in these effects, acting to regulate the lagging-strand pool of PCNA and its interacting factors, while leaving the leading-strand pool unaffected. They elucidate the mechanism of this protection, showing the leading-strand pool of PCNA is protected from PAF15 by CLASPIN and TIMELESS and that the interaction of PAF15 with PCNA is important for its functions. Finally, the work shows that PAF15 levels are tightly regulated and that this control is important, since excess PAF15 leads to loss of replisome integrity and problems on the leading strand, while a decrease in PAF15 disrupts lagging-strand Okazaki fragment maturation. The control of PAF15 expression in cells is also determined.

This manuscript provides an exciting, unexpected and detailed understanding of the functions of PAF15 while also providing surprising and important insights into the regulation of PCNA on the leading and lagging strands. The work also elegantly elucidates the mechanism by which PAF1 controls PCNA dynamics and interactions on the two strands, and how it is regulated itself. The mechanism of PCNA control by PAF1 is itself novel and unlike that of other factors which interact with and regulate PCNA. Most important, perhaps, the work defines a previously unrecognized, rate-limiting factor in eukaryotic DNA replication that acts in a strand-specific manner to control global DNA synthesis. The authors use a broad range of experimental approaches, including single-cell imaging, proteomics, structural modeling, and functional genetics, and their work reveals how local replisome

composition is coupled to replication dynamics. Overall, I think manuscript makes an important conceptual advance that is appropriate for publication in Nature. With a few exceptions, the data also largely support the conclusions made.

We are thrilled to receive such a positive response to our work and sincerely thank the reviewer for the constructive and insightful suggestions, which have helped us to further improve the manuscript.

Major Comments

1. What is the normal stoichiometric relationship between PAF15 and PCNA?

Thank you for this insightful comment. We have quantitatively defined the stoichiometric relationship between PAF15 and PCNA in both whole-cell lysates and chromatin-bound fractions of asynchronous U2OS cells using Orbitrap Astral mass spectrometry-based deep proteomics (**Rebuttal Fig. 11; ED Fig. 8d**). These analyses reveal that PAF15 is markedly substoichiometric relative to PCNA, with an approximate PCNA:PAF15 ratio of ~9 in the total cellular pool and ~12 in the chromatin-bound fraction.

Rebuttal Fig. 11. Mass spectrometry analysis of PCNA/ PAF15 ratio in whole cell lysate and chromatin bound fraction. The data is derived from normalizing raw abundance to expected iBAQ peptides (PCNA=16, PAF15= 6). The data points represent three biological repeats.

Consistently, transcriptomic profiling across 85 independent cell lines using publicly available datasets demonstrates a similarly low PAF15-to-PCNA ratio at the RNA level (**Ext. Data Fig. 16f**), underscoring the intrinsic paucity of PAF15 as a rate-limiting factor within the PCNA interactome.

2. PAF15 overexpression phenotype: PAF15 transient overexpression is toxic and slows down nascent DNA synthesis rate. The authors predict that this might be due to PAF15 blocking PCNA interactions with POLE1 on the leading strand. When PAF15 is overexpressed, does it suppress PCNA-POLE1 interaction by PLA?

We really appreciate the reviewers suggestion. However, this experiment proved technically challenging. Under conditions of PAF15 overexpression, PCNA becomes rapidly and excessively stabilized on chromatin, resulting in high background and nonspecific PLA signals that preclude reliable quantification of specific PCNA–POLE1 interactions.

Moreover, while leading-strand synthesis by Pol ϵ is stimulated by its interaction with PCNA (Baris et al., *Nature* 2022; PMID: 35585232), structural analyses of the human Pol ϵ –PCNA holoenzyme (He Q, et al. *Nat Commun.* 2024. PMID: 39516490) demonstrate that the polymerase possesses intrinsic DNA-binding and processivity features that enable stable engagement with DNA even in the absence of PCNA. In contrast, the lagging-strand polymerase Pol δ and other lagging strand maturation factors are recruited to DNA by PCNA.

Consistent with this notion, our chromatin fractionation and immunoblot analyses revealed no detectable displacement of POLE1 from chromatin upon PAF15 overexpression. In contrast, the levels of canonical lagging-strand component such as POLD1 was markedly reduced despite enhanced chromatin retention of PCNA. These new results suggest that excessive PAF15 occupancy on PCNA preferentially interferes with the recruitment or stability of lagging-strand proteins, which depend more critically on PCNA for DNA binding and processivity (**Ext. Data Fig. 13e**).

Complementarily, co-immunoprecipitation experiments demonstrated that either loss of TIMELESS/CLASPIN or acute PAF15 overexpression drives mislocalization of PAF15 to the leading strand POLE1, concomitant with a pronounced replication fork slowdown revealed by DNA fiber assays (**Fig. 5f, g**). Together, these observations underscore that aberrant PAF15–PCNA dynamics perturb the coordination of both leading- and lagging-strand synthesis.

Therefore, in the revised manuscript, we have balanced the discussion to highlight that PAF15 overexpression compromises replisome dynamics at multiple levels—potentially altering leading-strand synthesis as well as disrupting several key interactions between PCNA and lagging-strand factors.

3. Mechanism of PAF15–POLE1 interaction upon TIMELESS loss: In Fi. 5d, TIMELESS loss increases PAF15–POLE1 PLA signal, presumably because TIMELESS shields leading-strand PCNA from PAF15. However, the authors also suggest that PAF15 and POLE1 bind PCNA mutually exclusively. How then does TIMELESS loss result in increased PAF15–POLE1 interaction?

We thank the reviewer for this excellent question. Our results suggest that TIMELESS and CLASPIN act together to keep PAF15 away from PCNA on the leading strand, where it would otherwise interfere with the essential interaction between PCNA and the leading-strand polymerase POLE1 (**Fig. 5f, g**). TIMELESS appears to act upstream by positioning CLASPIN at the replisome (**Ext. Data Fig. 15a**), potentially allowing CLASPIN to form a protective interface that stabilizes the POLE1–PCNA connection and prevents PAF15 from competing for the same binding surface.

Using AlphaFold modeling, we identified a region within CLASPIN (residues 304–341) that could directly contact PCNA (**Rebuttal Fig. 12**). This region may help maintain the proper alignment of POLE1 with PCNA, thereby ensuring continuous leading-strand synthesis. Although we attempted to generate CLSPN knockout U2OS cells to test this experimentally through complementation assays, we have so far obtained only hypomorphic clones (**Rebuttal Fig. 13**), consistent with CLASPIN's essential role in DNA replication.

Successful generation of a conditional or complete CLSPN knockout line would enable systematic structure–function analysis to dissect the specific contribution of the putative PCNA-interacting region to CLASPIN’s functions in opposing PAF15 at the leading strand.

Rebuttal Fig. 12. AlphaFold analysis of CLASPIN interaction with PCNA on dsDNA in presence of PAF15.

Rebuttal Fig. 13. Western blot of whole cell extracts from *CLSPN* KO screen. The data shows incomplete gene knockout efficiency in representative clones.

[Redaction]

4. Limitation of universality in PAF15 regulation: While data in Fig. 6 and Ext. Data Figs 19–20 support E2F4-mediated repression of PAF15, these data are somewhat distracting and actually rather peripherally related to the overall manuscript (which is already dense). I would suggest these data actually be removed.

We thank the reviewer for this comment, which is aligned with similar feedback from Reviewer 3. Reviewer 1 appreciated the inclusion of the E2F-related data, as it provides a conceptual basis for PAF15 rate-limiting behavior in cancer.

To incorporate suggestions from all reviewers, we have refined the manuscript by removing redundant E2F-related datasets to maintain focus on the central mechanistic aspects of PAF15 function.

As also detailed in our response to Reviewer 3, we have revised the text to provide a more measured interpretation. While the combined loss of PAF15 and E2F4 mitigate the phenotype observed upon E2F4 depletion alone, the underlying mechanism is likely multifactorial. The revised discussion now briefly summarizes this relationship to highlight its conceptual relevance while avoiding overinterpretation.

Technical/Minor Comments

1. Ext. Data Fig. 1c: It is unclear how the fork speed is measured. (Is it derived from the measurement of IdU track length of progressing forks?)

We have now clearly indicated on the Y-axis that both CldU and IdU tracts were accounted for when measuring replication fork speed.

2. Are Fig. 2a and Ext. Data Fig. 6f scatter plot identical?

Thank you for noticing this. The PCNA QIBC total and chromatin plots are indeed only shown in **Fig. 2a**.

3. Lines 235-236: In Ext. Data Fig. 9e, the higher PCNA signal in the western blot could be due to a more efficient PCNA antibody compared to PAF15 antibody. The reviewer thus does not agree with the authors' statement that "immunoblotting across various cell types, which consistently show higher chromatin levels of PCNA than PAF15" based on this data.

We have removed the immunoblotting data, and as indicated in response to point 1, we have quantitatively defined the stoichiometric relationship between PAF15 and PCNA in both whole-cell lysates and chromatin-bound fractions of asynchronous U2OS cells using high-resolution Orbitrap Astral mass spectrometry (**Ext. Data Fig. 8d**).

4. Lines 280-283: Lacks figure call out (Ext. Data Fig. 11f?). In addition, why is PCNA

chromatin loading increased in PAF15 overexpressing cells following siPAF15 (middle panel)?

Thank you again for noticing this. We have now repeated the experiment, and the new data—presented as **Ext. Data Fig. 10c**—are appropriately referenced in the revised manuscript.

5. Fig. 3c: DNA spreading, in which the leading- and lagging-strand nascent DNA collapse during the process, is used following S1 nuclease digestion. S1-nuclease sensitivity in PAF15-KO cells therefore indicates not only lagging-strand processing issue, but also indicates S1-sensitivity on the leading strand as well. How would the authors reconcile this with the hypothesis that PAF15 functions exclusively on lagging strand? The authors could better address this with combing (Meroni et al., PMID: 38315097), or alternatively change the text slightly.

Thank you again for this valuable comment. Indeed, as highlighted by the work of Meroni et al., detecting nascent gaps on a single strand requires DNA combing rather than DNA fiber assays, which, at first glance, may not fully align with the proposed role of PAF15 being exclusively linked to the lagging strand. Based on the reviewer's suggestion, we have revised the text to refer more broadly to nascent DNA gaps, noting that while these are most likely to arise first on the lagging strand, they could also occur on both strands. Moreover, sustained loss of PAF15 and associated lagging-strand defects may alter replisome dynamics, thereby promoting gap formation across both template arms.

However, motivated by the robust S1 nuclease results in PAF15-deficient cells and the role of PAF15 at the lagging arm of the replisome, we discussed this issue with Prof. Alessandro Vindigni (Meroni et al., PMID: 38315097). Through personal communication, he helped clarify a critical technical point: S1-based nuclear protocols involving cell detachment by trypsinization can promote partial strand separation, thereby increasing the apparent S1 sensitivity of single strands. Indeed, our protocol involves cell detachment followed by hypotonic treatment and *in situ* S1 digestion prior to DNA fiber stretching (Somyajit et al., *Dev Cell*, 2021; PMID: 33621493), which could explain our observed results.

For scientific accuracy, we have therefore modified the manuscript text to refer simply to nascent DNA or daughter strand gaps without implying strand specificity.

Referee #3:

This manuscript investigates whether the checkpoint pathways, particularly those mediated by ATR, control origin firing through the regulation of rate-limiting replisome components whose molecular identity remains incompletely defined. To address this, the authors examined how ATR inhibition affects the assembly and dynamics of replication factors, focusing primarily on PCNA loading and lagging-strand processing. A key methodological approach used throughout the study is quantitative image-based cytometry (QIBC), which enabled high-content and automated single-cell analysis to capture the detailed dynamics of replication stress responses. They demonstrated that ATR inhibitor (ATRi) treatment enhanced origin firing and activated CDK1/2, leading to increased chromatin association of replisome components such as Cdc45. However, PCNA levels on chromatin did not exhibit significant changes under these conditions. These findings suggest that when the ATR-mediated checkpoint is turned off, further loading of PCNA onto chromatin does not occur, despite increased origin firing and replisome assembly. Therefore, lagging-strand synthesis is compromised, resulting in defects in Okazaki fragment ligation and accumulation of DNA

damage.

Notably, the authors identified PAF15 as a dosage-sensitive regulator of PCNA. Under unperturbed replication conditions, nearly the entire soluble pool of PAF15 was bound to chromatin, where it stabilizes the association of PCNA with DNA and ensures proper lagging-strand synthesis. Part of this stabilization appears to be mediated by PAF15, opposing the action of the PCNA unloader ATAD5. However, when ATRi deregulated origin firing, PAF15 could not accommodate additional PCNA loading onto chromatin. Consequently, lagging-strand synthesis was disrupted, leading to incomplete Okazaki fragment maturation. Consistent with this, PAF15-deficient cells showed reduced levels of chromatin-bound PCNA and accumulation of unligated Okazaki fragments, accompanied by markers of replication stress. Interestingly, both overexpression of PAF15 and its mislocalization to the leading strand were found to impede replication fork progression and induce cell death.

These findings suggest that the role of the DNA replication checkpoint in suppressing excess origin firing is crucial to prevent exhaustion of the PAF15–PCNA axis on the lagging strand. Moreover, they indicate that a strand-specific, rate-limiting mechanism is closely linked to the global dynamics of chromosome replication.

Overall, the findings provide intriguing insights into how PCNA dynamics and lagging-strand processing are regulated under replication stress and reveal that PAF15 acts as a strand-specific, rate-limiting factor whose availability is crucial for maintaining genomic stability. However, several aspects of the study warrant further clarification and validation to strengthen the mechanistic conclusions and fully establish the broader significance of these observations.

We are delighted that the reviewer finds our work intriguing and are grateful for their constructive suggestions to further strengthen the conclusions.

Specific points:

1. The authors showed that siRNA-mediated ATAD5 depletion rescued replication catastrophe and PCNA chromatin loading defects induced by HU+ATRi or PAF15 knockout (Fig. 1g,h; Ext. Data Fig. 14a). However, as ATAD5 suppression is known to cause delays in replication progression (Lee et al., J. Cell Biol., 2013), the authors should clarify whether Okazaki fragment maturation was genuinely restored under these conditions or whether the observed rescue merely reflects slowed replication dynamics.

This is indeed a very valuable comment. We fully agree with the reviewer that ATAD5 suppression is known to cause delays in replication progression, and that the apparent bulk rescue of PCNA and lagging-strand factors in ATAD5-depleted PAF15 knockout cells may not directly reflect genuine restoration of Okazaki fragment maturation. To address this, we systematically tested the relationship between ATAD5 and PAF15 through multiple complementary approaches.

First, PLA-QIBC assays confirmed that the PCNA–DNA ligase 1 interaction was restored in S phase in PAF15 knockout cells when ATAD5 was simultaneously depleted (**Fig. 4f**). Second, S1-nuclease assays revealed that although ATAD5 loss alone caused replication perturbation and increased S1-nuclease sensitivity of nascent DNA, the combined loss of PAF15 and ATAD5 significantly reduced this sensitivity compared to PAF15 knockout cells alone, indicating a rescue of nascent DNA gap accumulation (**Fig. 4g**). Finally, co-depletion of PAF15 and ATAD5 also rescued the elevated 53BP1 foci observed in PAF15 knockout

cells (**Ext. Data Fig. 12c**), further supporting recovery from replication-associated DNA damage. Moreover, we provide detailed cell-cycle control profiles for all genetic conditions examined in these experiments (**Ext. Data Fig. 12b**).

Altogether, these data indicate that ATAD5 depletion largely rescues the replication defects observed upon PAF15 loss, supporting a functional antagonism between these factors in regulating PCNA unloading and lagging-strand maturation.

2. It has been reported that the regulate is regulated not only by its interaction with PCNA but also by its ubiquitination status. However, in the current study, most analyses and discussions regarding chromatin-bound PAF15 appear to focus exclusively on the non-ubiquitinated form detected at approximately 15 kDa. To fully interpret their findings, the authors should examine and present data on the ubiquitinated forms of PAF15 in their chromatin fractionation assays and discuss how these species might influence chromatin association and the function of PAF15, particularly in stabilizing PCNA under the tested conditions. Moreover, the absence of deubiquitinase (DUB) inhibitors in the lysis buffers and the use of RIPA buffer containing 150 mM NaCl may compromise detection of chromatin-bound ubiquitinated PAF15. The authors should consider repeating these experiments under milder extraction conditions and with appropriate DUB inhibitors to ensure accurate assessment of both ubiquitinated and non-ubiquitinated forms of PAF15.

This is an important point, and we fully agree with the reviewers that ubiquitination of PAF15 by UHRF1 plays a crucial role in regulating PAF15 functions. In most of the immunoblots presented, we have shown the double-ubiquitinated forms of PAF15 (for example, **Fig. 1h, 4d, 5a; Ext. Data Fig. 8e, 8f, 13e, 16b**).

To directly address the reviewers' point regarding whether loss of PAF15 ubiquitination affects its chromatin loading, we depleted UHRF1 using two independent siRNAs and assessed PAF15 chromatin association by QIBC. QIBC provides a quantitative and cell cycle–resolved measurement of chromatin loading specifically in S-phase cells, thereby avoiding potential confounding effects arising from cell-cycle differences. Furthermore, to ensure that detection of PAF15 was not influenced by its ubiquitination status, we used an antibody whose epitope lies outside the N-terminal region where PAF15 is ubiquitinated. Specifically, the Santa Cruz antibody sc-390515 (mouse monoclonal, clone G-11) recognizes an epitope mapped to amino acids 58–78 of PAF15/PCLAF. As shown below, while UHRF1 depletion completely abolished PAF15 double-ubiquitination, it had a negligible impact on the chromatin loading of either PAF15 or PCNA during S phase (**Rebuttal Fig.14 a, b**).

Rebuttal Fig. 14. a, Western blot analysis of whole cell extracts from U2OS cells treated with control, *UHRF1-A*, *UHRF1-B* siRNAs. **b**, QIBC of chromatin bound PAF15 (left) and PCNA (right) in U2OS cells treated with control, *UHRF1-A*, *UHRF1-B* siRNAs. Thin dotted lines indicate average values. A total of > 5,000 cells were analysed per condition; 2n: G1, 4n: G2.

However, consistent with a role for PAF15 ubiquitination in supporting DNMT1 function (Nishiyama et al., *Nat. Commun.* 2020; PMID: 32144273), we find that loss of UHRF1 completely abolishes DNMT1 chromatin loading in S phase. Similarly, loss of PAF15 compromises DNMT1 chromatin association, specially evident upon DNMT1 immobilization using decitabine (**Rebuttal Fig.15; Ext. Data Fig. 8g**).

Rebuttal Fig. 15. QIBC analysis of chromatin-bound DNMT1 in parental and PAF15 KO U2OS cells treated with siRNA against *UHRF1* and exposed to Decitabine (5-aza-2'-deoxycytidine) for 60 min. 2n corresponding to G1 phase and 4n to G2 phase; >10,000 cells per condition.

These findings indicate that PAF15 may start associating with PCNA at replicating chromatin before it becomes ubiquitinated. The subsequent ubiquitination of PAF15, particularly on the lagging strand, may then link PAF15 more directly to DNMT1-related functions and the maintenance of DNA methylation on newly replicated DNA, consistent with previous reports. We also find that loss of PAF15 ubiquitination parallels the loss of DNA ligase 1 (**Ext. Data Fig. 8e, f**), supporting a functional connection between PAF15 modification and lagging-strand processing. Thus, we propose that while PAF15 ubiquitination by UHRF1 promotes DNMT1 recruitment to replicating chromatin, the initial chromatin association of PAF15 during S phase does not appear to be strictly dependent on its ubiquitination status.

Notably, these observations open an important direction for future work that provide a framework for investigating how the lagging strand contributes to the coupling of DNA replication and methylation.

3. The authors observed that DNMT1 levels on chromatin remain unchanged under conditions of excess origin firing. However, DNMT1 is not exclusively a lagging-strand-associated factor, and it would be important for the authors to clarify why DNMT1 does not similarly increase. This raises questions about potential strand-specific regulation of DNMT1 loading or broader changes in replication dynamics under these conditions. Furthermore, given that PAF15 has been implicated in pathways involved in DNA methylation maintenance, its loss might be expected to result in global or locus-specific DNA hypomethylation.

[Redaction]

[Redaction]

Although unexpected, these observations are also consistent with recent structural studies that have reconstituted a complex containing DNMT1, hemimethylated DNA, doubly monoubiquitinated PAF15, and PCNA—and found that such arrangement is potentially more compatible with the architecture of the lagging arm of the replisome (Ruiz-Albor et al., *Int J Biol Macromol* 2025; PMID: 40543772)

It would be informative to examine how depletion or inhibition of key DNA methylation regulators, such as UHRF1 or DNMT1, affects PCNA chromatin binding

As suggested by the reviewer, we directly tested this, and as shown below, loss of UHRF1 or DNMT1 does not lead to any detectable defects in PCNA chromatin loading during S phase (**Rebuttal Fig. 19; Ext. Data Fig. 8h**).

Rebuttal Fig. 19. QIBC of cell cycle phase chromatin-bound PCNA (left) in U2OS parental cells treated with control, *UHRF1*, *DNMT1* siRNAs. A total of > 5,000 cells were analysed per condition; 2n: G1, 4n: G2. The dotted horizontal lines represent average S-phase chromatin-bound PCNA levels. Quantification of chromatin bound PCNA (right) levels in different phases of cell cycle from QIBC (left).

4. The authors showed in Fig. 3g and Ext. Data Fig. 12d that the combination of PAF15 knockout and PARP inhibition led to increased formation of micronuclei and γ H2AX accumulation, indicating elevated genomic instability. The authors should clarify whether unligated Okazaki fragments indeed accumulate in the context of combined PAF15 loss and PARP inhibition.

Based on the suggestions from this reviewer and Reviewer 1, we have extended our analysis of the impact of PAF15 loss and PARP1 inhibition by assessing S1-nuclease sensitivity of nascent DNA and RAD51 foci formation in S phase, providing further insight into how loss of canonical and non-canonical pathways enforces post-replication daughter-strand gaps. These new results have been incorporated into the revised manuscript (**Fig. 3 f, g**).

5. In Fig. 5, the authors showed that suppression of TIMELESS led to increased interaction between PAF15 and DNA polymerase ϵ (Pol ϵ), which they interpreted as evidence that PAF15 is aberrantly recruited to the leading strand. However, it remains unclear whether an alternative possibility has been excluded—namely, that Pol ϵ itself might instead relocate to the lagging strand under these conditions. The authors should clarify how they have ruled

out this possibility. Moreover, it would be essential to validate their interpretation that PAF15 is acting on the leading strand by employing complementary experimental approaches.

Under unperturbed conditions in the absence of exogenous DNA damage, Pol ϵ preferentially functions as a bona fide leading-strand polymerase. However, it remains unclear whether loss of TIMELESS or overexpression of PAF15 can perturb this strand specificity and misdirect Pol ϵ toward the lagging strand.

In response to the important point regarding the validation of aberrant PAF15 function on the leading strand, we have directly addressed this question using multiple complementary approaches designed to test both the biochemical and functional relationship between PAF15 localization and replisome organization.

First, our co-immunoprecipitation analyses consistently showed an increased association of PAF15 with the leading-strand polymerase POLE1 under conditions where replisome architecture was perturbed—either through depletion of TIMELESS and CLASPIN or through acute overexpression of PAF15 following doxycycline induction (**Fig. 5f**). These observations strongly suggest that, in the absence of the TIMELESS–CLASPIN complex or when PAF15 levels are excessive, PAF15 aberrantly associates with the leading-strand machinery.

Second, to determine the functional consequences of this mislocalization, we examined replication fork dynamics under conditions of altered PAF15 abundance and TIMELESS depletion. Strikingly, replication fork speed was markedly and consistently reduced when PAF15 was overexpressed together in the absence of TIMELESS, far exceeding the reduction observed upon alone in each condition (**Fig. 5g**). This synergistic effect suggests that TIMELESS (and CLASPIN) normally acts as a barrier that confines PAF15 to its physiological location on the lagging strand (also discussed in response to Reviewer 2 point 3).

Finally, to explore this question from an orthogonal perspective, we have incorporated into the revised manuscript SCAR-seq profiling of PAF15 integrated with OK-seq analysis in mouse embryonic stem cells. These complementary approaches provide strand-specific information on protein enrichment relative to replication fork directionality. Under physiological conditions, PAF15 displayed a strong strand bias, being preferentially partitioned to the lagging-strand arm of the replisome (**Fig. 2f, g; Ext. Data Fig. 8b, c**). These results are fully consistent with our PLA data in human cells, which independently demonstrate that PAF15 localization under normal conditions is confined to the lagging strand, and this feature is likely conserved in mammalian cells (**Fig. 2e**).

To extend these findings, we also attempted to perform SCAR-seq in rapidly induced PAF15-overexpressing cells using an anti-FLAG antibody to assess whether overexpression alters the lagging- versus leading-strand distribution of PAF15. However, these experiments proved technically demanding and will require further optimization of induction parameters and the timing of induction, since acute PAF15 overexpression rapidly stabilizes PCNA and influences replication dynamics.

6. In Fig. 6, the authors showed that E2F4 transcriptionally regulates PAF15 and that double knockdown of PAF15 and E2F4 rescued the phenotype associated with PAF15 depletion. However, given that E2F4 is known to regulate numerous genes involved in DNA replication and cell cycle control, it is difficult to conclude that the observed rescue effect is mediated solely through changes in PAF15 expression. The authors should consider whether the effects of E2F4 knockdown might also involve alterations in other replication-associated pathways or factors. Such broader effects could potentially confound the interpretation that PAF15 is the primary target responsible for the observed phenotypes.

We thank the reviewer for this excellent point, which is also aligned with the comments raised by Reviewer 2.

Based on these suggestions, we have toned down our interpretation to acknowledge that, while combined loss of PAF15 and E2F4 markedly rescues the phenotype observed upon E2F4 loss alone, the underlying mechanism is likely complex.

Indeed, E2F4 depletion leads to the derepression of several DNA replication–related genes, including replisome components and priming factors such as PRIM1 and PRIM2 (**source data, Fig. 6e**), which may elevate replication capacity. Conversely, loss of PAF15, a key regulator of PCNA stability and function, may offset this hyperactivation by destabilizing PCNA at replication forks—thereby limiting the impact of excessive and unscheduled replication events and potentially explaining the observed compensatory relationship between PAF15 and E2F4. Of note, our findings are conceptually aligned with data in Fan et al., *Nature Communications* (2020; PMID: 32665547), which demonstrated that loss of E2F6, a component of the E2F4 repressive complex, enhanced replication capacity but also induced replication stress in daughter cells.

In light of these insights and the reviewer’s feedback, we have streamlined the revised manuscript by removing redundant datasets related to E2F analysis and have balanced the discussion accordingly, acknowledging that additional mechanisms may also contribute to this phenotype.

Minor points:

1. In the current study, the authors tagged PAF15 with an N-terminal Myc-FLAG epitope. However, the N-terminus of PAF15 contains motifs critical for recognition by UHRF1, which mediates its ubiquitination. Adding an N-terminal tag may interfere with UHRF1 binding and consequently alter the ubiquitination status and chromatin association of PAF15. The authors should consider performing key experiments with a C-terminally tagged version of PAF15 to ensure that the observed interactions and functional effects are not artifacts resulting from N-terminal tagging.

We apologize for the earlier lack of clarity. All PAF15 constructs used in this study are C-terminally tagged, which is now explicitly stated in the Methods section and unified in the labeling throughout the manuscript (e.g., PAF15-Myc-FLAG or PAF15-FLAG). We fully agree with the reviewer that an N-terminal epitope tag would likely interfere with UHRF1-mediated PAF15 ubiquitination. To illustrate this point, we show below here that complementation of *PAF15* KO cells with PAF15-FLAG preserves fully functional PAF15 double mono-ubiquitination (**Rebuttal Fig 20; Ext. Data Fig. 13e**).

Rebuttal Fig. 20. Western blot analysis of PAF15 and its mono and dual ubiquitination forms in chromatin fractions from U2OS parental and U2OS *PAF15* KO with doxycycline inducible expression of FLAG tagged PAF15 cells.

2. Previous reports have shown that methylation of *LIG1* promotes the recruitment of *UHRF1*, which acts as the E3 ubiquitin ligase for PAF15. However, in the current study, the authors showed that a *LIG1*- Δ TARK mutant did not affect PAF15 ubiquitination, which seems inconsistent with this established pathway. Although the authors do not focus on this aspect, it would be helpful if they could comment on how this observation might be reconciled with previous findings.

We really thank the reviewer for pointing this out. We are currently constructing cell-line carrying a point mutation abrogating *UHRF1*-DNA ligase 1 interaction, before drawing firm conclusions about its impact on PAF15 ubiquitination, which, as the reviewer also notes, lies outside the scope of the present manuscript.

In the meantime, we have repeated the western blots in mESC *LIG1* and *UHRF1* knockout cells, as well as in the PAF15-K1524RR mutant, and consistently observed the same effects on PAF15 ubiquitination (**Ext. Data Fig. 8f**). We further confirmed these complementary findings in human U2OS cells (**Ext. Data Fig. 8e**).

3. In Ext. Data Fig. 9b, the authors showed that mutation of the KEN box in PAF15 led to increased protein stability but did not substantially enhance chromatin binding or recapitulate the phenotypes seen with PAF15 overexpression. It would be helpful for the authors to discuss why stabilization of PAF15 through KEN box mutation did not produce the same functional effects as overexpression, and whether this suggests that regulation of PAF15 activity involves mechanisms beyond simple protein abundance.

Thank you for pointing this out. Indeed, while loss of *APC/C^{Cdh1}*-mediated targeting of PAF15 results in its stabilization during G1 phase (constitutive expression complementation; **Rebuttal Fig. 21a**), this does not lead to a corresponding increase in PAF15 chromatin association under either normal or under excessive origin firing upon ATRi conditions (**Rebuttal Fig. 21b**). This reflects that PAF15 recruitment to chromatin occurs specifically through PCNA at active replisomes, downstream of origin firing and PCNA loading. Thus, stabilization of PAF15 in G1/M does not influence its assembly or function on replicating chromatin during S phase. In line with this, excessive stabilization of PAF15 in G1 does not recapitulate the effects of doxycycline-induced rapid overexpression (WT or KEN mutant), where both total levels and chromatin loading in S-phase are concurrently enhanced.

Rebuttal Fig. 21. a, QIBC of total pool of native PAF15 in U2OS parental cells and PAF15 wildtype, PAF15-KEN (K78A) mutant versions complemented in *PAF15* KO cells (left). A total of > 10,000 cells were analysed per condition; 2n: G1, 4n: G2. The dotted horizontal lines represent average total pool of PAF15. Line plot denotes the levels of PAF15 versions (native, WT, KEN mutant) in different phases of cell cycle (right) from QIBC (left). **b**, QIBC of cell cycle phase chromatin-bound PAF15 in U2OS parental and PAF15 wildtype, PAF15-KEN (K78A) mutant complemented in *PAF15* KO cells. A total of > 5,000 cells were analysed per condition; 2n: G1, 4n: G2. The dotted horizontal lines represent average S-phase chromatin-bound levels of PAF15 versions.

We also concur with the reviewer that regulation of PAF15 activity is likely governed by mechanisms beyond simple alterations in protein abundance. Importantly, PAF15 represents one of the very few core replisome factors known to undergo APC/C^{Cdh1}-mediated degradation. To address this unique regulatory feature, we are systematically investigating the functional consequences of impaired APC/C^{Cdh1}-dependent turnover of PAF15, with particular emphasis on the G1 phase and the G1/S transition, as part of an ongoing separate study.

In line with the reviewer's suggestion, this point has now been briefly discussed in the revised manuscript.

4. The authors should cite Ext. Data Fig. 11f in the main text (lines 280–283) where the data on PAF15 overexpression and increased PCNA/LIG1 chromatin binding are discussed.

This data, which is now **Ext. Data Fig. 10c**, is appropriately referenced in the revised manuscript.

5. Ext. Data Figs 14c and 15c,d refer to “ Δ N PAF15 (2–61),” but this mutant is not explained in the main text and may be mislabeled. The authors should clarify its identity and functional significance.

We thank the reviewer for this comment. In our original submission, we presented data showing that deletion of the full N-terminus of PAF15 (Δ N 2–61) markedly reduced protein stability. Subsequent mapping revealed that deletion of only the first 11 amino acids (Δ N 2–11) was sufficient to confer the same destabilizing effect. As these datasets were largely redundant, and in order to maintain clarity and avoid overcrowding in the manuscript, we have now removed the Δ N 2–61 data and retained the Δ N 2–11 data in the revised version (**Ext. Data Fig. 12e**).

Referee #4:

I co-reviewed this manuscript with one of the reviewers who provided the listed reports.

We sincerely thank the co-reviewer reviewer 4 for their time and valuable feedback.

Point-by-point response the Reviewers:

Once again, we sincerely thank the editor and all four reviewers for their continued enthusiasm for our work and for the thoughtful comments that have greatly strengthened our manuscript. We are delighted that the reviewers' assessments, together with the remaining minor points, indicate broad satisfaction with our revisions and support the publication of our study.

Details on how we have fulfilled all editorial requirements regarding formatting and space limitations are provided in the cover letter to the editor. Below, we respond directly to the reviewers' comments, with all responses highlighted in blue text.

Referee #1:

The authors have adequately addressed all the concerns I raised and have substantially strengthened their manuscript. Notably, they provide compelling new insights regarding points #7 (the roles of PAF15 and CLASPIN in cancer) and #9 (the interplay between PAF15 and RIF1). While these findings are intriguing, I agree with the authors that they extend beyond the scope of the current study and are best explored in a separate work. In my view, the manuscript is now ready for publication, but references 15, 55, 62 still need to be completed.

This referee is satisfied with our revision and has no additional specific comments. We are very grateful for their inspiring input throughout the review process. We also thank them for noticing that several references had not been updated; these have now all been corrected.

Referee #2:

I have gone back over the manuscript and the authors' revisions. They have addressed all of my concerns appropriately and in some cases extensively and I am very happy with the revised work.

This reviewer is likewise pleased with the revised manuscript and has no further specific comments. We sincerely appreciate their enthusiasm for our work and the thoughtful experimental suggestions they provided earlier in the review process, which helped strengthen the manuscript considerably.

Referee #3:

Most of my major concerns have been addressed, and the revised manuscript now provides sufficient support for the main conclusions. However, some points still require clarification.

In the revised version, the authors demonstrated that ATAD5 depletion largely rescues the replication defects observed upon PAF15 loss. Nevertheless, Lee et al. (*J. Cell Biol.*, 2013) reported that ATAD5 depletion promotes PCNA loading but also leads to the formation of inactive replication factories and delayed replication progression. It therefore remains unclear whether the replication factories restored by ATAD5 depletion in this study are indeed functionally active.

We appreciate that the reviewer is satisfied with how we strengthened the revised manuscript and reinforced the main conclusions.

We fully agree that ATAD5 depletion is known to promote the accumulation of inactive replication factories when PCNA is not properly unloaded. However, our results reveal an important mechanistic distinction. In the absence of PAF15, PCNA is intrinsically unstable on chromatin, and under these conditions ATAD5-mediated PCNA unloading becomes detrimental rather than protective. Premature or uncoordinated removal of PCNA displaces essential lagging-strand maturation factors—including FEN1 and DNA ligase 1—from the replisome, thereby worsening replication defects.

Importantly, and as the reviewer also highlights, we show that these replication defects are largely rescued when PAF15 and ATAD5 are co-depleted. Although ATAD5 loss alone is known to promote the formation of inactive replication factories by driving excessive chromatin-bound PCNA and loss of OkF factors, the combined loss of PAF15 and ATAD5 instead restores this balance. Under co-depletion conditions, the chromatin retention and interaction between PCNA and OkF processing factors increases significantly, and replication phenotypes observed upon PAF15 loss alone are markedly improved. These findings indicate that, in a PAF15-deficient background, attenuating PCNA unloading supports a functional replisome rather than reinforcing the inactive replication factories typically induced by ATAD5 loss alone. Moreover, our data are consistent with recent work from the Diffley laboratory (Bertolin et al., *Mol Cell* 2025; PMID: 40578346), which similarly shows that ATAD5 depletion can mitigate replication-stress phenotypes under conditions of checkpoint impairment.

In line with the reviewer's insightful comment, we also want to highlight that our findings uncover a previously unappreciated regulatory interplay between PAF15-dependent stabilization of PCNA and ATAD5-mediated PCNA unloading. We propose that this dynamic balance constitutes a central mechanism for maintaining replisome integrity and opens several exciting avenues for future mechanistic exploration.

Moreover, if PAF15 indeed prevents premature, ATAD5-dependent PCNA unloading as proposed, the PAF15-PCNA complex should engage with LIG1, FEN1, and polymerases to promote lagging-strand synthesis. Although the authors suggest in Ext. Data Fig. 14 that these interactions are structurally compatible with AlphaFold models, it would further strengthen the study if they provided experimental evidence, such as co-immunoprecipitation, to confirm their existence.

We thank the reviewer for this constructive suggestion. As requested, we performed CO-IP experiments in PAF15-myc-3×FLAG-expressing cells using whole-cell lysates, and the resulting data clearly demonstrate that PAF15 forms a complex with PCNA during active lagging-strand processing. These findings provide direct experimental validation of the interactions predicted by our AlphaFold models. Moreover, the new co-IP results are fully consistent with our proposed mechanistic framework, in which PAF15 acts as a lagging-strand-specific regulator of PCNA dynamics by binding an available PCNA monomer and thereby coordinating PCNA accessibility for key lagging-strand maturation enzymes such as Pol δ and FEN1. In this model, PAF15 helps partition PCNA-dependent processes asymmetrically between the leading and lagging strands, ensuring efficient Okazaki fragment processing and maintaining the balance of factors competing for PCNA on the lagging strand. These new experimental data reinforce this framework, which is further supported by our PLA-QIBC assays and SCAR-seq-based genomic analyses, both independently indicating that PAF15 is preferentially partitioned to the lagging arm of nascent DNA. Together, these complementary and mutually reinforcing lines of evidence robustly strengthen our mechanistic model of PAF15 as a critical determinant of lagging-strand PCNA engagement. The new experimental results have been incorporated into **Fig. 2h**.

For completeness, we also cite earlier work (Hosokawa et al., *Cancer Research* 2007; PMID: 17363575) that reported PAF15-associated enrichment of Pol δ , FEN1 and PCNA, which is broadly consistent with our observations.

Referee #4:

I co-reviewed this manuscript with one of the reviewers who provided the listed reports.